

# Effect of sampling error on ozone partial pressure trends within a unified ozonesounding dataset

Fabrizio Marra[1], Emanuele Tramutola[1], Marco Rosoldi[1], Fabio Madonna[1,2]

[1]Consiglio Nazionale delle Ricerche – Istituto di Metodologie per l'Analisi Ambientale, C.da S. Loja - Zona Industriale, I-85050 Tito Scalo (Potenza), Italy.
[2]Department of Physics, University of Salerno, 84084, Salerno, Italy.

*Correspondence to*: Fabrizio Marra (fabrizio.marra@cnr.it)

**Abstract.** This work discusses the impact of the sampling frequency on ozone partial pressure trends, estimating its impact at various latitudes and vertical layers in the upper troposphere/lower stratosphere (UT/LS) region. The trends are estimated in the periods 1978-1999 and 2000-2022, using a new unified dataset combining the ozonesounding profiles provided by SHADOZ (Southern Hemisphere ADditional OZonesondes - https://tropo.gsfc.nasa.gov/shadoz/), NDACC (Network for the Detection of Atmospheric Composition Change - https://ndacc.larc.nasa.gov/) and WOUDC (World Ozone and Ultraviolet Radiation Data Centre - https://woudc.org/). These datasets are combined to offer adequate coverage at various latitudes and to enhance the estimation of anomalies and trends in ozone concentration on a global scale. The available measurements are classified into three groups based on the temporal coverage of historical time series. Some regression approaches are utilised to estimate trends and the related difference to quantify structural uncertainty. Significant trends for the period 1978-1999 are estimated for the Northern Hemisphere mid-latitude (NH), which shows a negative trend of 5% per decade in the layer 50-1 hPa and a negative trend of 10% per decade at 100-50 hPa, and for the Tropics (TR), which shows a positive trend of about 5% per decade at 50-1 hPa and 7% per decade at 100-50 hPa, respectively. Furthermore, the sampling error between the clusters was investigated, revealing a small effect of less than 2% at 100-50 hPa and 1.5% at 50-1 hPa for NH and about 3% at 100-50 hPa and 3.5% at 50-1 hPa for TR, as well as the structural uncertainty between the regressors used, 1.05% at 100-50 hPa and 1.15% at 50-1 hPa for NH and about 2% at both 100-50 hPa and 50-1 hPa for TR.

## 1 Introduction

Ozone changes are crucial for comprehending climate dynamics, as ozone plays a significant role in regulating atmospheric processes. Ozone in the upper troposphere/lower stratosphere (UT/LS) acts as both a greenhouse gas and a shield against harmful solar ultraviolet (UV) radiation (WMO, 2014). Changes in ozone concentrations in troposphere and stratosphere can impact temperature patterns, atmospheric circulation, and the distribution of other gases, thereby influencing climate variability and long-term trends (Salby, 1995; IPCC/TEAP, 2005). In recent decades, significant progress has been made in measuring ozone concentrations through the integration of satellite and ground-based measurements, which provide a more comprehensive understanding of ozone dynamics across various atmospheric layers and geographical regions (Staehelin et



al., 2018; Weber et al., 2018). Numerous studies have delved into estimating both regional and global changes in ozone in the UT/LS, using several satellite and ground-based datasets acquired through diverse measurement techniques (e.g., Sofieva et al., 2017b; Steinbrecht et al., 2017; WMO, 2018; Arosio et al., 2019; Petropavlovskikh et al., 2019; Sofieva et al., 2021; Godin-Beckmann et al., 2022). These studies employ various regression methodologies, including linear parametric and non-
parametric regressions, as well as multiple linear regressions, to estimate trends in ozone concentrations. For instance, Godin-Beckmann et al. (2022) utilized the Long-term Ozone Trends and Uncertainties in the Stratosphere (LOTUS) multiple linear regression model, version 0.8.0, alongside several satellites, in situ, and ground-based datasets to estimate stratospheric ozone profile trends.

The inherent variability of ozone in both time and space underscores the necessity for long-term accurate measurements to
detect trends in ozone concentrations and evaluate their significance. Spatial and temporal coverage of ozone datasets emerge as critical factors in this assessment. For instance, Petropavlovskikh et al. (2019) employed selection criteria for different measurement methodologies to ensure consistent trend estimates. In the case of ozonesondes, ascents failing to reach 10-20 hPa are often disregarded due to their inability to be compared or normalized to the total ozone column. Monthly aggregation of ozone data helps mitigate sampling error, by ensuring a sufficient number of profiles are available.
However, unlike regular radiosoundings, ozonesonde measurements are performed less frequently, with profiles collected typically at 1-3 ascents per week.

This study presents a methodology focused on reducing sampling errors in ozone partial pressure trends by merging existing ozonesounding datasets into a unified database. This approach enhances spatial and temporal coverage, thereby enabling more robust analyses of ozone anomalies and trends at a global scale compared to using individual datasets. Providing
climate researchers with a harmonized quality-checked dataset eliminates the need to download multiple datasets, streamlining their workflow. Furthermore, the unified database is a comprehensive resource for ozonesounding data and metadata, facilitating satellite validation efforts and future data reprocessing initiatives aimed at homogenizing time series. It incorporates ozone vertical profiles sourced from various datasets, thereby enriching the dataset's depth and utility for atmospheric chemistry and climate science research, including the Southern Hemisphere Additional OZonesondes
(SHADOZ), Network for the Detection of Atmospheric Composition Change (NDACC), and World Ozone and Ultraviolet Radiation Data Centre (WOUDC) datasets. Exploratory analysis of trends across latitudes and pressure vertical ranges is conducted, while structural uncertainties are scrutinized through comparisons of trends calculated using different linear regression approaches. The estimated trends are further contextualized regarding their significance and compared with values reported in recent literature.

The paper is structured as follows: Section 2 delineates the dataset utilized in the study, encompassing total coverage, station-by-station coverage, quality checks applied to filter the data, and the representativeness of the stations; Section 3 expounds on the regression methods employed to estimate trends; Section 4 presents the results of the comparative analysis of the aforementioned trends; and Section 5 offers concluding remarks.





## 2 Ozonesonde dataset

### 2.1 Unified ozonesonde dataset

The dataset used in this work results from the merging of three existing datasets that give adequate data coverage in different latitude sectors, hereby facilitating more robust analyses of anomalies and trends at a global scale. The unified dataset was built from the ozonesoundings datasets of SHADOZ, NDACC, and WOUDC.

The SHADOZ network provides ozone profiles measured at 17 sites since 1998. In addition, each of the soundings also yields pressure, temperature, and relative humidity profiles for all stations. The SHADOZ network was designed to reduce data heterogeneity in ozonesonde profiles between several sites measuring in the tropics and subtropics, and the network data was recently reprocessed to homogenise the database further (Thompson et al., 2017; Witte et al., 2017; Sterling et al., 2018; Witte et al., 2018). The SHADOZ V6.0 has been considered in the unified dataset, which provides measurement uncertainties for 14 of the 17 sites, with the former in operation for over a decade. SHADOZ is the only one of the three datasets providing a detailed estimate of the observational uncertainty for each data record.

NDACC officially started operation in 1991 (although ozonesounding profiles have been available since 1966), and comprises more than 90 globally distributed, ground-based research stations including 33 ozonesounding stations, whose data were processed in this study. Despite the high data quality, NDACC does not routinely provide uncertainties for ozonesoundings measurements. An uncertainty estimation is provided only for a subset of measurements (De Mazière et al., 2018).

WOUDC, which is part of the Global Atmosphere Watch (GAW) programme of the World Meteorological Organization (WMO), was established in 1960 to collect, quality-control, archive and provide long-term access to high-quality observation data and metadata from the WMO GAW network of stations measuring ozone column and ozone vertical profiles. The WOUDC archive provides data from more than 500 stations since 1962, including the 150 ozonesounding stations whose data were analysed in this work. WOUDC is the most comprehensive initiative for collecting ozonesoundings measurements that, although quality checked for their consistency in the data and metadata are less characterised in terms of traceability and uncertainties. However, data providers are recommended to collect measurements following the guidelines provided by WMO (https://woudc.org/about/data-policy.php).

Handling duplicated profiles poses a significant challenge in merging the three ozonesounding datasets into a harmonized database. Duplications often arise when observations from the same station are transmitted to multiple networks, leading to discrepancies in transmission periods, data formats, and the number of data points, along with varying metadata. Additionally, not all networks provide identical number of data points at each level. To overcome these challenges and create a unified dataset, the following selection rules were applied:

- If, for two-hours interval, only one profile is available, i.e. no duplicates, this is quality checked before the storage in the database.
- In the case of duplicated profiles, the selection process prioritizes the profile with the highest number of successful quality checks (QCs) for inclusion in the unified dataset.




- If multiple duplicates meet the same number of QCs, the selection is guided by factors such as the dataset's maturity (Thorne et al., 2017) and the availability of measurement uncertainties on ozone concentration profiles. While

uncertainties are primarily sourced from the SHADOZ database, they may also be accessible in the NDACC database in certain instances.

Table 1 offers a summary of the contribution of NDACC, SHADOZ and WOUDC datasets to the unified data archive.

**Table 1. Overview of the networks used for the unified database, including key details for each dataset: number of available stations, coverage period, number of provided profiles, availability of uncertainty data and percentage of network contribution to the total unified database.**

| DATABASE | # STATIONS | PERIOD | # PROFILES | UNCERTAINTY | % UNIFIED |
|---|---|---|---|---|---|
| SHADOZ | 14 | 1998-2022 | 9343 | Yes | 14.97% |
| NDACC | 33 | 1969-2020 | 45382 | Yes (for a minor fraction of data). | 46.21% |
| WOUDC | 150 | 1962-2022 | 97252 | No | 38.82% |

Thanks to the contribution of the three networks, the unified database has a total of 155 stations, for 1962-2022, and 104373 ozonesondes profiles.

To show the benefits of using the unified ozonesounding database, an example of the comparison of the time series of ozone profiles for the Hilo station (19.57°N, 155.07°W, 11m a. s. l., WMO code: 0-20008-0-HIH) available from SHADOZ, NDACC, and WOUDC with the corresponding time series obtained in the unified database is shown in Figure 1.





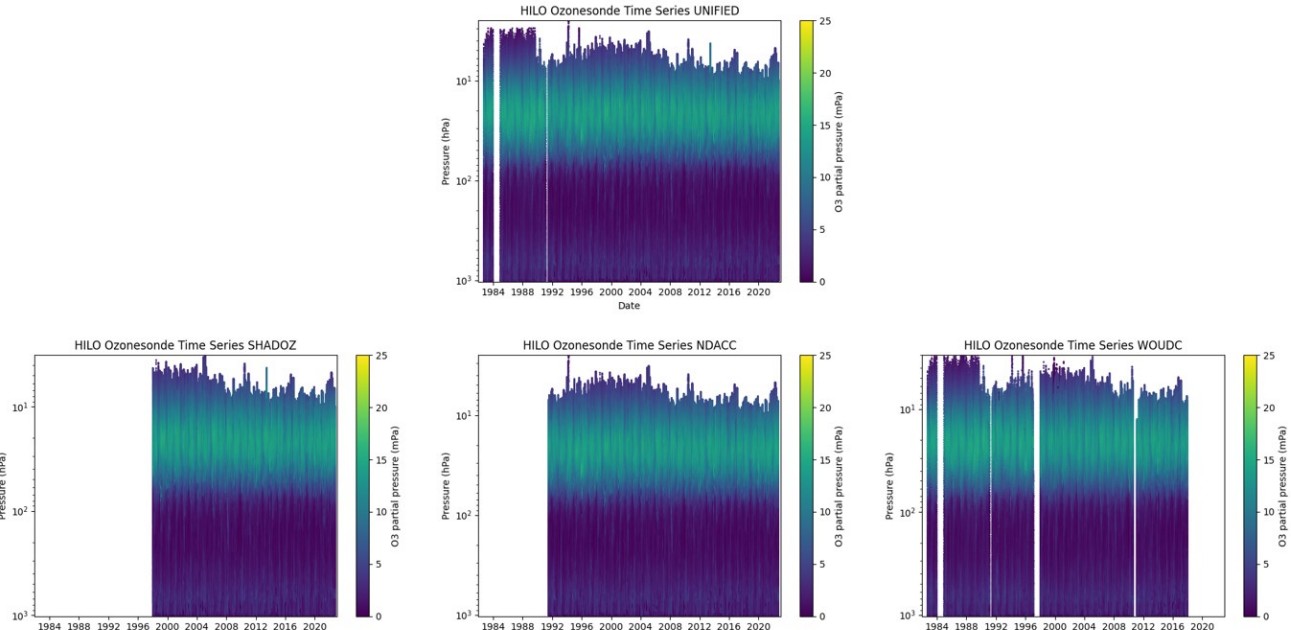

**Figure 1. Hilo station (19.57°N, 155.07°W, 11m a. s. l., WMO code: 0-20008-0-HIH) ozone profiles time series for the unified database (top panel), SHADOZ (bottom left), NDACC (bottom centre), and WOUDC (bottom right) datasets. The time series of the unified database is generated by the unification algorithm with the contribution of the ozonesonde profiles from SHADOZ, NDACC, and WOUDC networks. The contributions of the profiles to the time series of the unified database can be divided into four different periods: 1983-1991 only from WOUDC; 1991-1997 from NDACC and WOUDC; 1998-2018 from all networks; 2018-2022 from SHADOZ and NDACC. In case of duplicates, the highest quality profiles according to the selection rules introduced above are selected.**





## 2.2 Database data coverage

Figure 2 shows the number of ozonesounding profiles available in the unified database at a global scale.

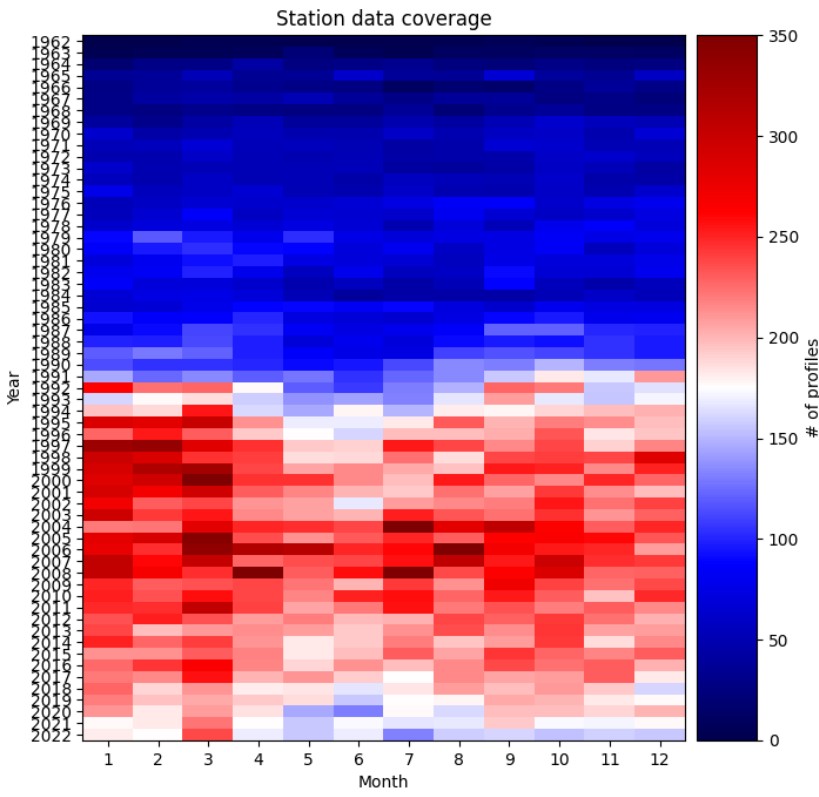

**Figure 2. The number of profiles available per month in the unified ozonesounding database since 1962.**


The increase in the number of profiles in the unified database post-1992 correlates with a corresponding increase in the number of ozonesounding stations. Additionally, temporal data coverage was assessed across four vertical ranges (300-200 hPa, 200-100 hPa, 100-50 hPa, and 50-1 hPa) and five latitudinal zones (Northern Hemisphere polar latitudes (NP; 60°N-90°N), Northern Hemisphere mid-latitudes (NH; 30°N-60°N), Tropics (TR; 30°S-30°N), Southern Hemisphere mid-latitudes
(SH; 60°S-30°S), and Southern Hemisphere polar latitudes (SP; 90°S-60°S)).

As an example, Figure 3 illustrates the temporal coverage of the data in the vertical range 50-1 hPa, which is indicative of the coverage observed also in the other vertical ranges.




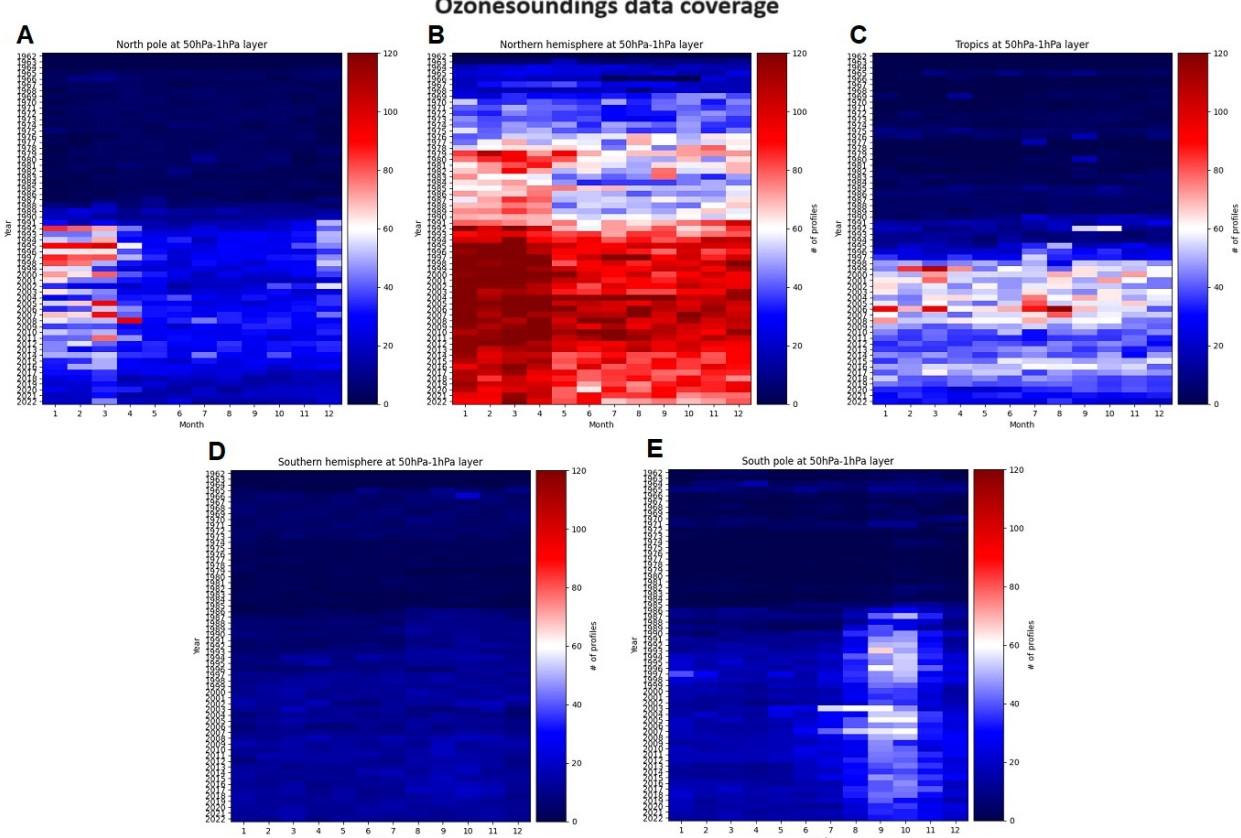

**Figure 3. Number of profiles available per month, since 1962, for the vertical range 50-1 hPa for the Northern Hemisphere polar latitudes (panel A), Northern Hemisphere mid-latitudes (panel B), Tropics (Panel C), Southern Hemisphere mid-latitudes (panel D), and Southern Hemisphere polar latitudes (panel E). The other vertical ranges taken into account (100-50 hPa, 200-100 hPa, 300-200 hPa) show a temporal coverage comparable to that shown for this range.**

Figure 3 indicates that most profiles are in the NH and TR, where also most of the stations are. Furthermore, there is a considerable difference in the amount of data available for each latitudinal sector before and after 1990; this is due to both the low number of ascents available before 1990 and the shorter vertical range covered. The latter rapidly expands after 1990. Figure 4 shows two profile sets from the station of Natal, Brazil (5.87°S, 35.2°W, 32m a.s.l., WMO code: 0-20008-0-NAT), measured before and after 1990.




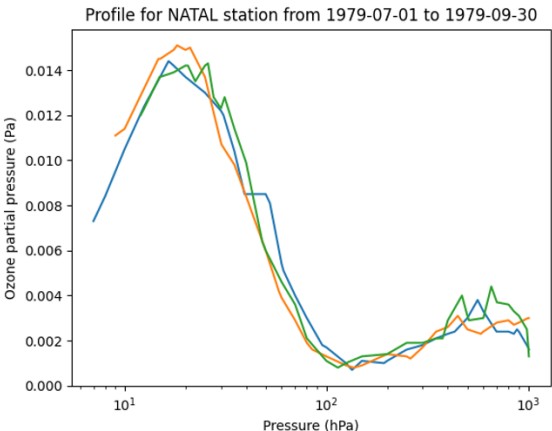
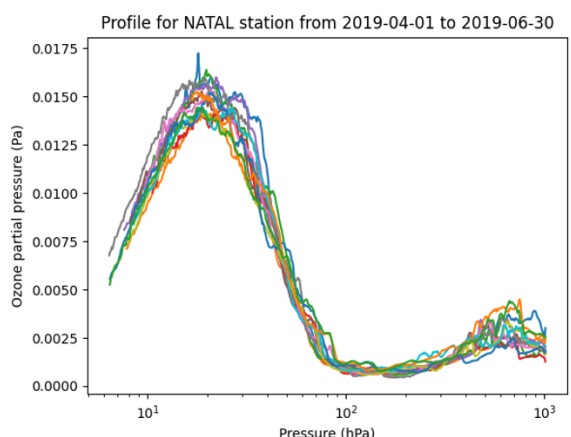

**Figure 4. Comparison of ozone profiles measured at the station of Natal, Brazil (5.87°S, 35.2°W, 32m a.s.l., WMO code: 0-20008-0-NAT) in summer 1979 (left panel) and spring 2019 (right panel). The figure first reveals a disparity between the two periods analysed in the number of profiles provided by the station (3 for summer 1979 compared to 13 for spring 2019) and also a clear difference in the vertical resolution of the spring 2019 profiles which appear to be much denser than the summer 1979 profiles.**

The number of profiles available in the summer of 1979 (3) is significantly lower than in the spring of 2019 (13). This discrepancy underscores an evident increase in station measurement activities in recent years compared to the early years. Additionally, despite their similar shape, the summer 1979 profiles exhibit a lower  vertical resolution compared to the spring 2019 profiles. This difference is primarily attributed to the technological evolution in radiosondes' hardware, enabling a denser vertical sampling of atmospheric ozone. Figure 5 juxtaposes two profiles, one from August 10, 1979, and the other from June 10, 2019, with the latter exhibiting more points on the vertical range compared to the former.

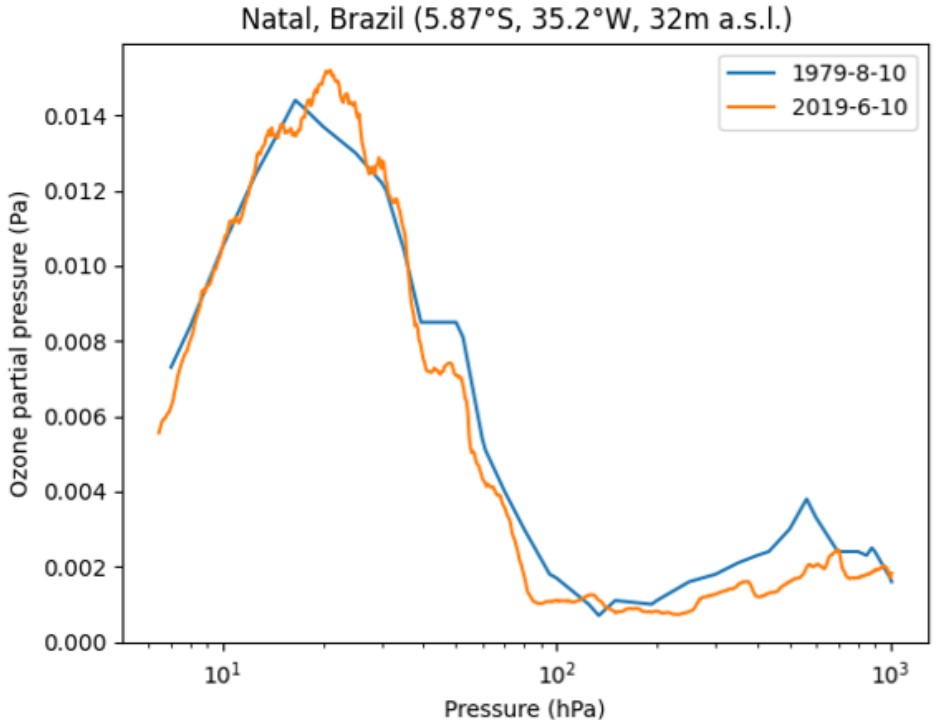

**Figure 5. Comparison of ozone profiles measured at the station of Natal, Brazil (5.87°S, 35.2°W, 32m a.s.l., WMO code: 0-20008-0-NAT) on August 10, 1979 (blue profile) and June 10, 2019 (orange profile). It reveals that the difference in the vertical resolution of the ozonesonde profiles is mainly due to the difference in the number of points for a single profile.**

## 2.3 Station classification

Data coverage plays a crucial role in accurately estimating anomalies and trends while reducing uncertainties. To address this, only the stations with at least one ozonesounding profile available each month were selected in the unified dataset, and these stations were categorized based on their time coverage.

Consequently, the 155 selected stations were classified into three categories:

- Long coverage (LC): 26 stations with data periods of at least 20 years;
- Medium coverage (MC): 23 stations with data time series ranging from 10 to 20 years;
- Short coverage (SC): 106 stations with data time series shorter than 10 years.

In Appendix A, Table A1 presents comprehensive information regarding the stations and their respective coverage classifications.



Only the first two clusters provide sufficient time coverage for estimating anomalies and trends on a decadal basis. Additionally, a fourth cluster, termed the Long and Medium coverage cluster (LMC), was derived by combining LC and MC to compare the relative impact of data completeness versus spatial coverage.

The disparity with the SC cluster is illustrated in Figure 6, where the coverage of the station South Pole (90°S, 0°E, 2835m a.s.l., WMO code: 0-20008-0-SPO), categorized as LC, is compared to Macquarie Island (54.5°S, 158.95°E, 6m a.s.l., WMO code: 0-20000-0-94998), designated as MC, and Praha (50.02°N, 14.45°E, 302m a.s.l., WMO code: 0-20000-0-11520), marked as SC.

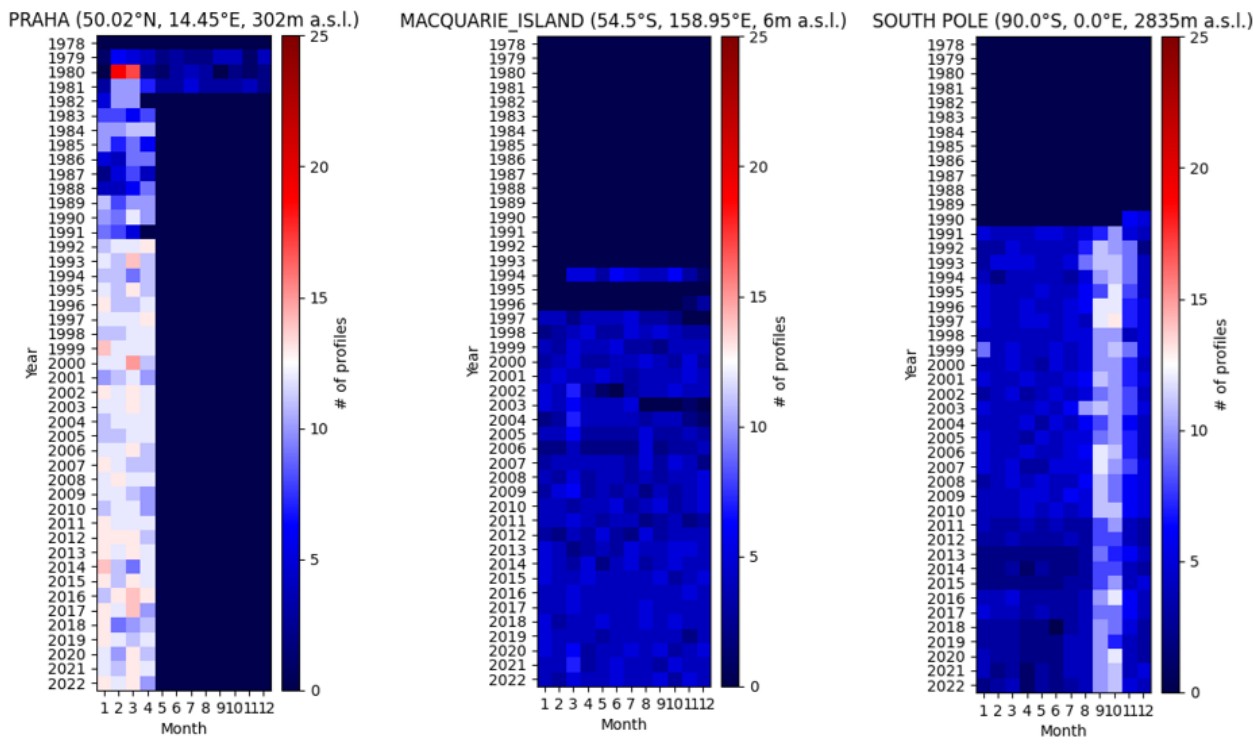


**Figure 6. Data coverage of three stations: Praha (50.02°N, 14.45°E, 302m a.s.l., WMO code: 0-20000-0-11520), classified as SC (left panel), Macquarie Island (54.5°S, 158.95°E, 6m a.s.l., WMO code: 0-20000-0-94998), labelled as MC (centre panel), and South Pole (90°S, 0°E, 2835m a.s.l., WMO code: 0-20008-0-SPO), categorized as LC (right panel). Macquarie Island's data coverage appears to align with the LC cluster, but a three-month gap in 2003 causes this station to be classified as MC. Praha station shows profiles**
**available only during the first four months of the year, resulting in an SC classification due to the missing ozone seasonal cycle. The South Pole station provides continuous data over 20 years, earning its LC classification.**

Figure 7 depicts the 49 stations (26 LC and 23 MC) whose data are analysed in this paper. It must be pointed out that there are only 4 LC stations in the TR and this is because the activation of most of the stations in this sector dates back to the

period 1998-2000 and this currently makes it difficult to have a continuous data record of at least 20 years necessary for LC classification.





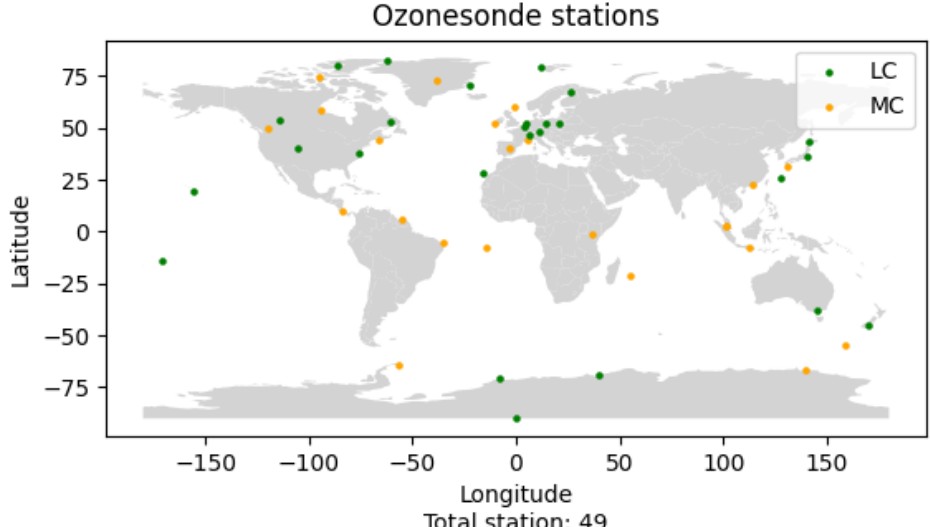

**Figure 7. Locations of the 49 ozonesonde stations are included in the unified database for analysis. 26 stations (depicted as green dots) belong to the LC cluster, and 23 stations (represented by orange dots) are part of the MC cluster.**

Table 2 reports, for each latitudinal sector, the number of stations per cluster.

**Table 2. Number of stations available in the unified data archive for each cluster and latitudinal sector.**

|  | # LC stations | # MC stations | # LMC stations |
|---|---|---|---|
| **NP** | 5 | 3 | 8 |
| **NH** | 12 | 7 | 19 |
| **TR** | 4 | 10 | 14 |
| **SH** | 2 | 1 | 3 |
| **SP** | 3 | 2 | 5 |

Due to the limited availability of LC and MC stations in the Southern Polar (SP) and Southern Hemisphere (SH), the analysis presented hereinafter is not conducted on these two regions. Although the related estimated trends are significant, the scarcity of stations' number and data availability in these sectors substantially enhances the uncertainties on the calculated trends, making them unreliable.





### 2.4 Quality check

Once stations are classified and duplicates identified, the following **quality checks** are applied to profiles of the ozone partial pressure, $p_{O_3}$:

- Plausibility check, to ensure reported values are within physically plausible ranges and to harmonise the measurement units (in Pa) to the International System of Units (SI). In Table 3, the outcome of these checks for the main variables of the three datasets is reported;

- Outliers check, to verify if there is a suspicious number of extreme values for $\rho_{O_3}$. Candidate outliers are identified using the Inter-Quartile Range method (Rocke, 1989; Vinutha et al., 2018) for the identification of strong outliers as follows:

$$median(p_{O_3}) - 3 \cdot IQR(p_{O_3}) \leq p_{O_3} \leq median(p_{O_3}) + 3 \cdot IQR(p_{O_3})$$

  If a high fraction of outliers is found, these are thoroughly investigated and, if verified as outliers, removed from the dataset;

- Completeness checks, to ensure time series have at least 5 ascents per month;

- Vertical coverage checks, performed on a monthly basis, to verify if ozone profiles reach 10 hPa. This check identified 59711 profiles terminating at less than 10 hPa (75.69%), 14598 (18.51%) terminating between 20 hPa and 10 hPa, and 4577 (5.8%) between 50 hPa and 20 hPa;

**Table 3. The outcome of the plausibility check applied to the main variables of SHADOZ, NDACC and WOUDC; the physically plausible range, and the percentage of flagged values for each dataset are reported. For the total uncertainty, the percentage refers to its unavailability in each dataset.**

| Variable | Plausible physical range | Flagged values SHADOZ | Flagged values NDACC | Flagged values WOUDC |
|---|---|---|---|---|
| Air pressure | 0 Pa ≤ x ≤110000 Pa | 0.008% | 0.04% | 0.01% |
| Air temperature | 150K ≤ x ≤ 330K | 0.21% | 2.78% | 0.46% |
| Relative humidity | 0% ≤ x ≤ 120% | 9.23% | 17.48% | 13.08% |
| Ozone partial pressure | 0 Pa ≤ x ≤ 0.25 Pa | 2.31% | 7.32% | 0.85% |
| Ozone concentration | 0 ppmv ≤ x ≤ 30 ppmv | 2.31% | 86.22% | N.A. |
| Ozone partial pressure total uncertainty | N.D. | 19.75% | 68.05% | N.A. |

- Vertical completeness checks for ensuring a minimum number of records are available for each vertical region covered by the ozonesoundings. This is quantified in at least one point every 50 m. In this work, 78885 profiles were checked and only 18281 (23.2%) did not pass the check in certain atmospheric regions.





Finally, null values are also investigated to check the coherency of the selected profiles with the source datasets. Table 4 shows the percentage of null values present in the unified database, justifying the reason.

235    **Table 4. List of the main variables within the unified database on which the percentage of null values is calculated. For each variable, the reason that caused the null values is also indicated.**

| Variable | % null values | Reason |
|---|---|---|
| Air temperature | 1.68% | Values flagged by plausibility checks. |
| Relative humidity | 14.30% | Values flagged by plausibility checks. |
| Ozone partial pressure | 4.61% | Values flagged by plausibility checks. |
| Ozone concentration | 78.5% | Only available from the SHADOZ and NDACC networks, but not always present in the source data. |
| Ozone partial pressure total uncertainty | 75.49% | Only available from the SHADOZ and NDACC networks, not always present in the SHADOZ data, sometimes present in NDACC data. |

The quality checks mentioned above are employed collectively to generate a single quality flag, indicating the percentage of successful data checks for each ozone profile. This approach simplifies the process for users, offering an accessible means to
240    assess the data quality comprehensively, indicating whether it meets the relevant criteria completely, partially, or not at all. This quality flag serves as a key component of the unified dataset, providing users with a straightforward tool to gauge data reliability.

Figure 8 illustrates examples of profiles that do not meet the established quality criteria, highlighting instances where data may be deemed less reliable or require further scrutiny. By visualizing profiles that fail the quality checks, users can readily
245    identify areas where data quality may be compromised and take appropriate steps for validation or data filtering.



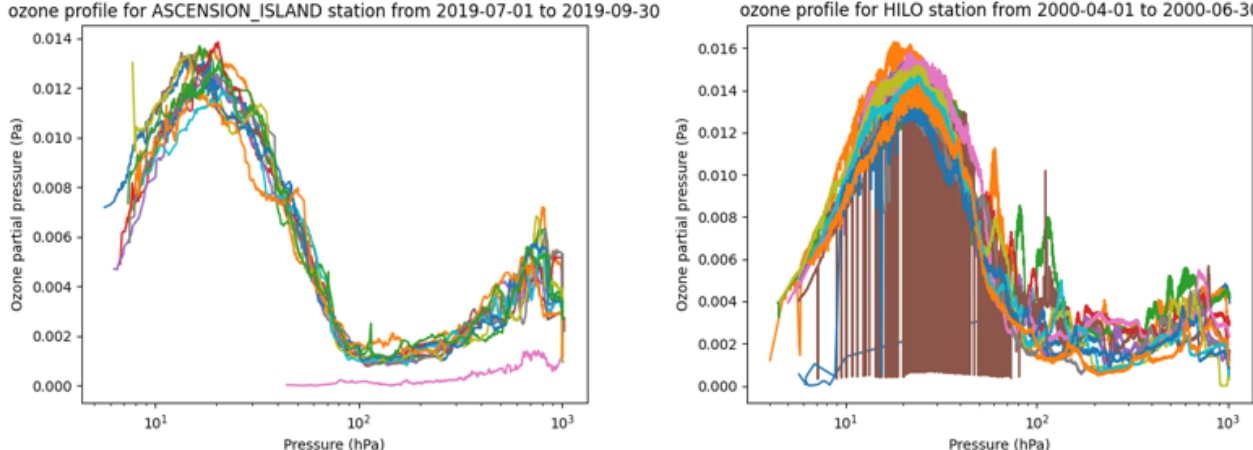

**Figure 8. Examples of ozonesonde profiles failing the quality criteria for outlier detection. The left panel displays erroneous profile values (magenta profile below 400 hPa) in comparison to the seasonal average profile. In the right panel, a profile exhibiting high noise levels is observed, likely stemming from measurement errors introduced by the station operator. This profile alternates between plausible and implausible values, with the latter being an order of magnitude smaller.**

## 2.5 Representativeness of the stations

Before examining trends in the Upper Troposphere/Lower Stratosphere (UT/LS), the representativeness of the network was initially assessed to ensure an accurate capture of ozone variability across different latitudes. This assessment utilized an approach developed by Weatherhead et al. (2017), which evaluates a measurement site's ability to replicate monthly variability in the seasonally adjusted Total Ozone Column (TOC) at nearby locations. Spatial representativeness was quantified using correlations, as depicted in Figure 9.

The study utilized Total Ozone Mapping Spectrometer-Earth Probe (TOMS-EP) ozone satellite data from July 1996 to December 2005 to estimate correlations of ozone observations for stations available in the unified database.





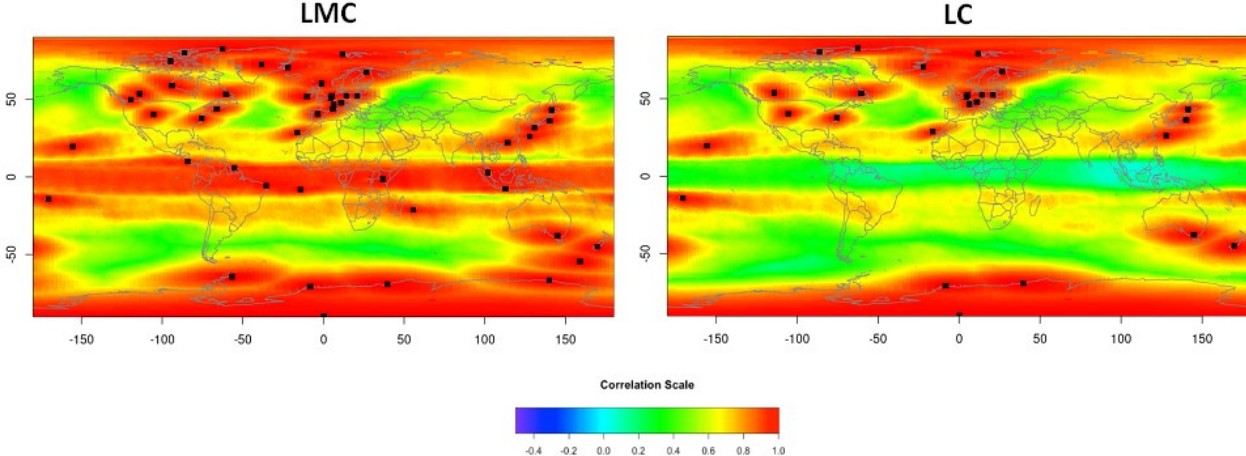

**Figure 9. Assessment of representativeness for unified stations, with the left panel depicting the LMC cluster and the right panel showing the LC cluster, utilizing EP TOMS ozone satellite data from July 1996 to December 2005. The map illustrates the correlation level between each station and its surrounding area. A correlation value of 1 indicates a complete correlation with the surrounding grid points, while 0 signifies no correlation with neighbouring points.**

Figure 9 indicates a strong correlation within the LC cluster for stations located in the NP, NH, and SP sectors compared to the overall set of stations in the unified database, albeit with a modest increase in the area exhibiting correlation values exceeding 0.8. Consequently, the additional stations within the LMC cluster may be deemed redundant for these regions, with preference given to utilizing data from the LC station due to its higher-quality correlated time series.

A notable disparity is observed near the equatorial regions, where the lack of long data records impedes the selection of suitable stations within the LC cluster, resulting in some locations exhibiting near-zero correlation. Conversely, within the remnants of the tropical belt, differences are comparatively minor. Consequently, in tropical regions, greater confidence may be placed in trends estimated using data from the LMC cluster rather than the LC cluster.

In the Southern Hemisphere, correlations do not exhibit significant differences, except notably in Oceania.

## 3 Methodology

### 3.1 Regression methods

Structural uncertainty pertains to whether a model's mathematical structure accurately represents its intended purpose (Baldissera Pacchetti, 2021). In this study, various linear regression approaches for trend estimation are examined to evaluate structural uncertainty in trend estimation. The first approach used in this investigation is least-square linear regression, using the model in the form (Reinsel et al. 2002):

$$y_t = \alpha + \beta x_t + \mu_t, \qquad t = 1, \dots, T$$



where $y_t$ is the monthly anomaly time series, $t$ is the time variable assigned to $y_t$, $\alpha$ a constant term, $x_t$ is the linear trend function, $\beta$ is the linear trend and $\mu_t$ is the residual term that is assumed to be autoregressive of the order of 1 [AR(1)].

The second method used is the Least Absolute Deviation (LAD) regression. It is a resistant and non-parametric regression method fitting the paired data to the linear model using a robust and resistant LAD method (Rice and White, 1964; Barrodale, 1968; Wong and Schneider Jr, 1989; Calitz and Rüther, 1996; Santer et al., 2000). This technique derives from an algorithm by Barrodale and Roberts (1974). This algorithm is resistant to outliers.

The third method used involves the computation of the slopes between every possible pair of points in the time series, taking
the median value as the trend estimate. It is called the Theil-Sen estimator and can be significantly more accurate than simple linear regression for skewed and heteroskedastic data and shows performances similar to non-robust least squares even for normally distributed data in terms of statistical power (Theil, 1950; Siegel and Benson, 1982; Helsel and Hirsch, 1992). This method is very similar to classic least squares (Lanzante, 1996).

In addition, the Mann-Kendal (MK) test (Mann, 1945; Kendall, 1975) is used to statistically evaluate whether there is a
monotonic upward or downward trend of the variable of interest over time. A monotonic upward (or downward) trend means that the variable increases (or decreases) consistently over time, but the trend may or may not be linear. The MK test can be used instead of a parametric linear regression analysis, which can be used to test whether the slope of the estimated linear regression line is different from zero. Regression analysis requires that the residuals of the regression line be normally distributed; an assumption not required by the MK test, which is a nonparametric (distribution-free) test.

**3.2 Trends calculation**

Decadal ozone trends are estimated using the linear regression methods, described in section 3.1, for the LC and the LMC clusters.

Data from the unified dataset have been aggregated by latitude into five sectors: NP (60°N-90°N), NH (30°N-60°N), TR (30°S-30°N), SH (60°S-30°S), and SP (90°S-60°S). Additionally, aggregation is done by vertical range (300-200 hPa, 200-
100 hPa, 100-50 hPa, and 50-1 hPa). Average monthly anomalies are calculated on aggregated data by sector and vertical range as follows:

$$An_{mm_x(YYYY)} = mm_x(YYYY) - \overline{mm}_{x_{1978-2022}}$$

For the percentage anomalies, we have:

$$An_{\%mm_x(YYYY)} = \frac{mm_x(YYYY) - \overline{mm}_{x_{1978-2022}}}{\overline{mm}_{x_{1978-2022}}} \cdot 100$$

Where $mm_x(YYYY)$ is the average of the ozone values of month $x$ of the year $YYYY$, $\overline{mm}_{x_{1978-2022}}$ is the average of the ozone values for month $x$ calculated from the average for that month for all years of the series (from 1978 to 2022). The average anomaly for month x in year YYYY is obtained by calculating the difference between these two values. A positive or negative number for the average monthly anomaly indicates a higher or lower amount of ozone, respectively, compared to





the average for the selected month. Trends are then assessed by applying linear regressors, described in section 3.1, to the
series of average monthly anomalies categorized by latitudinal sector and vertical range.

### 3.3 Time series homogenization

The time series of the unified dataset have been tested using the Standard Normal Homogeneity Test (SNHT) to assess how much the merging of the three ozonesounding datasets may increase the number of shifts or changes in the mean of a time series. It has been largely applied in identifying discontinuities in near-surface temperature or precipitation records (Venema
et al., 2012) as well as in upper air data series (Haimberger, 2007). The SNHT test statistic measures the degree of deviation of the sub-series mean from the overall mean of the entire time series. The interpretation of the test statistic for the SNHT depends on several factors, including the specific context of the data, the length of the time series, and the desired level of significance. In general, for the SNHT, larger absolute values of the test statistic indicate stronger evidence of a shift or change in the mean of the time series. There is no universally agreed value for the test statistic itself. Large values of the test
statistic suggest the presence of a change point. Haimberger suggested values larger than 100 should be considered change points for the investigation of temperature upper-air data. The time series provided at 100 hPa by NDACC stations belonging to the LC cluster have been investigated in comparison with the corresponding time series of the unified datasets, obtained for the same stations. In Table 5, the mean and maximum values of the SNHT test statistic are reported. Similar results are obtained also at levels below 100 hPa pressure (not shown).
Mean values are generally the same or only slightly altered by the merging of the NDACC data with other data sources. Maximum values, instead, although often not altered, for the station in Macquarie islands and Payerne are increased. It must be noted that the maximum values of the statistics were already exceeding the threshold set by Haimberger, indicating the presence of potential change points. However, through the inspection of the values of the test statistic, peak values have the same or close positions in the time series and their increase is mainly due to the increase in the number of measurements
available in one of the two sub-series in which the original is divided (before and after the potential changepoint) by the SNHT test. The homogenization of the time series is therefore not critically affected by the merging operation of the three datasets, and this is expected by the fact that measurements at each site are the same reported through different data archives and using different formats and variables.





**Mean and maximum values with suffix 1 are related to the original NDACC time series, while suffix 2 refers to the values obtained for the corresponding time series obtained in the unified dataset. Only time series with periods of missing data smaller than 1 month have been tested and reported in the table.**

| Station | Mean 1 | Mean 2 | Max. 1 | Max. 2 |
|---|---|---|---|---|
| LAUDER | 13.74 | 13.64 | 120.59 | 120.59 |
| IZANA | 3.15 | 3.15 | 28.76 | 28.76 |
| MACQUARIE_ISLAND | 10.37 | 7.78 | 72.08 | 133.00 |
| ALERT | 14.01 | 14.35 | 196.86 | 196.86 |
| BROADMEADOWS | 8.26 | 7.11 | 38.08 | 43.97 |
| EUREKA | 13.09 | 12.58 | 126.47 | 126.47 |
| PAYERNE | 4.18 | 4.85 | 59.82 | 264.12 |
| NY_ALESUND | 3.14 | 3.31 | 24.98 | 24.98 |
| OHP | 6.21 | 6.21 | 74.40 | 74.40 |
| SODANKYLA | 8.36 | 8.17 | 250.21 | 250.21 |

## 4 Result

The trend estimations calculated for the LC and LMC clusters are compared to assess the impact of sampling error. Table 6 presents the number of ascents stored in the database for each cluster across all latitude regions, along with the corresponding data percentages.

**Table 6. Number of profiles at all latitude ranges in the unified dataset for long coverage (left) and long and medium coverage**
**(middle) clusters. The percentage of profiles added to the long coverage cluster for creating the long and medium coverage cluster are also reported (right).**

| | Long coverage | Long and medium coverage | % of additions |
|---|---|---|---|
| **NP** | 9404 | 12508 | 33.01 |
| **NH** | 34702 | 41592 | 19.85 |
| **TR** | 5394 | 12573 | 133.09 |
| **SH** | 3171 | 4327 | 36.45 |
| **SP** | 6007 | 7886 | 31.28 |

Except for the TR area, which exceeds 100%, the right column of Table 6 demonstrates that additions to other latitudinal zones are not above 37% of the total. This is because, as Table 2 shows, the LMC cluster has more stations than the LC
cluster, which has four stations. The MK test was used to estimate the trend in ozone concentration and assess the





significance of the expected trend. The MK test results for two different periods—1978–1999 and 2000–2022—are shown in Tables 7 and 8, respectively. The purpose of the study that follows is to measure how geographical sampling affects trend estimates and, consequently, our understanding of the variability of ozone across time.

**Table 7. MK test results for the period 1978-1999 for long coverage (upper row) and long and medium coverage (lower row) clusters, for all latitudinal regions at all vertical ranges. The monotonic upward trends are indicated with "I" (increasing), the monotonic downward trends with "D" (decreasing), and the non-significant trends with "NT" (no trend).**

| 1978-1999 | | SP | SH | TR | NH | NP |
|---|---|---|---|---|---|---|
| LC | 50-1hPa | D | D | I | D | D |
| | 100-50hPa | D | NT | I | D | NT |
| | 200-100hPa | D | NT | I | D | NT |
| | 300-200hPa | NT | NT | I | D | I |
| LMC | 50-1hPa | D | D | I | D | D |
| | 100-50hPa | D | NT | I | D | D |
| | 200-100hPa | D | NT | I | D | D |
| | 300-200hPa | NT | NT | NT | D | D |

**Table 8. Same as Table 7 but for the period 2000-2022.**

| 2000-2022 | | SP | SH | TR | NH | NP |
|---|---|---|---|---|---|---|
| LC | 50-1hPa | I | D | I | D | D |
| | 100-50hPa | I | D | NT | I | NT |
| | 200-100hPa | NT | D | NT | I | NT |
| | 300-200hPa | NT | NT | NT | I | NT |
| LMC | 50-1hPa | I | D | I | NT | D |
| | 100-50hPa | I | D | I | I | NT |
| | 200-100hPa | NT | NT | I | I | NT |
| | 300-200hPa | NT | NT | NT | I | NT |


The significance of trends in the LMC cluster is influenced by whether the LC or MC clusters predominate, determined by the number of observations and stations available for each. Sometimes, the LMC falls between the LC and MC results, suggesting a potential improvement in the LC.

For the period 1978-1999, differences emerge among the two clusters across latitude ranges. In the SP, the LC and LMC
clusters align, showing consistent trends except for the 300-200 hPa layer. Conversely, few significant trends are found in the SH, highlighting the scarcity of observations and stations. In the TR, results resemble those of the SP. In the NH, both



clusters exhibit consistent trends. However, at the NP, the LC lacks significant trends in certain pressure ranges due to the limited number of stations, necessitating the use of the LMC cluster.

As the period 2000-2022, at the SP, the LC and LMC clusters provide significant trends only at the two higher vertical layers. In the SH, all clusters perform similarly, with the LC showing slightly better performance via the MK test. Notably, at the TR, all clusters exhibit agreement, with the LC yielding non-significant trends in most pressure ranges and the LMC offering an intermediary solution. In the NP, trends in the LMC, despite being influenced by incorporated MC stations, do not differ from the LC. In the NH, the LC cluster demonstrates significance across pressure ranges, while the LMC cluster shows a non-significant trend at the 50-1 hPa vertical layer. In the NP, both clusters provide significant trends only in the 50-

1 hPa vertical layer.

Overall, the LC and LMC clusters show comparable abilities to estimate significant changes in ozone concentration, except for the NP in 1978-1999 and the TR in 2000-2022. In the latter period, LMC trends are significant across most pressure ranges and align with LC cluster trends in the NH. In other cases, the LC cluster, with fewer but longer time series, provides significant trends without the inclusion of shorter time series from MC stations.

Tables 9-12 display ozone partial pressure trends estimated using linear methods for LC and LMC clusters, reported as percentages per decade for each vertical range and latitudinal sector, covering the 1978-1999 and 2000-2022 periods. This aims to assess the impact of structural uncertainty arising from the regression algorithm used to estimate trends, as well as the sampling error.

**Table 9. Trend estimates as percentage per decade (% dec⁻¹) obtained with the Linear (LIN), Least Absolute Deviation (LAD) and Theil-Sen (TS) regressors for all latitudinal sectors at the 50-1hPa vertical range for each cluster. Legend: Positive value (increasing trend); Negative value (decreasing trend); NT (No Trend).**

| 50-1hPa | | % dec⁻¹ | SP | SH | TR | NH | NP |
|---|---|---|---|---|---|---|---|
| **1978-1999** | **LC** | **LIN** | - | -10.14 | 7.20 | -5.94 | -9.58 |
| | | **LAD** | - | -8.34 | 8.06 | -5.52 | -9.59 |
| | | **TS** | - | -8.87 | 7.49 | -6.65 | -7.00 |
| | **LMC** | **LIN** | - | -10.09 | 3.75 | -5.63 | -14.27 |
| | | **LAD** | - | -8.26 | 5.39 | -4.94 | -11.97 |
| | | **TS** | - | -8.13 | 5.48 | -5.09 | -11.42 |
| **2000-2022** | **LC** | **LIN** | 6.29 | -2.40 | 2.09 | -1.33 | -5.83 |
| | | **LAD** | 6.53 | -2.79 | 1.80 | -1.09 | -5.97 |
| | | **TS** | 6.94 | -2.06 | 2.02 | -0.71 | -5.55 |
| | **LMC** | **LIN** | 6.02 | -2.0 | 0.95 | NT | -5.70 |
| | | **LAD** | 7.24 | -1.97 | 0.62 | NT | -5.71 |
| | | **TS** | 7.82 | -2.03 | 0.99 | NT | -3.05 |



At the 50-1 hPa vertical range (Table 9) for the SP during the period 1978-1999, as depicted in Figure 3, the significant

difference in data coverage before and after 1996 notably amplifies overall uncertainty in estimated trends. Consequently, these data are not used for the comparison among linear regression approaches. This same scenario applies to the 100-50 hPa vertical range (Table 10). In the SH, trends estimated for the clusters in both periods exhibit consistent agreement, with differences among clusters never surpassing 1%. Conversely, in the TR, estimates display discrepancies of up to about 3.5%. This discrepancy likely arises from the greater number of stations available for the LMC cluster, enhancing coverage and

representativeness compared to the LC cluster, as discussed in Chapter 2.5.

For the NH, the clusters agree for the 1978-1999 period (with a maximum discrepancy of about 1.5%), but in contrast for 2000-2022, where the LMC cluster's trend fails to achieve significance in the MK test.

Finally, in the NP, trends correspond only in the 2000-2022 period (with a maximum discrepancy of 2.5% for the TS regressor). For 1978-1999, differences among clusters nearly reach 5%.


**Table 10. Same as Table 9 but for the vertical range 100-50 hPa.**

| 100-50 hPa | | % dec⁻¹ | SP | SH | TR | NH | NP |
|---|---|---|---|---|---|---|---|
| **1978-1999** | **LC** | LIN | - | NT | 8.91 | -8.33 | NT |
| | | LAD | - | NT | 10.29 | -7.58 | NT |
| | | TS | - | NT | 3.29 | -8.23 | NT |
| | **LMC** | LIN | - | NT | 7.6 | -10.23 | -12.5 |
| | | LAD | - | NT | 7.12 | -9.74 | -11.55 |
| | | TS | - | NT | 5.96 | -9.09 | -12.56 |
| **2000-2022** | **LC** | LIN | 4.83 | -8.53 | NT | 3.40 | NT |
| | | LAD | 2.52 | -9.37 | NT | 4.07 | NT |
| | | TS | 2.78 | -8.80 | NT | 4.45 | NT |
| | **LMC** | LIN | 2.44 | -5.49 | 4.13 | 2.5 | NT |
| | | LAD | 1.55 | -5.55 | 3.58 | 3.47 | NT |
| | | TS | 3.01 | -6.92 | 4.73 | 3.2 | NT |

In Table 10, focusing on the 100-50 hPa range, it is worth noting that the data for the SH are not utilized for comparison. Regarding trends estimated for the 1978-1999 period, they are found to be non-significant for the MK test in SH. However,

for the period 2000-2022, there is an observed discrepancy up to nearly 4% between the LC and LMC clusters. The NH sector stands out with trend estimates in the period 2000-2022 showing less than about 1% sampling error and significant trend estimates, attributed to higher data density and greater variability in this vertical range. Nonetheless, even in NH, the LC and LMC clusters exhibit discrepancies in the period 1978-1999 of around 1% to 2% in estimated trends. Conversely, the NP sector lacks valid trends, except for the LMC cluster during 1978-1999, with especially noticeable trends. Additionally,

the TR sector only presents a valid trend for the LC cluster during the 1978-1999 period but with a considerable discrepancy



among regressors (structural uncertainty) of 7%. Interestingly, the trends estimated by the LMC cluster are significant for both periods and show lower discrepancies among regressors compared to LC, with structural uncertainties of 1.6% for 1978-1999 and 1.2% for 2000-2022. This performance improvement can be attributed to the greater availability of stations in the LMC cluster, featuring 10 more stations than LC (refer to Table 2). For conducting trend analysis with data from the unified database, high-quality measurements akin to those from the LC cluster are essential. However, in sectors with few LC stations, such as in the TR, data from the LMC cluster can ensure an enhanced representativeness than the LC cluster.

**Table 11. Same as Table 9but for the vertical range 200-100 hPa.**

| 200-100 hPa | | % dec⁻¹ | SP | SH | TR | NH | NP |
|---|---|---|---|---|---|---|---|
| **1978-1999** | **LC** | **LIN** | - | NT | - | -9.88 | NT |
| | | **LAD** | - | NT | - | -10.05 | NT |
| | | **TS** | - | NT | - | -9.42 | NT |
| | **LMC** | **LIN** | - | NT | 9.83 | -12.87 | - |
| | | **LAD** | - | NT | 7.48 | -13.72 | - |
| | | **TS** | - | NT | 2.58 | -11.54 | - |
| **2000-2022** | **LC** | **LIN** | NT | -6.89 | NT | 7.98 | NT |
| | | **LAD** | NT | -4.92 | NT | 7.52 | NT |
| | | **TS** | NT | -5.81 | NT | 10.37 | NT |
| | **LMC** | **LIN** | NT | NT | 6.76 | 5.86 | NT |
| | | **LAD** | NT | NT | 3.97 | 4.89 | NT |
| | | **TS** | NT | NT | 4.82 | 5.21 | NT |

In Table 11, both in the SP and NP sectors trend estimates are considered unreliable for the period 1978-1999 and non-significant for 2000-2022. Trend assessments for the LC cluster in the TR are also deemed unreliable due to the significantly smaller amount of data available before 1995 compared to subsequent years, vanishing the evaluation of decadal trends. Conversely, trend estimates for the LMC cluster in TR are significant, but exhibit considerable discrepancies among regressors, with structural uncertainty of almost 7.5% for 1978-1999 and 3% for 2000-2022. Notably, the performance of the LMC cluster in the TR sector outperforms that of the LC cluster in this layer. Moreover, the NH sector is the only one with all valid trends, although there is a disagreement between LC and LMC (up to about 3.5 % for 1978-1999 and 5% for 2000-2022). Furthermore, it is worth considering the discrepancy between the results of the TS regressor with those of the LIN and LAD regressors that occurred for LC in the period 2000-2022. The LIN and LAD regressors agree with each other, reporting a structural uncertainty of only 0.5%, on the contrary TS where this uncertainty rises to around 3%.





**Table 12. Same as Table 9 but for the vertical range 300-200hPa.**

| 300-200 hPa | | % dec⁻¹ | SP | SH | TR | NH | NP |
|---|---|---|---|---|---|---|---|
| **1978-1999** | **LC** | **LIN** | NT | NT | 12.05 | -8.75 | 5.20 |
| | | **LAD** | NT | NT | 14.45 | -8.18 | 11.21 |
| | | **TS** | NT | NT | 13.88 | -6.66 | 9.83 |
| | **LMC** | **LIN** | NT | NT | NT | -12.34 | - |
| | | **LAD** | NT | NT | NT | -11.24 | - |
| | | **TS** | NT | NT | NT | -8.42 | - |
| **2000-2022** | **LC** | **LIN** | NT | NT | NT | 7.29 | NT |
| | | **LAD** | NT | NT | NT | 8.10 | NT |
| | | **TS** | NT | NT | NT | 6.74 | NT |
| | **LMC** | **LIN** | NT | NT | NT | 5.16 | NT |
| | | **LAD** | NT | NT | NT | 6.48 | NT |
| | | **TS** | NT | NT | NT | 6.97 | NT |

The scenario for the 300-200 hPa layer (Table 12) is similar to the previous layer. Additionally, this vertical interval exhibits the fewest significant trends. The most suitable vertical ranges for trend analysis are 50-1 hPa and 100-50 hPa due to their
richer data content, providing a more robust dataset for trend estimation. Table 13 highlights the most significant disparities in the trends obtained through different regression techniques, facilitating the discussion of structural uncertainties.





**Table 13. Maximum difference in trends among different regressors estimated for all vertical ranges, latitudinal sectors, clusters and periods (NT means No Trends).**

| | | | Maximum slope difference | | | |
|---|---|---|---|---|---|---|
| | | % dec⁻¹ | 50-1hPa | 100-50hPa | 200-100hPa | 300-200hPa |
| **1978-1999** | **LC** | **SP** | - | - | - | NT |
| | | **SH** | 1.8 | NT | NT | NT |
| | | **TR** | 0.86 | 7.0 | - | 2.4 |
| | | **NH** | 1.13 | 0.75 | 0.63 | 2.09 |
| | | **NP** | 2.59 | NT | NT | 6.01 |
| | **LMC** | **SP** | - | - | - | NT |
| | | **SH** | 1.96 | NT | NT | NT |
| | | **TR** | 1.73 | 1.64 | 7.25 | NT |
| | | **NH** | 0.69 | 1.14 | 2.18 | 3.92 |
| | | **NP** | 2.85 | 1.01 | - | - |
| **2000-2022** | **LC** | **SP** | 0.65 | 2.31 | NT | NT |
| | | **SH** | 0.73 | 0.84 | 1.97 | NT |
| | | **TR** | 0.29 | NT | NT | NT |
| | | **NH** | 0.62 | 1.05 | 2.85 | 1.36 |
| | | **NP** | 0.42 | NT | NT | NT |
| | **LMC** | **SP** | 1.8 | 1.46 | NT | NT |
| | | **SH** | 0.06 | 1.43 | NT | NT |
| | | **TR** | 0.37 | 1.15 | 2.79 | NT |
| | | **NH** | NT | 0.97 | 0.97 | 1.81 |
| | | **NP** | 2.66 | NT | NT | NT |


In the SH, trends, for both periods, are estimated using only two stations for the LC cluster and three stations for the LMC cluster. Therefore, although the structural uncertainties are small, caution is advised in utilizing these estimates. Conversely, for other latitudinal sectors, the uncertainties are minimal except for 1978-1999 in the TR at 100-50 hPa (LC cluster) and 200-100 hPa (LMC cluster) and in the NP at 300-200 hPa (LC cluster), and the amount of data and number of stations are

adequate to characterize the vertical variability of ozone concentration.

The trend estimations provided in Tables 9-12 are influenced by the varying amounts of data available for each latitudinal area. The NH sector has the largest quantity of profiles, followed by the TR. In the 50-1 hPa and 100-50 hPa vertical ranges, the patterns of these two latitudinal sectors have been compared to the literature and the results are presented below.

**4.1 Comparison of Northern Hemisphere mid-latitude trends**

Figures 10-11 illustrate the computed trends for the NH sector within the 50-1 hPa and 100-50 hPa vertical ranges. For this latitudinal sector, the LC cluster was selected for the following trend analysis.





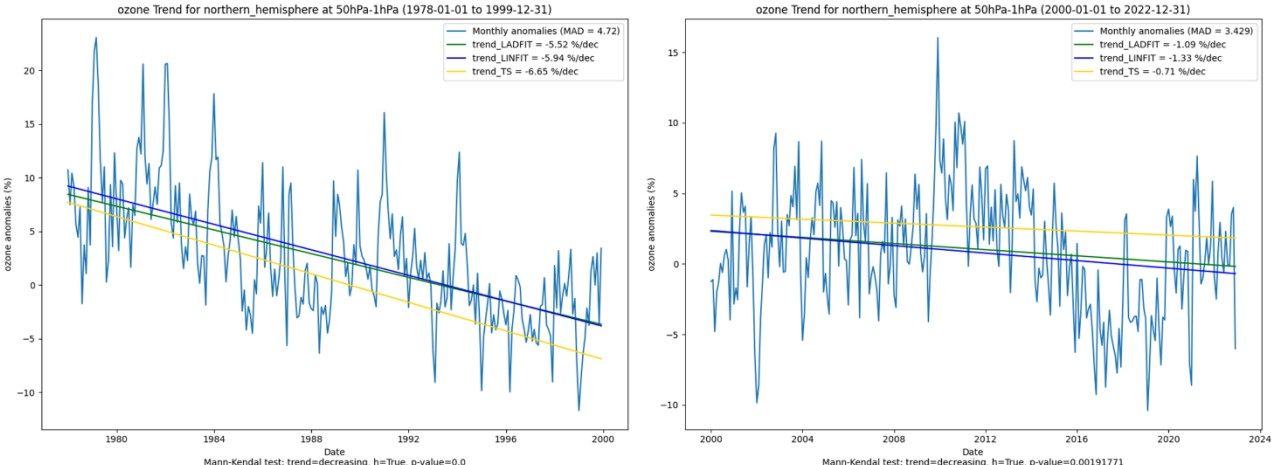

**Figure 10. Estimated trends for the NH latitudinal sector in the 50-1 hPa vertical range using LC cluster data. On the left is the 1978-1999 trend, while on the right is the 2000-2022 trend. The figure legend shows the resulting percentage per decade (% dec$^{-1}$) for each regressor utilized and the mean absolute difference (MAD) calculated on the average anomalies. Finally, the MK test result is shown at the bottom centre.**


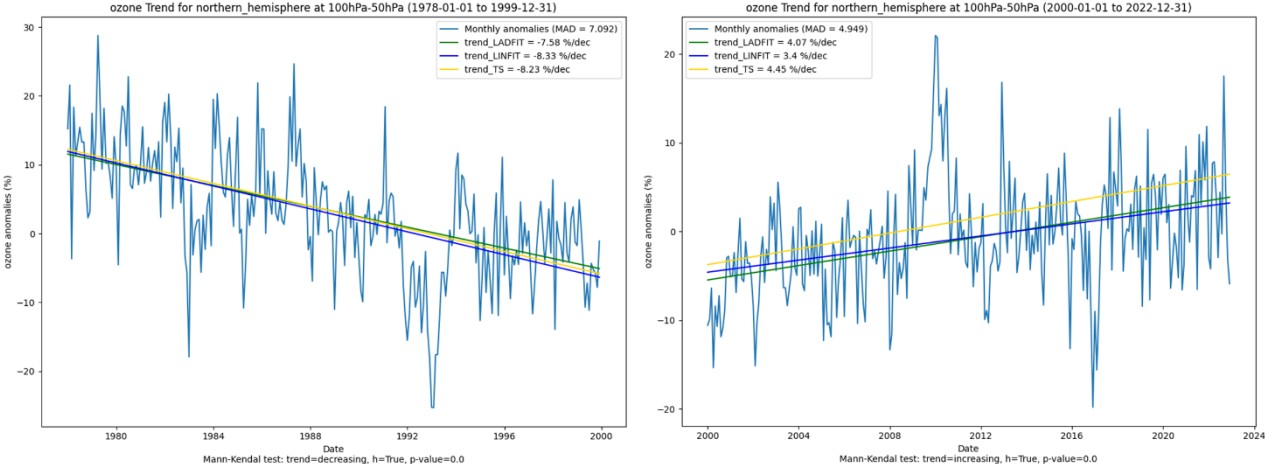

**Figure 11. Same as Figure 10 but for the vertical range100-50hPa.**


The trends depicted in Figures 10 and 11 can be cross-referenced with the ozone trends outlined in Petropavlovskikh et al. (2019) and Sofieva et al. (2021), where more sophisticated regression models and broader dataset, including both in situ soundings and remote sensing (ground-based and satellite) measurements, were used. In Petropavlovskikh et al. (2019), both satellite data (from SBUV Merged Ozone Data Set (SBUV MOD), SBUV Cohesive data set (SBUV COH), GlobalOZone

Chemistry And Related trace gas Data records for the Stratosphere (GOZCARDS), Stratospheric Water and OzOne Satellite Homogenized (SWOOSH),Global Ozone Monitoring by Occultation of Stars (SAGE-GOMOS), and Optical Spectrograph and InfraRed Imaging System(SAGE-OSIRIS)) and data from ground-based stations (from the NDACC, WOUDC and



SHADOZ ozonesoundings, LIDAR, Microwave, FTIR and Brewer/Dobson Umkehr) were used to estimate the trends; in Sofieva et al. (2021), instead, only satellite data from the MErged GRIdded Dataset of Ozone Profiles (MEGRIDOP) were

used. In addition, in Petropavlovskikh et al. (2019), LOTUS regression was employed to estimate trends, incorporating a lag-1 autocorrelation correction. Conversely, Sofieva et al. (2021) utilized multiple linear regression, with autocorrelation addressed using the Cochrane-Orcutt transformation (Cochrane and Orcutt, 1949).

Regarding the 1978-1999 period as presented in Petropavlovskikh et al. (2019), negative trends of 5% per decade at 50-1 hPa and 10% per decade at 100-50 hPa were observed in the NH lower stratosphere. These estimates align with the trends

identified in this study: a negative trend of 6% per decade at 50-1 hPa and 8% per decade at 100-50 hPa. For the period 2000-2022, the trend reported by Petropavlovskikh et al. (2019) at 50-1 hPa was negative 1% per decade, and similar to the results of the LC cluster. Similarly, at 100-50 hPa, the reported negative trend of 2% per decade, with an uncertainty range of ±7%, is consistent with the LC cluster value (4% per decade positive trend), falling within the above uncertainty range. Additionally, Sofieva et al. (2021) present a 1-2% negative trend for the NH lower stratosphere at 50-1 hPa, consistent with

the trends shown with the LC cluster for the period 2000-2022. Indeed, the result presented by Sofieva et al. (2021) is based on a 15-year dataset from 2003 to 2018, which is comparable with the period 2000-2022 of this study.

### 4.2 Comparison of Tropics trends

Figures 12-13 illustrate the trends computed for the TR sector within the 50-1 hPa and 100-50 hPa vertical ranges, respectively, for the periods 1978-1999 and 2000-2022. As previously mentioned, the LMC cluster likely outperforms the

LC cluster due to its greater spatial representativeness. Hence, the LMC cluster is selected for the trend analysis in the tropics.

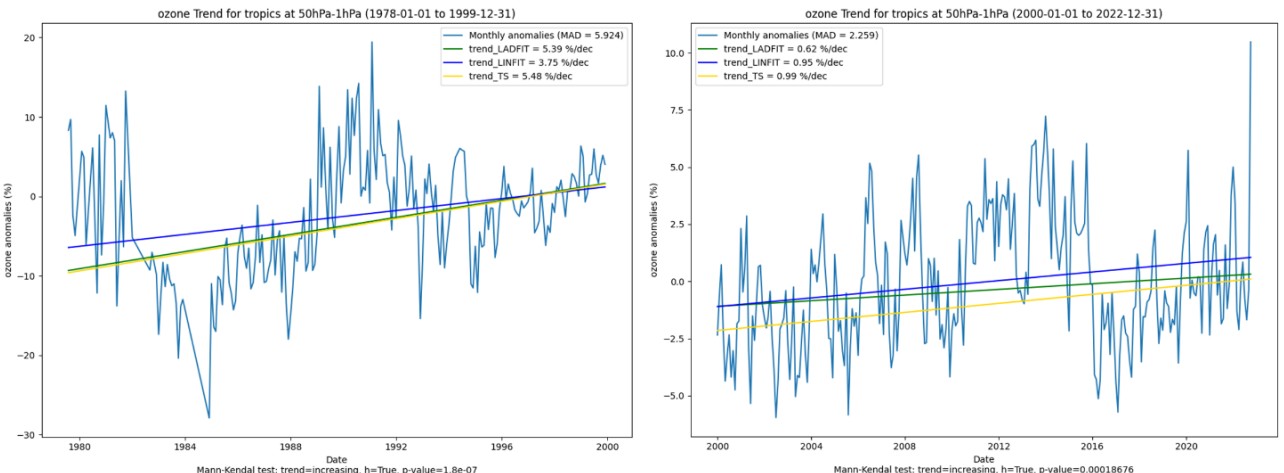

**Figure 12. Same as Figure 10 but for the TR latitudinal sector using LMC cluster data.**




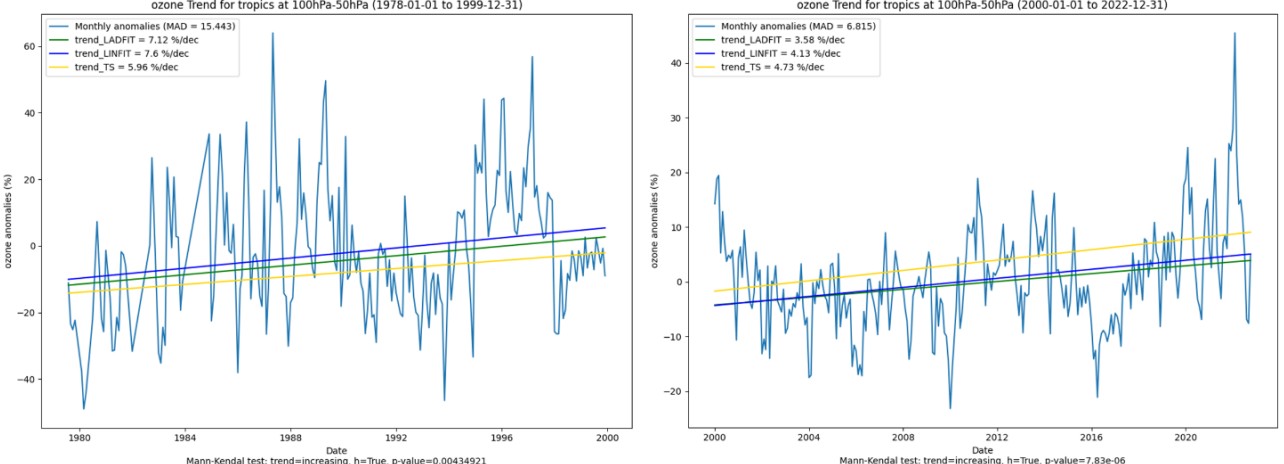

**Figure 13. Same as Figure 12 but for the vertical range 100-50hPa.**

In contrast to the estimates discussed in this study, which in the period 1978-1999 indicate a positive trend of about 5% per
decade in the lower stratosphere at 50-1 hPa, reaching 7% per decade at 100-50 hPa, the trends presented by
Petropavlovskikh et al. (2019) suggest a negative trend of 2% per decade at 50-1 hPa and a more pronounced negative trend
of 10% per decade at 100-50 hPa for the 1978-1999 period. This discrepancy is likely attributed to the limited sample size
available for the period 1978-1999, despite utilizing LMC cluster data to estimate the trend. However, for the 2000-2022
period, Petropavlovskikh et al. (2019) report a positive trend of 2% per decade in the TR lower stratosphere at 50-1 hPa,
consistent with Sofieva et al. (2021) (2% per decade positive trend), and a positive trend of 8% per decade at 100-50 hPa.
The latter value, with an uncertainty limit of 7%, aligns with the trend estimated from the LMC cluster (4% per decade
positive trend), falling within the range of uncertainty. Furthermore, for the lower stratosphere at 50-1 hPa, the LMC cluster
shows a positive trend of 1% per decade for 2000-2022, which is comparable to the values reported by Petropavlovskikh et
al. (2019) and Sofieva et al. (2021).

Figure 12 also shows the sudden post-2013 total column ozone (TCO) "dropoff" of a few percent in the ozone anomalies due
to one of the two instruments used for the vertical soundings (Stauffer et al., 2020; Stauffer et al., 2022; Nakano and
Morofuji, 2023), attributed to apparent anomalous losses of ozone in the lower and middle stratosphere, resulting in several
percentage points deviation from the averages observed from 2014 to 2017. This dropoff could influence ozone trend
calculations using tropical ozonesonde data. In this study, data from Izana, Naha, Hilo, and Samoa stations were utilized to
estimate ozone trends for 2000-2022 in the TR sector. Stauffer et al. (2022) indicate that not all tropical stations are affected
by this sudden low bias; specifically, Izana and Naha do not experience a dropoff, unlike Hilo and Samoa, which are
impacted. However, considering that the trends estimated with the LMC cluster align with those reported by
Petropavlovskikh et al. (2019) and Sofieva et al. (2021), not exclusively based on in situ soundings (Stauffer et al., 2020), it
can be inferred that the influence of the dropoff is not dramatically affecting for trend estimation over the last two decades.



The anomalies depicted in Figures 10-13 also underscore the impact of sulphate aerosols released from volcanic eruptions on ozone concentrations in the stratosphere. Two major eruptions occurred during the period considered for analysis: El Chichón in 1982 and Pinatubo in 1991. Solomon et al. (1998) elucidates the effects of stratospheric sulphate aerosols on ozone, highlighting the challenges faced by some instruments, both satellite and ground-based, in measuring ozone during periods of elevated stratospheric aerosol concentrations, leading to the exclusion of several years of data from datasets

following major volcanic eruptions. Unlike analyses that omit data from the years immediately following eruptions (e.g., Wang et al., 1996; Randel and Wu, 2007; Harris et al., 2015), this study considers the entire time series. For the NH, two negative peaks of approximately 20%, in 1983 and 1993, resulting from the eruptions, are evident in Figure 11 (left panel). Nonetheless, the estimated trend appears consistent with that reported by Petropavlovskikh et al. (2019), indicating that the impact of these eruptions does not significantly alter the overall trend for the 1978-1999 period in the NH. In the case of

TRs, Figure 12 (left panel) displays two negative peaks, one of 27% and one of 15%, in 1985 and 1993, respectively. The estimated trend exhibits larger disparities compared to those reported by Petropavlovskikh et al. (2019), primarily due to the limited sample size available for 1978-1999.

## 5 Summary

This study investigates the impact of the sampling frequency on the estimation of the decadal trends of ozone partial

pressure. These are evaluated using a unified dataset comprised of ozonesounding profiles provided by SHADOZ, NDACC and WOUDC datasets, selected with sufficient quality in terms of plausibility, coherency, and consistency in the measurements. Moreover, the trends have been estimated for different atmospheric layers and latitude ranges, using three linear regression methods (least-square, least absolute deviation and Theil-Sen), which have been compared to evaluate the structural uncertainties. The effect on the time series homogeneity of the dataset merging has been quantified using the

SNHT test and results indicate moderate to minimal changes post-merging. The individual stations have been classified into clusters, depending on their data coverage, that have been compared to evaluate the impact of spatial sampling on the estimation of trends.

The comparison finally indicated that:

- The difference between the trends for the clusters reveals that usage of the smallest but longer cluster, the "long

coverage" cluster (LC), provides the largest number of significant trends in the Northern Hemisphere mid-latitude (NH) at the different vertical and latitude ranges, with a small effect of the spatial sampling, lower than 2% at 100-50 hPa and 1.5% at 50-1 hPa, and the smallest structural uncertainties, not exceeding 1.05% at 100-50hPa and 1.15% at 50-1 hPa. This cluster has good representativeness on a global scale, except for the tropical region due to the paucity of available stations meeting the quality criteria, above all in terms of data coverage and completeness.

For this reason, in the Tropics, it is recommended the usage of the "long and medium coverage" cluster (LMC), which provides reliable trend estimation and better spatial representativeness compared to LC, with sampling errors up to about 3% at 100-50 hPa and 3.5% at 50-1 hPa and structural uncertainties lower than 2% at both 100-50 hPa and at 50-1 hPa.



- In the NH, there is a negative trend of 5% per decade for the period 1978-1999 at 50-1 hPa layer, reaching a negative trend of 10% per decade at 100-50 hPa, and a negative trend of 1% per decade for the period 2000-2022 at 50-1 hPa, with a positive trend of 4% per decade at 100-50 hPa. For the Tropics sector in the period 1978-1999, a positive trend of about 5% per decade at 50-1 hPa is found, reaching 7% per decade at 100-50 hPa and, for 2000-2022, a positive trend of 1% per decade at 50-1 hPa, and a positive trend of 4% per decade at 100-50 hPa. These trends have been also compared with other studies already present in the literature (Petropavlovskikh et al., 2019;

Sofieva et al., 2021), indicating compatibility for the NH in both periods and for the TR only in the period 2000-2022, due to disagreement for the period 1978-1999 caused by the limited sample size available.

The ozonesounding unified dataset improves spatial and temporal coverage, facilitating more robust analyses of ozone anomalies and trends worldwide compared to utilizing individual datasets. Offering a harmonized and quality-checked solution, such a dataset eliminates the necessity for climate researchers to download multiple datasets, thus streamlining their
workflow and enhancing efficiency, although minimizing the negative effect on the quality of the ozone concentration data due to the dataset merging. Future work will be oriented to make comparisons with other regression models, such as the LOTUS (Petropavlovskikh et al., 2019; Godin-Beckmann et al., 2022), which is based on the classic multiple linear regression method, and with time series from the HEGIFTOM-II (Harmonization and Evaluation of Ground-based Instruments for Free Tropospheric Ozone Measurements) homogenized NDACC dataset (https://hegiftom.meteo.be/).

**Appendix A: station list for the UNIFIED dataset**

**Table A1. Information about the station's coverage classification.**

| Station | Longest continuous data series | Classification |
|---------|-------------------------------|----------------|
| ABERYSTWYTH | 1 year | Short coverage |
| AINSWORTH | 1 month | Short coverage |
| ALAJUELA | 4 years | Short coverage |
| ALBROOK | 1 month | Short coverage |
| ALERT | 33 years | Long coverage |
| AMUNDSEN-SCOTT | 2 years | Short coverage |
| ANKARA | 6 years and 5 months | Short coverage |
| ASCENSION_ISLAND | 11 years and 3 months | Medium coverage |
| ASPENDALE | 4 years and 8 months | Short coverage |
| BARBADOS | 1 month | Short coverage |
| BARROW | 1 month | Short coverage |
| BEDFORD | no data between 1978-01-01 and 2021-07-31 | Short coverage |
| BELGRANO | 4 years and 2 months | Short coverage |
| BELTSVILLE | 1 month | Short coverage |
| BERLIN/TEMPLEHOF | no data between 1978-01-01 and 2021-07-31 | Short coverage |
| BISCARROSSE/SMS | 5 years | Short coverage |





| | | |
|---|---|---|
| BOGOTA | 2 years and 2 months | Short coverage |
| BOMBAY | no data between 1978-01-01 and 2021-07-31 | Short coverage |
| BOULDER | 29 years and 2 months | Long coverage |
| BRATTS_LAKE | 8 years | Short coverage |
| BRAZZAVILLE | 2 years and 6 months | Short coverage |
| BROADMEADOWS | 21 years and 11 months | Long coverage |
| BYRD | no data between 1978-01-01 and 2021-07-31 | Short coverage |
| CAGLIARI/ELMAS | 2 years and 6 months | Short coverage |
| CANTON_ISLAND | no data between 1978-01-01 and 2021-07-31 | Short coverage |
| CARIBOU | 1 month | Short coverage |
| CHEJU | 4 months | Short coverage |
| CHILCA | no data between 1978-01-01 and 2021-07-31 | Short coverage |
| CHRISTCHURCH | no data between 1978-01-01 and 2021-07-31 | Short coverage |
| CHURCHILL | 19 years | Medium coverage |
| COLD_LAKE | 4 years | Short coverage |
| COOLIDGE_FIELD | no data between 1978-01-01 and 2021-07-31 | Short coverage |
| COSTA RICA | 11 years | Medium coverage |
| COTONOU | 2 years | Short coverage |
| CUIABA | 2 months | Short coverage |
| DAVIS | 6 years and 8 months | Short coverage |
| DEBILT | 28 years and 1 month | Long coverage |
| DENVER | no data between 1978-01-01 and 2021-07-31 | Short coverage |
| DUMONT | 11 years | Medium coverage |
| EASTER_ISLAND | 2 years | Short coverage |
| EDMONTON STONY_PLAIN | 42 years and 11 months | Long coverage |
| EGBERT | 8 years | Short coverage |
| EL_ARENOSILLO | 1 month | Short coverage |
| ETOSHA_PAN | 2 months | Short coverage |
| EUREKA | 27 years and 2 months | Long coverage |
| FAIRBANKS | no data between 1978-01-01 and 2021-07-31 | Short coverage |
| FT._SHERMAN | no data between 1978-01-01 and 2021-07-31 | Short coverage |
| GIMLI | 5 years and 1 month | Short coverage |
| GOOSE_BAY | 35 years and 1 month | Long coverage |
| GREAT_FALLS | no data between 1978-01-01 and 2021-07-31 | Short coverage |
| HALLETT | no data between 1978-01-01 and 2021-07-31 | Short coverage |
| HANOI | 5 years | Short coverage |
| HEREDIA | 1 year and 2 months | Short coverage |
| HILO | 38 years | Long coverage |





| | | |
|---|---|---|
| HOBART | 1 month | Short coverage |
| HOHENPEISSENBERG | 43 years and 5 months | Long coverage |
| HOLTVILLE | 1 month | Short coverage |
| HONG_KONG_OBSERVATORY | 12 years and 11 months | Medium coverage |
| HOUSTON | 5 months | Short coverage |
| HUNTSVILLE | 8 years and 8 months | Short coverage |
| IQALUIT | 5 months | Short coverage |
| IRENE | 7 years and 7 months | Short coverage |
| ISFAHAN | 2 years and 1 month | Short coverage |
| IZANA | 25 years and 7 months | Long coverage |
| JOKIOINEN | 2 years | Short coverage |
| KAASHIDHOO | 2 months | Short coverage |
| KAGOSHIMA | 14 years | Medium coverage |
| KELOWNA | 11 years and 7 months | Medium coverage |
| KOUROU | no data between 1978-01-01 and 2021-07-31 | Short coverage |
| KUALA LUMPUR | 12 years and 1 month | Medium coverage |
| LA_REUNION_ISLAND | 12 years and 7 months | Medium coverage |
| LAUDER | 35 years and 2 months | Long coverage |
| LAVERTON | 6 years and 4 months | Short coverage |
| LEGIONOWO | 30 years and 8 months | Long coverage |
| LERWICK | 12 years and 10 months | Medium coverage |
| LINDENBERG | 37 years | Long coverage |
| LONG_VIEW | no data between 1978-01-01 and 2021-07-31 | Short coverage |
| MACQUARIE_ISLAND | 17 years and 1 month | Medium coverage |
| MADRID BARAJAS | 17 years | Medium coverage |
| MAITRI | 4 years and 11 months | Short coverage |
| MALINDI | 6 years and 10 months | Short coverage |
| MARAMBIO | 15 years and 7 months | Medium coverage |
| MCDONALD_OBSERVATORY | no data between 1978-01-01 and 2021-07-31 | Short coverage |
| MCMURDO | 11 months | Short coverage |
| MIRNY | 2 years and 5 months | Short coverage |
| MOUNT_ABU | no data between 1978-01-01 and 2021-07-31 | Short coverage |
| NAHA | 29 years and 8 months | Long coverage |
| NAIROBI | 13 years and 7 months | Medium coverage |
| NARRAGANSETT | 5 months | Short coverage |
| NATAL | 12 years and 6 months | Medium coverage |
| NEUMAYER | 28 years and 7 months | Long coverage |
| NEW_DELHI | 6 years | Short coverage |





| | | |
|---|---|---|
| NOVOLASAREVSKAYA/FORSTER | 5 years and 9 months | Short coverage |
| NY_ALESUND | 30 years | Long coverage |
| OHP | 16 years and 9 months | Medium coverage |
| OVEJUYO | no data between 1978-01-01 and 2021-07-31 | Short coverage |
| PAGO PAGO AMERICAN SAMOA | 25 years | Long coverage |
| PALESTINE | 6 years and 8 months | Short coverage |
| PAPEETE | 4 years and 5 months | Short coverage |
| PARADOX | 2 months | Short coverage |
| PARAMARIBO | 13 years and 2 months | Medium coverage |
| PAYERNE | 42 years and 8 months | Long coverage |
| PELLSTON | 2 months | Short coverage |
| PENGCHIAYU | 7 months | Short coverage |
| PETALING_JAYA | 2 years and 3 months | Short coverage |
| POHANG | 3 years | Short coverage |
| POKER_FLAT | 3 years | Short coverage |
| PORT_HARDY | 2 years and 6 months | Short coverage |
| PORTO_NACIONAL | 2 months | Short coverage |
| PRAHA | 3 years | Short coverage |
| PUERTO_MONTT | no data between 1978-01-01 and 2021-07-31 | Short coverage |
| PUNE | 6 years and 10 months | Short coverage |
| RESOLUTE | 15 years and 10 months | Medium coverage |
| RICHLAND | 2 months | Short coverage |
| S.PIETRO_CAPOFIUME | 2 years and 9 months | Short coverage |
| SABLE_ISLAND | 2 months | Short coverage |
| SALEKHARD | 2 years and 1 month | Short coverage |
| SAN_CRISTOBAL | 3 years and 6 months | Short coverage |
| SAN_DIEGO | no data between 1978-01-01 and 2021-07-31 | Short coverage |
| SAN_JUAN | no data between 1978-01-01 and 2021-07-31 | Short coverage |
| SAPPORO | 28 years and 2 months | Long coverage |
| SCORESBYSUND | 26 years and 10 months | Long coverage |
| SEPANG_AIRPORT | 12 years and 1 month | Medium coverage |
| SINGAPORE | 4 years | Short coverage |
| SODANKYLA | 23 years and 2 months | Long coverage |
| SOFIA | 9 years and 10 months | Short coverage |
| SOUTH POLE | 29 years and 9 months | Long coverage |
| SPOKANE | no data between 1978-01-01 and 2021-07-31 | Short coverage |
| STERLING | no data between 1978-01-01 and 2021-07-31 | Short coverage |
| SUMMIT | 11 years and 8 months | Medium coverage |



| SUVA | 8 years and 9 months | Short coverage |
|---|---|---|
| SYOWA | 34 years and 7 months | Long coverage |
| TABLE_MOUNTAIN | 6 months | Short coverage |
| TAIPEI | 7 years and 2 months | Short coverage |
| TATENO | 31 years and 6 months | Long coverage |
| TECAMEC | 6 months | Short coverage |
| THALWIL | no data between 1978-01-01 and 2021-07-31 | Short coverage |
| THIRUVANANTHAPURAM | 7 years | Short coverage |
| THULE | 4 months | Short coverage |
| TOPEKA | no data between 1978-01-01 and 2021-07-31 | Short coverage |
| TORONTO | 3 months | Short coverage |
| TRINIDAD_HEAD | 2 years and 5 months | Short coverage |
| UCCLE | 33 years and 5 months | Long coverage |
| USHUAIA | 3 years and 8 months | Short coverage |
| VALENTIA | 16 years | Medium coverage |
| VALPARAISO | 4 months | Short coverage |
| VANSCOY | 2 years and 2 months | Short coverage |
| VIGNA_DI_VALLE | 4 years 10 months | Short coverage |
| WALLOPS_ISLAND | 26 years and 2 months | Long coverage |
| WALSINGHAM | 2 months | Short coverage |
| WATUKOSEK | 15 years and 2 months | Medium coverage |
| WILKES | no data between 1978-01-01 and 2021-07-31 | Short coverage |
| YAKUTSK | 4 months | Short coverage |
| YARMOUTH | 17 years and 2 months | Medium coverage |
| YORKTON | 1 month | Short coverage |

**Data Availability**

Data supporting this study are openly available from Zenodo at https://doi.org/10.5281/zenodo.12544883 (Marra et al.,
575   2024).

**Contribution of the authors**

Fabrizio Marra created the unified database on which he performed the analyses, developing the software in Python, and
wrote the manuscript. Emanuele Tramutola helped create the software, ensuring its efficiency and correctness. Marco
Rosoldi provided suggestions for improving the manuscript. Fabio Madonna initiated the project, coordinated with Fabrizio





Marra to ensure that the analysis was in line with published advances in the literature, and contributed to the editing and final drafting of the manuscript.

**Competing interest**

The authors declare that they have no conflict of interest.

**Acknowledgments**

Station representativeness maps were produced using an R code developed by Weatherhead E. According to the approach presented in Weatherhead and co-authors (2017), DOI: https://doi.org/10.1175/JAMC-D-17-0040.1.

This work was done on behalf of the European Union's Copernicus Climate Change Service implemented by ECMWF. Use of the WOUDC data as stated in the Copernicus license agreement is acknowledged.

The data used in this publication were obtained from NDACC PIs as part of the Network for the Detection of Atmospheric

Composition Change (NDACC) and are available through the NDACC website www.ndacc.org.

The use of high-quality SHADOZ data records in this work was possible thanks to the meticulous work of PIs and staff at ozonesonde station.

The SHADOZ and NDACC ozonesonde data were downloaded from the NASA/GSFC (at https://tropo.gsfc.nasa.gov/shadoz/) and NASA/LaRC (at https://www-air.larc.nasa.gov/pub/NDACC/), while WOUDC from

C3S/CDS (at https://cds.climate.copernicus.eu/cdsapp#!/dataset/insitu-observations-woudc-ozone-total-column-and-profiles).

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
