# Peer review of "Effect of sampling error on ozone partial pressure trends within a unified ozonesounding dataset"

_EGUsphere, 2024_

## Editor Comment (EC1)

Dear authors,

as a representative for the ozonesonde network, I'm really grateful for your effort in filtering and cleaning up the ozonesonde data, which are available in three (major) existing international archives, but in different formats and not necessarily identical. The Unified Ozonesonde Dataset is certainly a step in the right direction to ease data access and data use!

As for every ground-based (or in-situ) dataset, one of the major challenges of the ozonesonde data is the spatial and temporal representativeness for measuring (here) stratospheric ozone. In this study, an attempt has been done to analyze its impact on stratospheric ozone trends by considering two different subsets of the Unified Ozonesonde Dataset: the "Large coverage" and Large and medium coverage" datasets, and this for different latitude bands.

**General comments**

The authors should also give some remaining caveats associated with the Unified Ozonesonde Dataset:

- It should be clearly mentioned that "unified" is a different concept than "homogeneous". As you know, the ozonesonde community has been undertaken a homogenization activity to process all the ozonesonde data across the world according to the same principles (e.g. removal of known biases by referring all data to the same standards, consistent correction procedures for known effects, uncertainty calculation). These homogenized data are stored on a ftp-server with access details provided on the HEGIFTOM website, and also on SHADOZ, and for a handful of sites only in WOUDC or NDACC.
- As a consequence of the previous point, the Unified Ozonesonde Dataset will merge data at a given site that have been processed differently (e.g. homogenized in SHADOZ vs. non-homogenized in WOUDC or NDACC; the Payerne data has been corrected for total ozone normalization in one archive, but not in the other archive), which can give rise to biases between data points. This should be mentioned.

Although the primary focus and expertise of the authors lie in the data polishing to construct the Unified Ozonesonde Dataset, the authors also want to contribute to the calculation of stratospheric ozone trends from this dataset. They apply common approaches or assumptions, but no justification or interpretation is given why

- two different trend periods are considered (1978-1999, 2000-2022)
- a linear regression trend model is used
- zonal trends are considered (i.e. different trends for tropics, mid-latitudes, polar regions)
- different vertical trends are calculated (i.e. between 300-200, 200-100, 100-50, and 50-1 hPa)

The authors should give an interpretation or explanation why trends differ (or not) between the two periods, between the different latitudinal zones, and between the different vertical ranges. The latest WMO Scientific Assessment of Ozone Depletion can be used as guidance here (https://ozone.unep.org/sites/default/files/2023-02/Scientific-Assessment-of-Ozone-Depletion-2022.pdf).

The assessment of the representativeness of the unified stations in terms of representing the total ozone column field (Fig. 9) is a nice feature, but also raises some important questions:

- what is the consequence of these correlations for the representativeness of the LMC and LC datasets for the ozone concentrations in the vertical ranges considered (300-200, 200-100 hPa, etc)? Should this analysis not be performed for those vertical ranges (e.g. making use of the MLS satellite vertical ozone retrievals) instead?
- Can we draw the conclusion from this graph that LC dataset is not representative for the tropical total ozone band, while the LMC might be?
- Can we draw the conclusion that neither the LMC or the LC datasets are representative for the entire NH mid-latitude total ozone band, and what does this mean for the NH mid-latitude trends calculated from these datasets in the different vertical ranges?

On top of this correlation analysis, to assess the impact that the LMC and LC datasets have on the different vertical ozone trends, it would also be good to show in the manuscript (or provide in the supplementary material) the monthly anomaly time series (and calculated trends) for both datasets (i.e. LMC for Figs. 10-11 and LC for Figs. 12-13), as in the PhD Thesis this manuscript is based on. Those plots would be more illustrative and informative (i.e. better guidance for interpretation) than the tables 7 to 13). Based on those plots in your PhD Thesis, when comparing the ozone time variability (not only the linear trend) in different layers and for different latitude bands (Figs. 32-39), a very different time variability (in terms of features, spread in the variability) arises between the "Large coverage (LC)" and "Large and Medium coverage (LMC)" clusters, illustrating the strong dependence of this time variability on the chosen individual sites in the clusters. The merging of individual sites with different onsets of the ozonesonde observation program (as is done in the comparison between the LC and LMC clusters) gives a different spatial and temporal distribution of ozonesonde sites (or at least a different weighting of the contributions of individual sites) between the beginning and end of the periods for which ozone anomalies/trends have been calculated. This has of course consequences for the calculated trends, but this impact has not been investigated in enough detail.

The discussion of the trend differences (based on the tables 7 to 13) between the different trend estimation tools and the different datasets or vertical ranges really needs the inclusion (in the table and in the interpretation) of the trend uncertainties! I know that the significance of the trend is mainly assessed by using the MK test statistic, but every used linear regression trend estimation tool provides an uncertainty estimation on the slope coefficient (i.e. trend), which should be displayed and used in the discussion between the trend estimates. In the trend discussion, it is also important to note that "no significant trend" can be a physical result, and the presence of a trend does not mean that one dataset is more "reliable", more "robust", or "outperforms" another one. In my specific comments, I'll point to these wordings.

Nowadays, most trend estimation tools for stratospheric ozone trend assessment make use of different proxies for attributing some dynamical behavior (QBO, ENSO, solar cycle, stratospheric AOD) in the ozone time series (e.g. LOTUS multiple linear regression model). In the ozone monthly anomaly time series plots (Figs. 10 to 13), which show a distinct non-linear behavior, only the impact of volcanic eruptions on the ozone data has been discussed. The authors might also discuss the possible impact of e.g. QBO and ENSO

on the tropical ozone time series or, even better, use a multiple linear regression technique to give more credibility to the estimated trends and better enable the interpretation of the long-term time behavior.

At several places, the authors do not give enough details and more clarifications or interpretation is needed. In the specific comments, I'll give an overview.

**Specific comments**

- Lines 42-43: nowadays, ozonesonde measurements are not normalized to the total ozone column anymore
- In the introduction: mention the existence of the homogenized ozonesonde data (O3S-DQA activity or HEGIFTOM website) and the difference in concept between "unified" and "homogenized".
- Line 100: it is not clear which selection criterion is used here. Please specify "the dataset's maturity and availability of measurement uncertainties on ozone concentration profiles selection criteria"!
- Figures 4 and 5: ozone(sonde) profiles are commonly displayed with the ozone partial pressure (in mPa) on the horizontal axis (x-axis), with the vertical coordinate (pressure here, decreasing in amplitude) on the vertical axis (y-axis).
- Lines 165-170. The criteria for the different datasets (LC, MC, SC) are incomplete and perhaps even not objective. As it is written now, Praha and Macquarie Island sites should be LC stations, not a SC and MC station, respectively. It looks like the criterion is "20 full years" for a SC station, and Macquarie Island is "degraded" to a MC station because it has a 3-month gap in 2003. This rises two questions: (i) how flexible have you been with those criteria for other sites (this is not obvious and some subjective assessment cannot be ruled out → make it objective), (ii) to which extent would a 3-month gap (or larger) has a significant impact on the calculated trend over an at least 20 year time period?
- Figures 7 and 9: as your different latitudinal regions are marked by 30°, use an increment of 30° instead of 25° (Figure 7) or 50° (Figure 9) for the latitude axis.
- Line 219: same comment as Reviewer 1: I don't understand the completeness check criterion, as most ozonesonde sites have weekly launches.
- Table 3: the difference between the flagged NDACC percentages for the ozone partial pressures and ozone concentrations is striking. It is not because the ozone concentrations (in ppmv) are not provided, that they could not be easily calculated from the ozone partial pressures (in Pa). The flagged percentages for those two variables should be very similar, as for SHADOZ. So, what is the meaning (and relevance) of this 86.22% flagged NDACC data for the ozone concentrations?
- Line 229: vertical completeness check. "At least one point every 50 m in an ozone profile is required". With a typical rise rate of 5 m/s and an inherent response time of the ozonesonde of about 18-28s, the effective vertical resolution of the ozone profile is 100-150 m. So, how relevant is the 50m criterion?
- Line 238: Give more details on the single quality flag you generated for each ozone profile: what are the possible values and how have the different quality checks been compiled to obtain one single flag?

- Fig 8: specify that the bad profile in the right panel is the brown one.
- Section 2.5: Several clarifications/additions are needed here. First, clearly mention if the correlations between the total ozone column measurements have been done for deseasonalized monthly values. Secondly, it is also important to add that/if the total ozone column from the ozonesonde profiles have been obtained by integrating the profile, and that the ozone column above the burst altitude is missing w.r.t. the TOMS-EP total ozone measurements. Thirdly, also add how the correlation maps in Fig. 9 have been obtained (you calculate for each grid point the correlation coefficient with each unified station, so which of those correlation coefficients is displayed in Fig. 9? Maximum correlation coefficient, as suggested by Weatherhead et al.?). And finally, mention from which correlation coefficient we can speak of good correlation (according to Weatherhead et al., I think it is higher than 0.7).
- Caption Fig. 9: you should mention that the (linear) Pearson correlation coefficients are calculated between the deseasonalized monthly averages from the total column ozone measurements (please confirm).
- Section 3.3: this section does not provide what is announced in its first sentence (lines 317-318). As far as I understand from this sentence (and from my experience with statistical breakpoint detection tools as SNHT), you compare the individual ozonesonde time series with the corresponding unified ozonesonde time series at the same site and the SNHT test looks for a breakpoint in the difference time series (a significant shift in the mean of the segments before and after the breakpoint). The presence of such breakpoints in the unified vs. individual time series will bring very relevant information. Instead, in Table 5, you provide the mean and maximum values of the SNHT test statistic, but it is far from clear what those values represent and no real interpretation is given. Much more useful information would be the time epochs for which breakpoints in the difference time series are detected. A breakpoint (shift in the mean) in one time series w.r.t. the other would result in trend differences between the two time series, which is very relevant for the remaining of the paper. This section should be seriously rewritten to better reflect its aim, as was outlined in the first sentence.
- Tables 7 to 13: I find it very strange that you include the latitudinal zones SP and SH in your analysis, as you write in lines 201-204: "Due to the limited availability of LC and MC stations in the Southern Polar (SP) and Southern Hemisphere (SH), the analysis presented hereinafter is not conducted on these two regions. Although the related estimated trends are significant, the scarcity of stations' number and data availability in these sectors substantially enhances the uncertainties on the calculated trends, making them unreliable." This is really a contradiction with including the trends of these regions in the tables. Therefore: drop those zones in the tables and discussion, you gave the arguments for doing so.
- Tables 9, 10, 11, 12, 13: add the (two-sigma) uncertainties of the trend estimations, also in the discussion!
- Line 356: argue why you consider two different periods 1978-1999 and 2000-2022 for calculating trends.
- Line 357: in your analysis, you do not explicitly discriminate between geographical and temporal sampling impacts on trends, so add "temporal" here.

- Line 367-368: as you do not consider the MC cluster trends (time series not long enough, see line 172), this statement cannot be verified.
- Line 371: I don't have a clue what is meant with "In the TR, results resemble those of the SP."
- Line 372 "the LC lacks significant trends in certain pressure ranges due to the limited number of stations", Line 375 "LC showing slightly better performance via the MK test", Line 381 "show comparable abilities to estimate significant changes", Line 418 "This performance improvement", Line 425 "trend estimates are considered unreliable", Line 429 "Notably, the performance of the LMC cluster in the TR sector outperforms that of the LC cluster in this layer", Line 440 "providing a more robust dataset for trend estimation, …: all these statements assume that no trend means a worse performance of a dataset for trend calculation, and do not take into account that no significant trend can be a true, physical result for some zones, vertical ranges or time periods. The entire discussion in this section should be realigned in this sense. As mentioned earlier, the comparison of the monthly anomaly time series plots for the two datasets (LC vs. LMC) will reveal more insight whether or not they both display a similar time variability (or trend).
- Line 420-421: what is meant with "enhanced representativeness". This cannot be deduced from the trend analysis presented here, right?
- Line 439-440: "The most suitable vertical ranges for trend analysis are 50-1 hPa and 100-50 hPa due to their richer data content, providing a more robust dataset for trend estimation." It is not the amount of points in the vertical range that plays a role, but the dispersion/variability between the points, because the means are considered to calculate trends. If you want to make a comparison between the different vertical ranges, the standard deviation of the means would be a better metric to compare instead of the number of data points.
- Section 4.1: when comparing your trends with the estimates from other studies, you only consider the trend uncertainties from the other studies; include yours in the comparison as well.
- Section 4.2, lines 502-503: which tropical stations have data before 1998? Mention those!

---

## Author Comment (AC1)

**Dear Reviewer,**

**Thank you very much for your thorough review and for the time you have dedicated to evaluating our paper. We greatly appreciate your positive feedback and constructive comments. Below, we have included your comments in *italics* and our responses in bold.**

*"Review of "Effect of sampling error on ozone partial pressure trends within a unified ozonesounding dataset" Fabrizio Marra, et al.*

*Summary and General Comments:*

*This paper describes the construction of a "unified" ozonesonde dataset gathered from WOUDC, SHADOZ, and NDACC archives and the calculation of ozone profile trends in segments from the upper troposphere to the lower stratosphere for several latitudinal bands and two time periods. Various methods are used to derive the trend estimates to understand their uncertainties and how data sampling affects them. The authors find that ozone trends calculated over the periods 1978-1999 and 2000-2022 from their unified dataset agree fairly well with independent studies that include satellite datasets with greater spatiotemporal sampling.*

*This paper is well motivated and written. Analyzing a fully global set of ozonesonde data that appears in various archives is a challenge, so this study results in two key accomplishments: 1) constructing a quality-checked global ozonesonde dataset that avoids duplication and captures data that would be missed by using only one archive, and 2) demonstrating the utility of the unified dataset for trends calculations and comparison to existing studies."*

**Thank you for your positive feedback and for acknowledging the key findings of our study. We are glad to hear that you found our article well motivated and written.**

*"Uncertainties of the trends (95% confidence interval at least, and perhaps also p-value) presented in Tables 9-12 should be reported so we can understand the significance (or insignificance) of the differences in trends computed using various methods. This would also allow you to also report a trend that is not statistically significant, rather than listing "NT"."*

**We have modified Tables 9-12 to include trend uncertainties, calculated via bootstrapping, for each regressor. Additionally, we have incorporated the LOTUS regressor along with the methods used previously. We have focused on reporting trends for the Northern Hemisphere (NH), the Tropics (TR), and the North Pole (NP), as data availability and the number of stations for the Southern Hemisphere (SH) and the South Pole (SP) are insufficient to ensure reliable trend estimates. These modifications aim to provide a clearer understanding of both significant and non-significant trends. Below, as also reported in our response to the editor's review, are the new Tables 9-12 with updated discussion at lines 391-442:**

**"**

*Table 9. Trend estimates, with the uncertainty, as percentage per decade (% dec-1) obtained with the Linear (LIN), Least Absolute Deviation (LAD) and Theil-Sen (TS) and LOTUS regressors for all latitudinal sectors considered at the 50-1hPa vertical range for each cluster. Legend: Positive value (increasing trend); Negative value (decreasing trend); NT (No Trend).*

| 50-1hPa | | % dec$^{-1}$ | TR | NH | NP |
|---|---|---|---|---|---|
| **1978-1999** | **LC** | LIN | 7 ± 1 | -5.9 ± 0.5 | -10 ± 2 |
| | | LAD | 8 ± 1 | -5 ± 1 | -10 ± 6 |
| | | TS | 7 ± 2 | -6 ± 2 | -8 ± 8 |
| | | LOTUS | 8 ± 1 | -6.3 ± 0.2 | -10 ± 1 |
| | **LMC** | LIN | 4 ± 1 | -5.6 ± 0.5 | -14 ± 2 |
| | | LAD | 5 ± 2 | -5 ± 1 | -12 ± 3 |
| | | TS | 7 ± 3 | -6 ± 2 | -11 ± 5 |
| | | LOTUS | 4 ± 1 | -6.0 ± 0.2 | -14.9 ± 0.4 |
| **2000-2022** | **LC** | LIN | NT | -1.1 ± 0.4 | -6 ± 1 |
| | | LAD | NT | -1 ± 1 | -5 ± 2 |
| | | TS | NT | -1 ± 1 | -6 ± 2 |
| | | LOTUS | NT | -1.1 ± 0.1 | -5.4 ± 0.2 |
| | **LMC** | LIN | 0.7 ± 0.2 | NT | -5 ± 1 |
| | | LAD | 0.6 ± 0.6 | NT | -5 ± 2 |
| | | TS | 0.7 ± 0.7 | NT | -5 ± 2 |
| | | LOTUS | 0.9 ± 0.1 | NT | -4.6 ± 0.2 |

At the 50-1 hPa vertical range (Table 9) for the TR during the period 1978-1999, estimates display discrepancies of up to about 4% ±2% per decade. This discrepancy likely arises from the greater number of stations available for the LMC cluster, enhancing coverage and representativeness compared to the LC cluster, as discussed in Section 2.5.

For the NH, the clusters agree for the 1978-1999 period (with a maximum discrepancy less than 0.5% ±0.7% per decade), but in contrast for 2000-2022, where the trend of the LMC cluster, for the MK test, is not significant.

Finally, in the NP, trends correspond only in the 2000-2022 period (with a maximum discrepancy of 1% ±2% per decade for the TS regressor). For 1978-1999, differences among clusters nearly reach 5% ±2% per decade.

*Table 10. Same as Table 9 but for the vertical range 100-50 hPa.*

| 100-50hPa | | % dec⁻¹ | TR | NH | NP |
|---|---|---|---|---|---|
| **1978-1999** | **LC** | **LIN** | 12 ± 2 | -9 ± 1 | NT |
| | | **LAD** | 14 ± 7 | -8 ± 1 | NT |
| | | **TS** | 11 ± 10 | -9 ± 2 | NT |
| | | **LOTUS** | 11 ± 1 | -8.7 ± 0.4 | NT |
| | **LMC** | **LIN** | 8 ± 2 | -10 ± 1 | -13 ± 1 |
| | | **LAD** | 8 ± 7 | -10 ± 2 | -12 ± 3 |
| | | **TS** | 2 ± 9 | -9 ± 3 | -12 ± 4 |
| | | **LOTUS** | 8 ± 1 | -10.4 ± 0.4 | -12.9 ± 0.5 |
| **2000-2022** | **LC** | **LIN** | 4 ± 1 | 3 ± 1 | NT |
| | | **LAD** | 3 ± 2 | 4 ± 1 | NT |
| | | **TS** | 4 ± 3 | 4 ± 1 | NT |
| | | **LOTUS** | 2.5 ± 0.2 | 2.8 ± 0.1 | NT |
| | **LMC** | **LIN** | 4 ± 1 | 3 ± 1 | NT |
| | | **LAD** | 3 ± 2 | 3 ± 1 | NT |
| | | **TS** | 2 ± 3 | 3 ± 1 | NT |
| | | **LOTUS** | 1.6 ± 0.4 | 2.9 ± 0.1 | NT |

In Table 10, focusing on the 100-50 hPa range, the NH sector stands out with trend estimates in the period 2000-2022 showing 1% ±1% per decade sampling error and significant trend estimates, attributed to higher data density and greater variability in this vertical range. Nonetheless, even in NH, the LC and LMC clusters exhibit discrepancies in the period 1978-1999 of around 2% ±3% per decade in estimated trends. Conversely, the NP sector lacks significant trends, except for the LMC cluster during 1978-1999, with especially noticeable trends.

Additionally, the TR sector presents a considerable discrepancy, in the 1978-1999 period, of 9% ±10% per decade while, in the period 2000-2022, the LC and LMC clusters show discrepancies of 2% ±3% per decade. Interestingly, the trends estimated by the LMC and LC clusters show low discrepancies among regressors with structural uncertainties of 2.4% ±3% per decade and 1.5% ±3% per decade for 1978-1999 respectively. The same cannot be said for 2000-2022 due to the discrepancies of 6% ±9% per decade for the LMC cluster and 3% ±10% per decade for the LC. This performance can be attributed to the paucity of available data for the TR, in 1978-1999, despite the LMC cluster comprised 8 more stations than LC (see Table 2). For conducting trend analysis using data from the unified database, high-quality measurements similar to those from the LC cluster are

**crucial. However, in regions with a limited number of LC stations, such as the tropics (TR), data from the LMC cluster can offer better representativeness than the LC cluster.**

*Table 11. Same as Table 9 but for the vertical range 200-100 hPa.*

| 200-100hPa | | % dec$^{-1}$ | TR | NH | NP |
|---|---|---|---|---|---|
| **1978-1999** | **LC** | LIN | - | -10 ± 1 | NT |
| | | LAD | - | -10 ± 3 | NT |
| | | TS | - | -10 ± 5 | NT |
| | | LOTUS | - | -10 ± 1 | NT |
| | **LMC** | LIN | 10 ± 3 | -13 ± 1 | -19 ± 2 |
| | | LAD | 8 ± 9 | -14 ± 3 | -15 ± 4 |
| | | TS | 10 ± 15 | -13 ± 5 | -19 ± 6 |
| | | LOTUS | 10 ± 1 | -13 ± 1 | -19 ± 1 |
| **2000-2022** | **LC** | LIN | 10 ± 1 | 7 ± 1 | NT |
| | | LAD | 9 ± 4 | 7 ± 2 | NT |
| | | TS | 8 ± 4 | 5 ± 3 | NT |
| | | LOTUS | 7.5 ± 0.4 | 8.1 ± 0.2 | NT |
| | **LMC** | LIN | 6 ± 1 | 6 ± 1 | NT |
| | | LAD | 3 ± 4 | 6 ± 2 | NT |
| | | TS | 6 ± 4 | 8 ± 3 | NT |
| | | LOTUS | 2.3 ± 0.4 | 7.5 ± 0.2 | NT |

**In Table 11, trend estimates in the NP sector are non-significant except for the LMC cluster in 1978-1999. Trend assessments for the LC cluster in the TR for the period 1978-1999 should be approached with caution due to the significantly small amount of data available before 1995 compared to subsequent years, which largely inflates uncertainties on decadal trends. However, it is important to acknowledge that the absence of a significant trend could indeed be a valid physical observation, particularly in regions or periods where ozone dynamics are more stable or less prone to significant change.**

**In addition, for 2000-2022, the LC cluster reveals a structural uncertainty of 2.5% ±4% per decade, and a discrepancy with the LMC of 6% ±4% per decade. Conversely, trend estimates for the LMC cluster in TR are significant, but exhibit considerable discrepancies among regressors, with structural uncertainty of 2% ±15% per decade for 1978-1999 and 3.7% ±4% per decade for 2000-2022. Moreover, the NH sector is the only one with all valid trends, although there is a disagreement between LC and LMC (up to about 4% ±5% per decade for 1978-1999 and 3% ±3% per decade for 2000-2022). Furthermore, it is worth considering the discrepancy between the results of the TS regressor with those of the LIN**

**and LAD regressors (2% ±3% and 3% ±4%, respectively) that occurred for LC in the period 2000-2022. The LIN, LAD and LOTUS regressors agree with each other, reporting a structural uncertainty of 1.1% ±1% per decade, on the contrary TS where this uncertainty rises to around 3% ±3% per decade.**

*Table 12. Same as Table 9 but for the vertical range 300-200hPa.*

| 300-200hPa | | % dec$^{-1}$ | TR | NH | NP |
|---|---|---|---|---|---|
| **1978-1999** | **LC** | **LIN** | 17 ± 3 | -10 ± 1 | 5 ± 5 |
| | | **LAD** | 19 ± 7 | -10 ± 4 | 11 ± 11 |
| | | **TS** | 17 ± 11 | -13 ± 6 | 10 ± 13 |
| | | **LOTUS** | 17 ± 1 | -10.2 ± 0.5 | 5 ± 2 |
| | **LMC** | **LIN** | NT | -12 ± 2 | - |
| | | **LAD** | NT | -11 ± 3 | - |
| | | **TS** | NT | -11 ± 6 | - |
| | | **LOTUS** | NT | -12.8 ± 0.4 | - |
| **2000-2022** | **LC** | **LIN** | 5 ± 1 | 6 ± 1 | NT |
| | | **LAD** | 3 ± 3 | 7 ± 2 | NT |
| | | **TS** | 4 ± 3 | 7 ± 4 | NT |
| | | **LOTUS** | 3.7 ± 0.2 | 7.6 ± 0.4 | NT |
| | **LMC** | **LIN** | NT | 6 ± 1 | NT |
| | | **LAD** | NT | 7 ± 2 | NT |
| | | **TS** | NT | 5 ± 3 | NT |
| | | **LOTUS** | NT | 7.2 ± 0.4 | NT |

**The scenario for the 300-200 hPa layer (Table 12) is similar to the previous layer. Additionally, this vertical interval exhibits the fewest significant trends. The most suitable vertical ranges for trend analysis are 50-1 hPa and 100-50 hPa, due to their more stable ozone concentrations and lower variability. In contrast, the high variability of ozone near the tropopause complicates the detection of trends in that region. Table 13 highlights the most and least significant disparities in the trends obtained by comparing the different non-parametric regression techniques used with the parametric regressor LOTUS, taken as a reference, facilitating the discussion of the structural uncertainties. The LOTUS regressor was taken as a reference because of the use of indicators such as ENSO (El Niño-Southern Oscillation) and QBO (Quasi-Biennial Oscillation) within it that allow a refinement of the trend estimates. These indicators can have variable impacts depending on the latitude band, improving the accuracy of the trend analysis in some regions and reducing the influence in others (Olsen et al., 2019). In comparison between the**

regressors, the uncertainties were propagated in quadrature (Morgan & Henrion,1990; Stauffer et al., 2022)."

Following the reference added:

Morgan MG, Henrion M. The Propagation and Analysis of Uncertainty. In: *Uncertainty: A Guide to Dealing with Uncertainty in Quantitative Risk and Policy Analysis*. Cambridge University Press; 1990:172-219.

Olsen, M. A., Manney, G. L., & Liu, J. (2019). The ENSO and QBO impact onozone variability andstratosphere-troposphere exchangerelative to the subtropical jets. Journalof Geophysical Research: Atmospheres,124, 7379–7392. https://doi.org/10.1029/2019JD030435

*"The fact that trends computed using the unified ozonesonde dataset generally agree, except for 1978-1999 tropical stratospheric trends, with independent studies that include satellite data is an important message. This should be communicated in the abstract and will lead to the paper having greater impact."*

We have revised the abstract to emphasize the alignment of trends derived from the unified ozonesonde dataset with independent studies that incorporate satellite data. Below is the revised abstract: "This work discusses the impact of the sampling frequency on ozone partial pressure trends, estimating its impact at various latitudes and vertical layers in the upper troposphere/lower stratosphere (UT/LS) region. The trends are estimated in the periods 1978-1999 and 2000-2022, using a new unified dataset combining the ozonesounding profiles provided by SHADOZ (Southern Hemisphere ADditional OZonesondes), NDACC (Network for the Detection of Atmospheric Composition Change), and WOUDC (World Ozone and Ultraviolet Radiation Data Centre). These datasets are combined to offer adequate coverage at various latitudes and to enhance the estimation of anomalies and trends in ozone concentration on a global scale. The available measurements are classified into three groups (short coverage, medium coverage and long coverage) based on the temporal coverage of historical time series. The representativeness of medium coverage and long coverage cluster have been also studied at each altitude level using independent nadir-viewing satellite data. Parametric, non-parametric and multiple linear regressors are utilized to estimate trends and the related differences to quantify structural uncertainty. Significant trends for the period 1978-1999 are estimated for the Northern Hemisphere mid-latitude (NH), which shows a negative trend of 6% ±1% per decade in the layer 50-1 hPa and a negative trend of 9% ±1% per decade at 100-50 hPa, and for the Tropics (TR), which shows a positive trend of about 5% ±2% per decade at 50-1 hPa and 8% ±2% per decade at 100-50 hPa, respectively. Regarding the 2000-2022, the NH reveals a negative trend of 1% ±1% per decade at 50-1 hPa and a positive trend of 4% ±1% per decade at 100-50 hPa, and the TR shows a positive trend of 0.7 ±0.6% per decade at 50-1 hPa and a positive trend of 3% ±2% per decade at 100-50 hPa. Furthermore, the sampling error between the clusters was investigated, revealing a small

**effect of less than 2% per decade at 100-50 hPa and 0.5% per decade at 50-1 hPa for NH and about 9% per decade at 100-50 hPa and 4% per decade at 50-1 hPa for TR, as well as the structural uncertainty between the regressors used, 1.2% per decade at 100-50 hPa and 1.3% per decade at 50-1 hPa for NH and 6% per decade at 100-50 hPa and lower than 3% per decade at 50-1 hPa for TR. The trends computed using the unified ozonesonde dataset generally agree with independent studies based on more sophisticated datasets, including satellite data. However, discrepancies are observed in the tropical stratospheric trends during the period 1978-1999, primarily due to the limited availability of observations."**

*"Recommendation:*

*The general and specific comments that I have should result in mostly minor corrections and additions. After they are addressed, I recommend that this paper be published.*

*Line-by-Line and Technical Comments:*

*Abstract Line 15: Please list the three groups here."*

**Ok, please see our previous comment regarding the Abstract.**

*"Line 34: Suggest also citing Thompson et al., (2021; https://agupubs.onlinelibrary.wiley.com/doi/full/10.1029/2021JD034691) and Stauffer et al., (2024; https://acp.copernicus.org/articles/24/5221/2024/). Additionally, two HEGIFTOM papers (Van Malderen et al., 2025a, b) will be posted as preprints to the TOAR-II special issue not long after this review is submitted. Those will also be excellent and contemporary references."*

**OK, we will add these citations into our manuscript.**

*"Line 36: Change "utilized" to "applied"."*

**OK.**

*"Line 36: Change "alongside" to "to"."*

**OK.**

*"Line 46: To my knowledge there are only three stations that launch ~3 times per week: Uccle, Payerne, and Hohenpeissenberg. Weekly launches are much more typical."*

**We rephrased line 46 as: "However, unlike regular radiosoundings, ozonesonde measurements are performed less frequently, with profiles collected typically once a week."**

*"Line 95: In the case of a dataset existing in more than one archive, what is done to ensure that the homogenized ozonesonde data are being used? Are there cases where duplicate profiles were found in different archives with slightly different (e.g., non-homogenized vs. homogenized) ozone values?"*

To explain clearly the concept reported at lines 91-93 ("Duplications often arise when observations from the same station are transmitted to multiple networks, leading to discrepancies in transmission periods, data formats, and the number of data points, along with varying metadata. Additionally, not all networks provide identical number of data points at each level."), we added a phrase at line 103, specifying what is written in your comment: "The Unified Ozonesonde Dataset merges data as the same are provided by the considered networks: this may imply the merging of higher and lower quality vertical profiles (e.g. homogenized in SHADOZ vs. non-homogenized in WOUDC or NDACC) and the investigation of time series comprising different quality data from different data archives. The merging may potentially affect the bias between different profiles."

Regarding the cases where duplicate profiles were found in different archives with slightly different ozone values, two approaches are possible:

- Use samples from discarded profiles to densify the vertical sampling if they provide measurements at different pressure levels not available in the selected profiles.

- Choose the profile with the highest number of successful quality checks, discarding the other profiles.

The second approach was adopted as it can be considered more suitable and less prone to issues affecting one of the merged datasets.

The author used the first approach for his PhD thesis, available at https://hdl.handle.net/11563/177616, to study and compare the results with the literature by merging the missed samples from the discarded profiles.

*"Figures 4, 5, and 8: Please flip the axes so pressure is the y-axis"*

Ok. We modified these Figures, flipping the axes and reporting the ozone partial pressure in mPa, as follows:

Figure 4:

[Figure]

[Figure]

**Figure 5:**

[Figure]

**Figure 8:**

[Figure]

*"Line 200: How did you determine not to include SP and SH even though the trends are significant? What criteria were used to exclude them? Here is where the reporting of uncertainties might help."*

**Regarding SH, as shown in Figure 9, the structural uncertainties (between the considered regressors) vary in a large range between 6% and 104% for 1978-1999, while for 2000-2022 the variability is smaller and comparable to that of NH values only for 50-1 hPa and 100-50**

hPa pressure levels. Therefore, lines 201-205, have been rephrased as: "Due to the limited availability of LC and MC stations in the Southern Polar (SP) and Southern Hemisphere (SH) regions, the analysis presented hereinafter is not conducted on these two regions. Although the related estimated trends are significant, the paucity of available stations' number and data in these sectors substantially inflates the uncertainties on the estimated trends. Specifically, for SH, as shown in Figure 9, the number of stations stored in the database is not sufficiently representative. For the SP, despite the good representativeness of the available stations, the structural uncertainties (between the considered regressors) vary in a large range between 6% and 104% for 1978-1999, while for 2000-2022 the variability is smaller and comparable to that of NH values only for 50-1 hPa and 100-50 hPa pressure levels. Therefore, the comparative analysis is limited by the broad differences in the estimated trends among the considered regressors, suggesting that deceptive conclusions should be avoided for this latitude sector."

Below the updated Figure 9, where the representativeness of the different stations clusters is shown at different vertical ranges, using the ozone monthly gridded profiles derived from nadir-viewing satellite instrument merged in the MERGED-NP dataset, available in the Copernicus Climate Data Store (https://cds.climate.copernicus.eu/datasets/satellite-ozone-v1?tab=overview), that merges 5 vertical profiles products from UV sensors GOME, GOME2-A, GOME2-B, OMI and SCIAMACHY, still following the approach developed in Wheatherhead et al. (2017).

[Figure]

*"Line 219: What is meant by "5 ascents per month"? Is this for latitudinally-aggregated stations, or do you require an individual site to have at least 5 profiles in a given month over all years of the timeseries?"*

**We meant that an individual site has at least 5 profiles in a given month over all years of the time series. We rephrase line 219 as: "Completeness checks were performed to ensure that each individual site had at least 5 ascents per month in all years of the time**

**series, allowing for up to 5% of months not covered, as mentioned in the classification criteria;"**

*"Line 230: I am assuming the large majority of profiles that did not pass this check were from before 1990 or so, correct?"*

**The profiles that did not pass this check are 5439 before 1990 (29.75% of profiles that did not pass the check). In addition, considering that the total stored profiles in the database before 1990 is 12506, the percentage of these profiles that do not pass the check is 43.49%. So, asking the question, we can say that the large majority of profiles that did not pass this check were from before 1990. We rephrased line 230 as: "In this work, 79603 profiles were checked and only 18358 (23.06%) did not pass the check in certain atmospheric regions. The majority of profiles that did not pass the checks were from before 1990 (5439 profiles, 29.75%)."**

*"Table 4: Are these null values what you have removed based on previous quality checks? Please explain more what this means."*

**Not exactly: some of null values were flagged by QCs and others are not provided in the source files, e.g. uncertainty values are always provided by SHADOZ and, for a fraction of data, by NDACC. Instead, they are not available from WOUDC. Better caption, at line 236, may be: "List of the main variables within the unified database on which the percentage of null values is calculated. The latter are due to either flagging by applied QCs or because missing in the source file. For each variable, motivation for retrieving null values is reported."**

*"Line 269: One might argue that some closely located ozonesonde stations could be somewhat redundant for total column and stratospheric ozone. However, with differing launch schedules at the stations this density of stations remains important, and they are certainly not redundant for tropospheric measurements. Please make this clear in the paper."*

**Ok, we rephrased line 269 as follows: "Consequently, the additional stations within the LMC cluster may be deemed redundant for total column and stratospheric ozone in these regions, with preference given to utilizing data from the LC stations due to their higher-quality and correlated time series. However, redundant measurements must be considered relevant for tropospheric measurements."**

*"Section 3.2: Can you explain in more detail how the anomaly timeseries are constructed? Are the ozone values in a latitudinal band first averaged to construct a climatology, and then the anomalies are computed? Have you explored calculating anomalies for individual stations from their individual climatology, and then averaging the individual site anomalies within the latitudinal bands? This could help avoid some step changes resulting from differing station records – having stations come online, dropping out, etc."*

Regarding the first two questions, we rephrased lines 306-307 to be clearer as follows: "The average monthly anomalies are calculated using data from all stations within each latitudinal and vertical range, using the following formula:".

Regarding the question "Have you explored calculating anomalies for individual stations from their individual climatology and then averaging the individual site anomalies within the latitudinal bands?": The preference was to aggregate the data at the latitude level to avoid large uncertainties in trend estimates due to gaps in many individual series. This approach may ensure more reliable trend calculations by minimizing the impact of missing data at individual stations. However, in the future, to compare this method with the one used in this work.

*"Line 374: Change "As" to "For"."*

**OK.**

*"Table 9: The large ozone increases in the tropical stratosphere for 1978-1999 are really surprising, even with limited sampling, and based on Figure 12 they appear to be driven by the years ~1982-1988. Do you know what is happening here? Again, uncertainty reporting will help here."*

**Probably, this effect is due to the eruption of El Chichón in 1982, as I mentioned in Section 4.2, lines 521-533.**

*"Line 400: Change "Chapter" to "Section"."*

**OK.**

*"Line 414-415: Better to say "significant" rather than "valid"."*

**OK.**

*"Line 421: Change "an enhanced" to "more"."*

**OK.**

*"Line 427: Change "vanishing" to "inhibiting"."*

**OK.**

*"Line 440: I wouldn't use "richer data content" to describe the difficulty in detecting trends in the upper troposphere vs. the stratosphere. The high variability of ozone near the tropopause is mostly what makes it difficult to detect trends in that region."*

**We rephrased lines 440-441 as follows: "The most suitable vertical ranges for trend analysis are 50-1 hPa and 100-50 hPa, due to their more stable ozone concentrations and lower variability. In contrast, the high variability of ozone near the tropopause complicates the detection of trends in that region."**

*"Line 512: Change how the ozonesonde TCO dropoff is described. Rather than stating "apparent anomalous losses of ozone in the lower and middle stratosphere", note that it is an instrumental artefact or artefacts that cause low-biased ozone measurements in the stratosphere at some stations."*

**We rephrased lines 510-514 as: "Figure 12 also shows the sudden post-2013 total column ozone (TCO) "dropoff" of a few percent in the ozone anomalies due to one of the two instruments used for the ozone soundings (Stauffer et al., 2020; Stauffer et al., 2022; Nakano and Morofuji, 2023). This dropoff is attributed to instrumental artefacts that cause low-biased ozone measurements in the stratosphere at some stations, resulting in several percentage points deviation from the averages observed from 2014 to 2017."**

*"Line 568: It is just HEGIFTOM, not HEGIFTOM-II"*

**OK.**

*Appendix A: I note that for the Costa Rica station that the three closely located sites where that station has moved around to have been separated. Were these data taken from the SHADOZ archive? It is generally considered a single station, especially for measurements in the upper troposphere and stratosphere as used here.*

**The data for Costa Rica station are retrieved from SHADOZ and WOUDC. SHADOZ provides the data from Alajuela, Heredia and San Pedro considering them as a single station, WOUDC instead provides them in the form of three different stations. In the analysis, we only used the San Pedro data from WOUDC, classifying Alajuela and Heredia as short coverage.**

**In the updated manuscript, considering all the contributions from WOUDC dataset related to the Costa Rica station appropriately as from the same station, the data starts in July 2005 and ends in December 2020, with a record of 15 years and 6 months, compared to the 11 years previously obtained. All other related parts, text and calculations, in the manuscript have been adjusted accordingly.**

---

## Author Comment (AC2)

Dear Reviewer,

Thank you for your review and for taking the time to evaluate our manuscript. We are sorry to hear your negative judgment, but we have worked diligently to address your concerns through substantial revisions. We have extensively modified the paper also in response to your comments, and we hope these changes will help convince you otherwise. Below, we have included your comments in italics, along with our responses in bold.

*"General remarks:*

*This paper presents (i) the development of a merged ozonesonde data combining vertical ozonesonde profiles collected from the SHADOZ, NDACC and WOUDC data archives to obtain better coverage in space and time (1979-1999 and 2000-2022); (ii) the study of the impact of the sampling frequency on long term trends derived from this merged ozonesonde data set at four pressure segments (1-50 hPa, 50-100 hPa, 100-200 hPa and 200-300 hPa and five latitude bands (SP: 90-60°S; SH: 60-30°S; TR: 30°S-30 °N; NH: 30-60°N; SP: 60-90°N). Hereby, different linear statistical models have been applied to derive the trends and discussed on their representativeness and the impact of the sampling frequency on that."*

**We made several significant improvements to our study:**

1. **We added the trend estimation of ozone concentration based on the LOTUS multiple linear regression and compared with parametric and non-parametric linear regressors. This addition allows for a more comprehensive analysis of the trends.**

2. **We calculated the representativeness of the stations at various pressure levels (10 hPa, 50 hPa and 100 hPa) using the ozone monthly gridded profiles derived from nadir-viewing satellite instrument merged in the MERGED-NP dataset, available in the Copernicus Climate Data Store (), that merges 5 vertical profiles products from UV sensors GOME, GOME2-A, GOME2-B, OMI and SCIAMACHY, still following the approach developed in Weatherhead et al. (2017). This expands the characterization of the stations clusters showing their value to characterize the trends at different latitudes and for different vertical ranges.**

3. **We excluded the analysis of trends for the South Pole (SP) and Southern Hemisphere (SH), as suggested by the editor in his review, due to the limited data availability and the insufficient number of stations, which do not guarantee reliable trend estimates.**

4. **We calculated the uncertainties of the trends for non-parametric methods and the LOTUS using the bootstrapping method. This approach provides a more robust assessment of the trend significance.**

*"This merged ozonesonde dataset, as the authors assigned it as an "unified ozonesounding dataset, offering a harmonized and quality checked dataset that eliminates the necessity for climate researchers to download multiple data sets and etc...." (Line 563-566). This in itself is a big challenge to aim for and may be very helpful for the atmospheric research community. However, to achieve this goal the authors fail here in several important aspects. This so called "unified data set" is far from fulfilling the hard criteria to pretend to be a standard scientific dataset that can be used in climate research. Technically spoken, the authors have collected the sounding data from the different archives and combined them on a common grid with some statistical cleaning. However, this is neither original nor scientifically sufficient to be published in AMT."*

**The primary goal of our unified dataset is indeed to provide a single, globally comprehensive database that eliminates the need for researchers to download multiple datasets. However, we also aim to study that the data produced by this unification is consistent with those already published in the literature. To achieve this, we estimated trends in ozone concentration, which is the most critical variable in this dataset, and compared them with the trends reported in the literature, using a subsampled cluster (LC cluster) compared to the entire database (LMC cluster), based on the time series of each station. Our analysis shows that the trends calculated from our unified dataset are very close to those derived from satellite datasets, both in terms of magnitude and uncertainty, although we do not pursue any homogenization effort of the time series, in the same way made for the HEGIFTOM dataset.**

**An additional objective of this work is to facilitate the work of data users, who often face the challenge of selecting which stations to use in order to estimate ozone variability. It is important to quantify how these station selections might impact trend values and their associated uncertainties. By providing a unified dataset with consistent quality, we aim to support users in making informed decisions and reducing uncertainty in their analyses. The unified dataset also removes a large quantity of ozonesounding profiles not properly reported, but at present available thought the data archives of the considered networks.**

**Ultimately, this work serves as a tool to guide users, not necessarily the most expert, encouraging them to engage with the data critically and consider the challenges associated with using discrete samples to represent continuous ozone field.**

*"Completely missing are scientific efforts addressing harmonization, homogenization or solid quality assessment of the ozonesonde data, although, many efforts over last two decades have been undertaken within the ozonesonde experts' community and reported in scientific literature frequently. Many important references have not been discussed in a proper way or are completely missing in the paper. Obviously, the authors did not have consulted and evaluated the scientific literature on the performance of the different types of ozonesondes used over 4-5 decades of long term sonde records that are stored in the different data archives. This is clearly demonstrated that an important quality criterium as the total ozone column normalization factor has been completely ignored by the authors. Also in this context, the eventual impact on*

*the trends derived for those stations that apply the normalization factor through linear scaling of the measured vertical ozone profile. Neither the authors describe or discuss the fact that the ozonesonde community has undertaken a big homogenization effort to re-process all global ozonesonde records according the same principles (removal all known biases, consistent correction procedures, referring to same standard), including uncertainty calculation."*

**We acknowledge the importance of harmonization, homogenization, and quality assessment in creating a reliable dataset. The HEGIFTOM dataset has been mentioned in previous version and now more broadly described.**

**Furthermore, also in our response to the Editor, we clearly specified at line 103 that: "The Unified Ozonesonde Dataset merges data as the same are provided by the considered networks: this may imply the merging of higher and lower quality data (e.g. homogenized in SHADOZ vs. non-homogenized in WOUDC or NDACC) and the comparison of data from the same station provided under a different quality to different data archives, potentially affecting the bias between different data points.".**

**Additionally, we described the approach adopted for trend uncertainty calculation, based on the bootstrapping method. We added the following explanation to Section 3.2, at line 316, of the manuscript: "Uncertainty of the regression slopes, for the robust regressors, have been estimated using the bootstrapping method. Bootstrapping is a statistical technique that involves resampling the data with replacement to create numerous simulated samples. This method allows for the estimation of the sampling distribution of a statistic and provides measures of accuracy such as confidence intervals and standard errors (Tibshirani & Efron, 1993). By applying bootstrapping, we provide a confidence in the estimated ozone trends (Hou & Shen, 2022)."**

**The unified dataset has been implemented applying comprehensive data quality checks, as detailed in Section 2.4 of our manuscript. We also added a detailed explanation of how these quality checks are applied rephrasing the sentence at line 238 to provide more details: "The quality checks mentioned above are employed collectively to generate a single quality flag for each ozone profile. This flag indicates the percentage of successful data checks, with possible values ranging from 0 to 3. The plausibility and outliers' checks are used to exclude anomalous or implausible values from the database, while the remaining three checks assess the structural quality of the ozonesounding profiles. The final quality flag is determined by the number of structural checks passed, with higher values indicating better quality profiles."**

**Regarding the quality criterion based on the total ozone column normalization factor, we recognize that this factor is an important aspect of ozonesonde data quality. However, it could not be applied to all stations in our study due to variations in the availability and quality of total ozone column measurements across different sites.**

**We have included a discussion in the revised manuscript, at line 253, to explain the limitations and challenges associated with the application of the total ozone column**

normalization factor: "Furthermore, we acknowledge the importance of the total ozone column normalization factor as a quality criterion for ozonesonde data. However, the application of this factor was not feasible for all stations due to variations in the availability and quality of total ozone column measurements across different sites. The normalization factor, calculated as the ratio of the spectrophotometer total ozone column (TOC) and the ozonesounding TOC, requires consistent and reliable TOC measurements (Ancellet et al., 2022). In cases where such measurements were not available or were of insufficient quality, the normalization factor could not be applied (Tarasick et al., 2021; Stauffer et al., 2022). This limitation impacts the derived trends, as the normalization factor helps to correct for biases in the measured vertical ozone profiles. Consequently, the absence of this correction for certain stations may introduce uncertainties in the trend analysis (Stauffer et al., 2022; Ancellet et al., 2022)."

Finally, we reviewed the scientific literature on the performance of different types of ozonesondes. This includes studies on the characteristics, biases, and corrections of ozonesonde data. Specifically, we added citations to the following papers, as suggested by the Editor and anonymous reviewer 1, to improve the existing knowledge base on which our study is based:

- Hou, Y., & Shen, Z. (2022). Research Trends, Hotspots and Frontiers of Ozone Pollution from 1996 to 2021: A Review Based on a Bibliometric Visualization Analysis. *Sustainability*, *14*(17), 10898. https://doi.org/10.3390/su141710898.
- Tibshirani, R. J., & Efron, B. (1993). An introduction to the bootstrap. *Monographs on statistics and applied probability*, *57*(1), 1-436.
- World Meteorological Organization (WMO). Scientific Assessment of Ozone Depletion: 2022, GAW Report No. 278, 509 pp.; WMO: Geneva, 2022.
- Thompson, A. M., Stauffer, R. M.,Wargan, K., Witte, J. C., Kollonige, D.E., & Ziemke, J. R. (2021). Regional and seasonal trends in tropical ozone from SHADOZ profiles: Reference for models and satellite products. Journal of Geophysical Research: Atmospheres,126, e2021JD034691.
- Stauffer, R. M., Thompson, A. M., Kollonige, D. E., Komala, N., Al-Ghazali, H. K., Risdianto, D. Y., Dindang, A., Fairudz bin Jamaluddin, A., Sammathuria, M. K., Zakaria, N. B., Johnson, B. J., and Cullis, P. D.: Dynamical drivers of free-tropospheric ozone increases over equatorial Southeast Asia, Atmos. Chem. Phys., 24, 5221–5234, https://doi.org/10.5194/acp-24-5221-2024, 2024.
- Roeland van Malderen, Anne M Thompson, Debra E Kollonige, Ryan M Stauffer, Herman G J Smit, et al.. Global Ground-based Tropospheric Ozone Measurements: Reference Data and Individual Site Trends (2000–2022) from the TOAR-II/HEGIFTOM Project. *Atmospheric Chemistry and Physics*, 2025, ⟨10.5194/egusphere-2024-3736⟩. ⟨hal-04901618⟩
- Roeland van Malderen, Zhou Zang, Kai-Lan Chang, Robin Björklund, Owen R Cooper, et al. Ground-based Tropospheric Ozone Measurements: Regional

tropospheric ozone column trends from the TOAR-II/ HEGIFTOM homogenized datasets. *Atmospheric Chemistry and Physics*, 2025, ⟨10.5194/egusphere-2024-3745⟩. ⟨hal-04901762⟩

- Tarasick, D. W., Smit, H. G. J.,Thompson, A. M., Morris, G. A., Witte,J. C., Davies, J., et al. (2021). ImprovingECC ozonesonde data quality:Assessment of current methods andoutstanding issues. Earth and SpaceScience, 8, e2019EA000914. https://doi.org/10.1029/2019EA000914

- Ancellet, G., Godin-Beekmann, S., Smit, H. G. J., Stauffer, R. M., Van Malderen, R., Bodichon, R., and Pazmiño, A.: Homogenization of the Observatoire de Haute Provence electrochemical concentration cell (ECC) ozonesonde data record: comparison with lidar and satellite observations, Atmos. Meas. Tech., 15, 3105–3120, https://doi.org/10.5194/amt-15-3105-2022, 2022.

- Morgan MG, Henrion M. The Propagation and Analysis of Uncertainty. In: *Uncertainty: A Guide to Dealing with Uncertainty in Quantitative Risk and Policy Analysis*. Cambridge University Press; 1990:172-219.

- Olsen, M. A., Manney, G. L., & Liu, J. (2019). The ENSO and QBO impact onozone variability andstratosphere-troposphere exchangerelative to the subtropical jets. Journalof Geophysical Research: Atmospheres,124, 7379–7392. https://doi.org/10.1029/2019JD030435

*"At present within the HEGIFTOM activity these data have been collected and stored on an ftp-server (for details see HEGIFTOM website: https://hegiftom.meteo.be). Under the bottom line, the paper misses any ozonesonde expertise at all, and maybe the authors should have consulted or included such expertise in their study from the very beginning."*

We would like to clarify that our unified database already incorporates ozonesonde data from three major networks: SHADOZ, WOUDC, and NDACC, which are also included in the HEGIFTOM dataset. While we were aware of the HEGIFTOM, the same was presented by the authors as a published dataset when the data analysis for this paper was already started in the frame of the C3S activities.

*"The dilemma of sampling frequency and geographical coverage has been treated rather poorly. Almost only based on a few general statistical thresholds the different station ozonesonde records have been selected and classified in three different clusters (long, middle and short coverage resp.). However, a clear scientific rational is missing."*

We have revised the manuscript to include a more detailed explanation of the criteria and their application, at lines 164-170, as follows: "Data coverage plays a critical role in accurately estimating anomalies and trends while minimizing uncertainties. To address this, only stations with at least one ozonesonde profile available per month were included in the unified dataset. These stations were then grouped according to their monthly coverage, with a month deemed covered if at least one ozonesonde ascent was available.

Based on their temporal coverage, the 153 selected stations were classified into three categories:

- **Long coverage (LC): 30 stations with a continuous data time series of at least 20 years with at most 5% of months not covered.**

- **Medium Coverage (MC): 17 stations with a continuous data time series of at least 10 years, but not more than 20 years, with at most 5% of months not covered.**

- **Short coverage (SC): 106 stations with continuous data time series less than 10 years, with at most 5% of months not covered, or no data available between 1978 and 2022."**

**These thresholds were chosen to identify stations with the most reliable and extensive time series, which is key for estimating decadal trends. Each station's data was carefully investigated to improve the trend analysis.**

*"The discussion on representativeness through a correlation analysis of total ozone columns between a station and the neighboring points by utilizing the EP TOMS satellite data has been done rather poorly and it does not give any information on what the consequences would be for the different vertical (i.e. pressure) ranges. A more detailed correlation analysis could be obtained using other satellites deriving vertical profiles (e.g. MLS). Particularly, this would be necessary in under-sampled regions like in the tropics."*

**As already mentioned above, we calculated the representativeness of the stations at various pressure levels (10 hPa, 50 hPa and 100 hPa) using the ozone monthly gridded profiles derived from nadir-viewing satellite instrument merged in the MERGED-NP dataset, available in the Copernicus Climate Data Store (https://cds.climate.copernicus.eu/datasets/satellite-ozone-v1?tab=overview), that merges 5 vertical profiles products from UV sensors GOME, GOME2-A, GOME2-B, OMI and SCIAMACHY, still following the approach developed in Wheatherhead et al. (2017).**

**We rewrote section 2.5 as follows: "Before examining trends in the Upper Troposphere/Lower Stratosphere (UT/LS), the representativeness of the network was initially assessed to ensure an accurate capture of ozone variability across different latitudes. This assessment utilized an approach developed by Weatherhead et al. (2017), utilizing the MERGED-NP dataset, that merges 5 ozone monthly gridded vertical profiles products, derived from nadir-viewing satellite instrument, from UV sensors GOME, GOME2-A, GOME2-B, OMI and SCIAMACHY. The dataset is available in the Copernicus Climate Data Store (https://cds.climate.copernicus.eu/datasets/satellite-ozone-v1?tab=overview) and provides measurements from January 2003 to December 2020. The correlations were calculated for deseasonalized monthly values. Spatial representativeness of the LMC and LC datasets for ozone concentrations was quantified using Pearson correlation coefficient for each grid point with each unified station considering specific vertical ranges (170 hPa, 100 hPa, 50 hPa and 1 hPa) provided by**

**MERGED-NP dataset, as depicted in Figure 9. According to Weatherhead et al. (2017), a good correlation is indicated by a coefficient higher than 0.7."**

[Figure]

*Figure 9. Assessment of representativeness for unified stations, with the left column depicting the LC cluster at 10 hPa (upper panel), 50 hPa (central panel) and 100 hPa (lower panel) and the same for the right panel showing the LMC cluster, utilizing the MERGED-NP dataset from January 2003 to December 2020. The map illustrates the correlation level between each station and its surrounding area. The linear Pearson correlation coefficients were calculated between the deseasonalized monthly averages from the ozone profile measurements. A correlation value of 1 indicates a complete correlation with the surrounding grid points, while 0 signifies no correlation with neighbouring points. A good correlation is indicated by a coefficient higher than 0.7.*

Figure 9 reveals the stations' representativeness for the LC (left column) and LMC (right column) at three different pressure levels: 10 hPa (upper row), 50 hPa (central row) and 100 hPa (lower row). These levels were considered the most representative among those available in the MERGED-NP dataset (1 hPa, 5 hPa, 10 hPa, 50 hPa, 100 hPa, 170 hPa and 450 hPa), with respect to the vertical ranges chosen for trend estimation. Figure 9 highlights that the LMC cluster generally exhibits greater representativeness than the LC cluster, particularly in the TR and the SH. This difference is particularly evident at 10 hPa, where LC stations show significantly lower representativeness compared to LMC stations, especially in the TR. At 50 hPa and 100 hPa, the representativeness of both clusters is largely similar, with only minor variations. In the other latitudinal sectors, the difference between the LC and LMC clusters is not as evident as in the TR and SH sectors, resulting in the LC cluster having a similar representativeness to the LMC cluster in those regions.

Consequently, the additional stations within the LMC cluster may be deemed redundant for total column and stratospheric ozone for these regions, with preference given to utilizing data from the LC station due to its higher-quality correlated time series. However, with different station launch schedules, this density remains important and not redundant for tropospheric measurements. Furthermore, it is worth noting that, at NH, the correlation is strong for only a few regions (North America, Europe and Japan), as the LC and LMC clusters do not include stations in the other regions. Given the limited representativeness of the LMC and LC datasets for the entire total ozone band of NH, it is difficult to draw definitive conclusions about trends in this region. This limitation implies that NH trends calculated from these datasets may not fully capture the ozone variability in all areas within this latitude band. Consequently, trends observed in different vertical ranges should be interpreted with caution, recognizing the potential gaps in spatial coverage and representativeness.

Figure 9 also reveals that the representativeness of ozone at 10 hPa is lower compared to that at 50 hPa and 100 hPa, primarily due to greater spatial and temporal variability in the upper stratosphere. This variability is influenced by complex dynamic and chemical processes, such as the Brewer-Dobson circulation and photochemical reactions, which cause increased ozone variability. Additionally, the density of measurements at 10 hPa may be lower due to instrumental limitations, reducing the representativeness of stations to the surrounding areas. Planetary waves, which significantly impact the distribution of ozone in the upper stratosphere, can also contribute to reducing the correlation between station measurements and surrounding areas (WMO, 2022). However, despite these challenges, the representativeness at 10 hPa is considered acceptable, as correlation coefficients are above 0.7 for most of the sector, indicating good correlation with the surrounding grid points."

*"Further, all trends should have been reported with their uncertainties, even when the trend is little and should be not labeled with "NT"."*

As also reported in the replies to Reviewer 1 and the Editor, we have modified Tables 9-12 to include trend uncertainties, calculated via bootstrapping, for each regressor. Additionally, we have incorporated the LOTUS regressor along with the methods used previously. We have focused on reporting trends for the Northern Hemisphere (NH), the Tropics (TR), and the North Pole (NP), as data availability and the number of stations for the Southern Hemisphere (SH) and the South Pole (SP) are insufficient to ensure reliable trend estimates. These modifications aim to provide a clearer understanding of both significant and non-significant trends. Below, as also reported in our response to the editor's review, are the new Tables 9-12 with updated discussion at lines 391-442:

"

*Table 9. Trend estimates, with the uncertainty, as percentage per decade (% dec-1) obtained with the Linear (LIN), Least Absolute Deviation (LAD) and Theil-Sen (TS) and LOTUS regressors for all latitudinal sectors considered at the 50-1hPa vertical range for each cluster. Legend: Positive value (increasing trend); Negative value (decreasing trend); NT (No Trend).*

| 50-1hPa | | % dec$^{-1}$ | TR | NH | NP |
|---|---|---|---|---|---|
| **1978-1999** | **LC** | LIN | 7 ± 1 | -5.9 ± 0.5 | -10 ± 2 |
| | | LAD | 8 ± 1 | -5 ± 1 | -10 ± 6 |
| | | TS | 7 ± 2 | -6 ± 2 | -8 ± 8 |
| | | LOTUS | 8 ± 1 | -6.3 ± 0.2 | -10 ± 1 |
| | **LMC** | LIN | 4 ± 1 | -5.6 ± 0.5 | -14 ± 2 |
| | | LAD | 5 ± 2 | -5 ± 1 | -12 ± 3 |
| | | TS | 7 ± 3 | -6 ± 2 | -11 ± 5 |
| | | LOTUS | 4 ± 1 | -6.0 ± 0.2 | -14.9 ± 0.4 |
| **2000-2022** | **LC** | LIN | NT | -1.1 ± 0.4 | -6 ± 1 |
| | | LAD | NT | -1 ± 1 | -5 ± 2 |
| | | TS | NT | -1 ± 1 | -6 ± 2 |
| | | LOTUS | NT | -1.1 ± 0.1 | -5.4 ± 0.2 |
| | **LMC** | LIN | 0.7 ± 0.2 | NT | -5 ± 1 |
| | | LAD | 0.6 ± 0.6 | NT | -5 ± 2 |
| | | TS | 0.7 ± 0.7 | NT | -5 ± 2 |
| | | LOTUS | 0.9 ± 0.1 | NT | -4.6 ± 0.2 |

At the 50-1 hPa vertical range (Table 9) for the TR during the period 1978-1999, estimates display discrepancies of up to about 4% ±2% per decade. This discrepancy likely arises from the greater number of stations available for the LMC cluster, enhancing coverage and representativeness compared to the LC cluster, as discussed in Section 2.5.

For the NH, the clusters agree for the 1978-1999 period (with a maximum discrepancy less than 0.5% ±0.7% per decade), but in contrast for 2000-2022, where the trend of the LMC cluster, for the MK test, is not significant.

Finally, in the NP, trends correspond only in the 2000-2022 period (with a maximum discrepancy of 1% ±2% per decade for the TS regressor). For 1978-1999, differences among clusters nearly reach 5% ±2% per decade.

Table 10. Same as Table 9 but for the vertical range 100-50 hPa.

| 100-50hPa | | % dec⁻¹ | TR | NH | NP |
|---|---|---|---|---|---|
| **1978-1999** | **LC** | **LIN** | 12 ± 2 | -9 ± 1 | NT |
| | | **LAD** | 14 ± 7 | -8 ± 1 | NT |
| | | **TS** | 11 ± 10 | -9 ± 2 | NT |
| | | **LOTUS** | 11 ± 1 | -8.7 ± 0.4 | NT |
| | **LMC** | **LIN** | 8 ± 2 | -10 ± 1 | -13 ± 1 |
| | | **LAD** | 8 ± 7 | -10 ± 2 | -12 ± 3 |
| | | **TS** | 2 ± 9 | -9 ± 3 | -12 ± 4 |
| | | **LOTUS** | 8 ± 1 | -10.4 ± 0.4 | -12.9 ± 0.5 |
| **2000-2022** | **LC** | **LIN** | 4 ± 1 | 3 ± 1 | NT |
| | | **LAD** | 3 ± 2 | 4 ± 1 | NT |
| | | **TS** | 4 ± 3 | 4 ± 1 | NT |
| | | **LOTUS** | 2.5 ± 0.2 | 2.8 ± 0.1 | NT |
| | **LMC** | **LIN** | 4 ± 1 | 3 ± 1 | NT |
| | | **LAD** | 3 ± 2 | 3 ± 1 | NT |
| | | **TS** | 2 ± 3 | 3 ± 1 | NT |
| | | **LOTUS** | 1.6 ± 0.4 | 2.9 ± 0.1 | NT |

In Table 10, focusing on the 100-50 hPa range, the NH sector stands out with trend estimates in the period 2000-2022 showing 1% ±1% per decade sampling error and significant trend estimates, attributed to higher data density and greater variability in this vertical range. Nonetheless, even in NH, the LC and LMC clusters exhibit discrepancies in the period 1978-1999 of around 2% ±3% per decade in estimated trends. Conversely, the NP sector lacks significant trends, except for the LMC cluster during 1978-1999, with especially noticeable trends.

Additionally, the TR sector presents a considerable discrepancy, in the 1978-1999 period, of 9% ±10% per decade while, in the period 2000-2022, the LC and LMC clusters show

discrepancies of 2% ±3% per decade. Interestingly, the trends estimated by the LMC and LC clusters show low discrepancies among regressors with structural uncertainties of 2.4% ±3% per decade and 1.5% ±3% per decade for 1978-1999 respectively. The same cannot be said for 2000-2022 due to the discrepancies of 6% ±9% per decade for the LMC cluster and 3% ±10% per decade for the LC. This performance can be attributed to the paucity of available data for the TR, in 1978-1999, despite the LMC cluster comprised 8 more stations than LC (see Table 2). For conducting trend analysis using data from the unified database, high-quality measurements similar to those from the LC cluster are crucial. However, in regions with a limited number of LC stations, such as the tropics (TR), data from the LMC cluster can offer better representativeness than the LC cluster.

*Table 11. Same as Table 9 but for the vertical range 200-100 hPa.*

| 200-100hPa | | % dec$^{-1}$ | TR | NH | NP |
|---|---|---|---|---|---|
| **1978-1999** | **LC** | LIN | - | -10 ± 1 | NT |
| | | LAD | - | -10 ± 3 | NT |
| | | TS | - | -10 ± 5 | NT |
| | | LOTUS | - | -10 ± 1 | NT |
| | **LMC** | LIN | 10 ± 3 | -13 ± 1 | -19 ± 2 |
| | | LAD | 8 ± 9 | -14 ± 3 | -15 ± 4 |
| | | TS | 10 ± 15 | -13 ± 5 | -19 ± 6 |
| | | LOTUS | 10 ± 1 | -13 ± 1 | -19 ± 1 |
| **2000-2022** | **LC** | LIN | 10 ± 1 | 7 ± 1 | NT |
| | | LAD | 9 ± 4 | 7 ± 2 | NT |
| | | TS | 8 ± 4 | 5 ± 3 | NT |
| | | LOTUS | 7.5 ± 0.4 | 8.1 ± 0.2 | NT |
| | **LMC** | LIN | 6 ± 1 | 6 ± 1 | NT |
| | | LAD | 3 ± 4 | 6 ± 2 | NT |
| | | TS | 6 ± 4 | 8 ± 3 | NT |
| | | LOTUS | 2.3 ± 0.4 | 7.5 ± 0.2 | NT |

In Table 11, trend estimates in the NP sector are non-significant except for the LMC cluster in 1978-1999. Trend assessments for the LC cluster in the TR for the period 1978-1999 should be approached with caution due to the significantly small amount of data available before 1995 compared to subsequent years, which largely inflates uncertainties on decadal trends. However, it is important to acknowledge that the absence of a significant trend could indeed be a valid physical observation, particularly in regions or periods where ozone dynamics are more stable or less prone to significant change.

In addition, for 2000-2022, the LC cluster reveals a structural uncertainty of 2.5% ±4% per decade, and a discrepancy with the LMC of 6% ±4% per decade. Conversely, trend estimates for the LMC cluster in TR are significant, but exhibit considerable discrepancies among regressors, with structural uncertainty of 2% ±15% per decade for 1978-1999 and 3.7% ±4% per decade for 2000-2022. Moreover, the NH sector is the only one with all valid trends, although there is a disagreement between LC and LMC (up to about 4% ±5% per decade for 1978-1999 and 3% ±3% per decade for 2000-2022). Furthermore, it is worth considering the discrepancy between the results of the TS regressor with those of the LIN and LAD regressors (2% ±3% and 3% ±4%, respectively) that occurred for LC in the period 2000-2022. The LIN, LAD and LOTUS regressors agree with each other, reporting a structural uncertainty of 1.1% ±1% per decade, on the contrary TS where this uncertainty rises to around 3% ±3% per decade.

*Table 12. Same as Table 9 but for the vertical range 300-200hPa.*

| 300-200hPa | | % dec$^{-1}$ | TR | NH | NP |
|---|---|---|---|---|---|
| **1978-1999** | **LC** | **LIN** | 17 ± 3 | -10 ± 1 | 5 ± 5 |
| | | **LAD** | 19 ± 7 | -10 ± 4 | 11 ± 11 |
| | | **TS** | 17 ± 11 | -13 ± 6 | 10 ± 13 |
| | | **LOTUS** | 17 ± 1 | -10.2 ± 0.5 | 5 ± 2 |
| | **LMC** | **LIN** | NT | -12 ± 2 | - |
| | | **LAD** | NT | -11 ± 3 | - |
| | | **TS** | NT | -11 ± 6 | - |
| | | **LOTUS** | NT | -12.8 ± 0.4 | - |
| **2000-2022** | **LC** | **LIN** | 5 ± 1 | 6 ± 1 | NT |
| | | **LAD** | 3 ± 3 | 7 ± 2 | NT |
| | | **TS** | 4 ± 3 | 7 ± 4 | NT |
| | | **LOTUS** | 3.7 ± 0.2 | 7.6 ± 0.4 | NT |
| | **LMC** | **LIN** | NT | 6 ± 1 | NT |
| | | **LAD** | NT | 7 ± 2 | NT |
| | | **TS** | NT | 5 ± 3 | NT |
| | | **LOTUS** | NT | 7.2 ± 0.4 | NT |

The scenario for the 300-200 hPa layer (Table 12) is similar to the previous layer. Additionally, this vertical interval exhibits the fewest significant trends. The most suitable vertical ranges for trend analysis are 50-1 hPa and 100-50 hPa, due to their more stable ozone concentrations and lower variability. In contrast, the high variability of ozone near the tropopause complicates the detection of trends in that region. Table 13 highlights the

most and least significant disparities in the trends obtained by comparing the different non-parametric regression techniques used with the parametric regressor LOTUS, taken as a reference, facilitating the discussion of the structural uncertainties. The LOTUS regressor was taken as a reference because of the use of indicators such as ENSO (El Niño-Southern Oscillation) and QBO (Quasi-Biennial Oscillation) within it that allow a refinement of the trend estimates. These indicators can have variable impacts depending on the latitude band, improving the accuracy of the trend analysis in some regions and reducing the influence in others (Olsen et al., 2019). In comparison between the regressors, the uncertainties were propagated in quadrature (Morgan & Henrion,1990; Stauffer et al., 2022)."

*"Summarizing, in the present form the paper misses almost any scientific originality. Without having a solid quality assessment of the merged ozonesonde dataset and their representativeness, the paper misses the scientific base to be published in AMT, therefore, I rate the paper as scientifically poor and reject it for publication in AMT."*

We have taken the reviewer's comments very seriously and made substantial revisions to address the concerns raised. We hope that the revisions added to the paper demonstrate the scientific utility of this work for the ozone sounding data users, and we kindly request a reconsideration of the reviewer's assessment based on the improvements made.

---

## Author Comment (AC3)

*"As a representative for the ozonesonde network, I'm really grateful for your effort in filtering and cleaning up the ozonesonde data, which are available in three (major) existing international archives, but in different formats and not necessarily identical. The Unified Ozonesonde Dataset is certainly a step in the right direction to ease data access and data use!*

*As for every ground-based (or in-situ) dataset, one of the major challenges of the ozonesonde data is the spatial and temporal representativeness for measuring (here) stratospheric ozone. In this study, an attempt has been done to analyze its impact on stratospheric ozone trends by considering two different subsets of the Unified Ozonesonde Dataset: the "Large coverage" and Large and medium coverage" datasets, and this for different latitude bands."*

**Dear Editor, thanks for your effort in providing a detailed review with the clear intent to improve the qualiy of the content and of the data analysis. Below, we provide our replies (in bold) to each of your comments (shown in italics).**

*"General comment:*

*The authors should also give some remaining caveats associated with the Unified Ozonesonde Dataset:*

- *It should be clearly mentioned that "unified" is a different concept than "homogeneous". As you know, the ozonesonde community has been undertaken a homogenization activity to process all the ozonesonde data across the world according to the same principles (e.g. removal of known biases by referring all data to the same standards, consistent correction procedures for known effects, uncertainty calculation). These homogenized data are stored on a ftp-server with access details provided on the HEGIFTOM website, and also on SHADOZ, and for a handful of sites only in WOUDC or NDACC."*

**The following parts have been rephrased, improving their clarity:**

**Lines 48-49 in the Introduction, previously reported as "This study presents a methodology focused on reducing sampling errors in ozone partial pressure trends by merging existing ozonesounding datasets into a unified database", rephrased as: "This study presents a methodology focused on reducing sampling errors in ozone partial pressure trends by merging existing ozonesounding datasets into a unified adjusted database. The latter provides the ozonesonding data, removing redundancies and checking the ozone profiles with the aim to remove inconsistencies and reporting mistakes. The final data are not homogenized (i.e. bias-adjsuted) as made in the TOAR-II/ HEGIFTOM (Tropospheric Ozone Assessment Report/ Harmonization and Evaluation of Ground-based Instruments for Free-Tropospheric Ozone Measurements) dataset (van Malderen et al., 2025)".**

**Following the reference added:**

Van Malderen, R., Zang, Z., Chang, K.-L., Björklund, R., Cooper, O. R., Liu, J., Maillard Barras, E., Vigouroux, C., Petropavlovskikh, I., Leblanc, T., Thouret, V., Wolff, P., Effertz, P., Gaudel, A., Tarasick, D. W., Smit, H. G. J., Thompson, A. M., Stauffer, R. M., Kollonige, D. E., Poyraz, D., Ancellet, G., De Backer, M.-R., Frey, M. M., Hannigan, J. W., Hernandez, J. L., Johnson, B. J., Jones, N., Kivi, R., Mahieu, E., Morino, I., McConville, G., Müller, K., Murata, I., Notholt, J., Piters, A., Prignon, M., Querel, R., Rizi, V., Smale, D., Steinbrecht, W., Strong, K., and Sussmann, R.: Ground-based Tropospheric Ozone Measurements: Regional tropospheric ozone column trends from the TOAR-II/ HEGIFTOM homogenized datasets, EGUsphere [preprint], https://doi.org/10.5194/egusphere-2024-3745, 2025.

Lines 67-68, previously reported as "The dataset used in this work results from the merging of three existing datasets that give adequate data coverage in different latitude sectors, hereby facilitating more robust analyses of anomalies and trends at a global scale.", rephrased as: "The dataset used in this work results from the merging of three existing datasets enhancing data coverage at different latitudes, as available from single datasets, hereby facilitating analyses of ozone measurements at a global scale."

"

- *As a consequence of the previous point, the Unified Ozonesonde Dataset will merge data at a given site that have been processed differently (e.g. homogenized in SHADOZ vs. non-homogenized in WOUDC or NDACC; the Payerne data has been corrected for total ozone normalization in one archive, but not in the other archive), which can give rise to biases between data points. This should be mentioned.*"

To explain clearly the concept reported at lines 91-93 ("Duplications often arise when observations from the same station are transmitted to multiple networks, leading to discrepancies in transmission periods, data formats, and the number of data points, along with varying metadata. Additionally, not all networks provide identical number of data points at each level."), we added a sentence at line 103 to clarify this aspect: "The Unified Ozonesonde Dataset merges data as the same are provided by the considered networks: this may imply the merging of higher and lower quality vertical profiles (e.g. homogenized in SHADOZ vs. non-homogenized in WOUDC or NDACC) and the investigation of time series comprising different quality data from different data archives. The merging may potentially affect the bias between different profiles."

*"Although the primary focus and expertise of the authors lie in the data polishing to construct the Unified Ozonesonde Dataset, the authors also want to contribute to the calculation of stratospheric ozone trends from this dataset. They apply common approaches or assumptions, but no justification or interpretation is given why*

- *two different trend periods are considered (1978-1999, 2000-2022)"*

At line 358 we added a phrase to clarify the reason of trend estimation into two different period: "The decision to separate the analysis into two periods is based on the significant

changes in the atmospheric concentrations of ozone-depleting substances (ODS), as outlined by Petropavlovskikh et al. (2019).The 1978-1999 period is characterized by increasing levels of ODS, leading to ozone depletion, while the 2000-2022 period reflects the beginning of ozone recovery due to the implementation of the Montreal Protocol and subsequent amendments (Petropavlovskikh et al., 2019)."

- *a linear regression trend model is used*

At line 295, we added a sentence to justify the use of linear regression and, according to the other amendment made in the new version of the manuscript, also information on the LOTUS multiple linear regression (we added it to the analysis, please see comments below for the description), as follows: "Linear regression models have been used as a simple tool for diagnosing the difference between the trends of the data clusters. Additionally, the LOTUS non-linear trend model has been employed to capture more complex variations and uncertainties in the ozone data (Godin-Beekmann et al., 2022). The comparison between the LOTUS and the linear trends is also informative about the fraction of ozone variability which can be captured adding more predictors to the linear regression model."

In addition, in the Conclusion Section, we modified the sentence at lines 538-540: "Moreover, the trends have been estimated for different atmospheric layers and latitude ranges, using four linear regression methods (least-square, least absolute deviation, Theil-Sen and LOTUS), which have been compared both to evaluate structural uncertainties and the added-value obtained from the usage of multiple linear regression models."

Finally, we updated the sentence at lines 567-570: "Future work will be oriented to make comparisons with other regression models, such as the LOTUS (Petropavlovskikh et al., 2019; Godin-Beckmann et al., 2022), which is based on the classic multiple linear regression method, and with time series from the HEGIFTOM-II (Harmonization and Evaluation of Ground-based Instruments for Free Tropospheric Ozone Measurements) homogenized NDACC dataset (https://hegiftom.meteo.be/)." As: "Future work will be oriented to make comparisons with other time series, such as the TOAR-II/HEGIFTOM dataset (https://hegiftom.meteo.be/)."

- *zonal trends are considered (i.e. different trends for tropics, mid-latitudes, polar regions)*
- *different vertical trends are calculated (i.e. between 300-200, 200-100, 100-50, and 50-1 hPa)"*

We added the following sentences to line 131: "In line with previous studies (Petropavlovskikh et al., 2019), the trends are analyzed across different latitude bands to account for the varying influences of atmospheric dynamics and chemistry across regions, as well as across different vertical layers to capture the distinct dynamical and chemical processes that dominate at various altitudes. This approach facilitates a more accurate assessment of ozone trends by considering the specific characteristics and processes dominant in each latitude band, thus isolating the effects of these distinct

processes. For example, the Dobson-Brewer circulation plays a crucial role in ozone distribution, particularly in the tropics and mid-latitudes of the lower stratosphere, influencing observed trends (Petropavlovskikh et al., 2019). Additionally, vertical stratification allows for the analysis of the ozone profile at irregular pressure ranges, avoiding the need for interpolation to fixed levels, which could introduce additional uncertainty."

*"The authors should give an interpretation or explanation why trends differ (or not) between the two periods, between the different latitudinal zones, and between the different vertical ranges. The latest WMO Scientific Assessment of Ozone Depletion can be used as guidance here (https://ozone.unep.org/sites/default/files/2023-02/Scientific-Assessment-of-Ozone-Depletion-2022.pdf)."*

We added the following test to line 479 for explaining the difference between the two-period trend for NH-mid latitude: "The trends in Figures 10-11 reveal significant differences between the two periods considered. For the 50-1 hPa range (Figure 10), the trend for 1978-1999 is approximately 5% per decade lower than that for 2000-2022, due to the higher atmospheric concentrations of ozone-depleting substances (ODS) during the earlier period. Similarly, for the 100-50 hPa range (Figure 11), the trend for 1978-1999 is about 13% per decade lower than for 2000-2022, again due to the elevated ODS concentrations. In both cases, ozone recovery from the implementation of the Montreal Protocol is evident in the 2000-2022 period, although a slight negative trend is still observed for the 50-1 hPa range."

For the Tropics, at line 500, we added: "The trends for 50-1 hPa (Figure 12) and 100-50 hPa (Figure 13) show differences of 4% per decade and 5% per decade, respectively, between 1978-1999 and 2000-2022."

*"The assessment of the representativeness of the unified stations in terms of representing the total ozone column field (Fig. 9) is a nice feature, but also raises some important questions:*

- *what is the consequence of these correlations for the representativeness of the LMC and LC datasets for the ozone concentrations in the vertical ranges considered (300-200, 200-100 hPa, etc)? Should this analysis not be performed for those vertical ranges (e.g. making use of the MLS satellite vertical ozone retrievals) instead?"*
- *Can we draw the conclusion from this graph that LC dataset is not representative for the tropical total ozone band, while the LMC might be?"*
- *Can we draw the conclusion that neither the LMC or the LC datasets are representative for the entire NH mid-latitude total ozone band, and what does this mean for the NH mid-latitude trends calculated from these datasets in the different vertical ranges?"*

We decided to recalculate the representativeness of the stations at different pressures. Given our familiarity with the Copernicus Climate Data Store, we used ozone monthly gridded profiles derived from nadir-viewing satellite instrument merged in the MERGED-NP dataset: it merges 5 vertical profiles products from UV sensors GOME, GOME2-A,

**GOME2-B, OMI and SCIAMACHY. As a consequence, the whole section 2.5 was reshaped accordingly. The section is reported in reply to the Specific Comments of this document.**

*"On top of this correlation analysis, to assess the impact that the LMC and LC datasets have on the different vertical ozone trends, it would also be good to show in the manuscript (or provide in the supplementary material) the monthly anomaly time series (and calculated trends) for both datasets (i.e. LMC for Figs. 10-11 and LC for Figs. 12-13), as in the PhD Thesis this manuscript is based on. Those plots would be more illustrative and informative (i.e. better guidance for interpretation) than the tables 7 to 13). Based on those plots in your PhD Thesis, when comparing the ozone time variability (not only the linear trend) in different layers and for different latitude bands (Figs. 32-39), a very different time variability (in terms of features, spread in the variability) arises between the "Large coverage (LC)" and "Large and Medium coverage (LMC)" clusters, illustrating the strong dependence of this time variability on the chosen individual sites in the clusters. The merging of individual sites with different onsets of the ozonesonde observation program (as is done in the comparison between the LC and LMC clusters) gives a different spatial and temporal distribution of ozonesonde sites (or at least a different weighting of the contributions of individual sites) between the beginning and end of the periods for which ozone anomalies/trends have been calculated. This has of course consequences for the calculated trends, but this impact has not been investigated in enough detail."*

**To avoid a huge increase in the length of the main text, we added the plots of the monthly anomalies time series to the supplementary material. We added sentences to line 389 to indicate that the plots are available in the supplementary material: "In the supplementary material, the plots of the monthly anomaly time series, with the calculated trends, are shown (see Figures S1-S3). These plots illustrate the strong dependence of time variability on the chosen individual sites within the clusters. The merging of individual sites with different onsets of the ozonesonde observation program, as done in the comparison between the LC and LMC clusters, results in a different spatial and temporal distribution of ozonesonde sites. This variation leads to a different weighting of the contributions of individual sites between the beginning and end of the periods for which ozone anomalies and trends have been calculated. The supplementary figures provide a comprehensive view of how the variability and representativeness of the individual sites influence the overall trends, ensuring a thorough understanding of the spatial and temporal distribution effects." In addition, at line 454, we added: "Furthermore, it is worth noting that for the TRs the estimated trends for the LMC and LC clusters are not consistent. This is due to the greater number of stations (and therefore data) available for the LMC cluster compared to LC and, consequently, also to a better spatial and temporal representativeness, shown in Figure 9. As for the other latitudinal sectors considered, the trends of the two clusters are consistent with each other as well as their representativeness, thus showing that the subsampled LC cluster provides results consistent with those of the LMC cluster."Moreover, Reviewer 1 mentioned the Costa Rica station and the addition of the NDACC ozonesoundings for the period 2021-2022, which were not included in the initial**

version of the database. Additionally, the author received a private email regarding the Macquarie Island station, thanking them for identifying the absence of ozonesounding data for 2003 (the data had never been submitted to WOUDC). These ozonesoundings, which have since been submitted to WOUDC, were re-ingested into the database. As a result, the values in Table 1 have been updated as follows:

| DATABASE | # STATIONS | PERIOD | # PROFILES | UNCERTAINTY | % UNIFIED |
|---|---|---|---|---|---|
| SHADOZ | 14 | 1998-2022 | 9331 | Yes | 14.05% |
| NDACC | 33 | 1969-2022 | 50045 | Yes (for a minor fraction of data). | 46.51% |
| WOUDC | 148 | 1962-2022 | 99228 | No | 39.44% |

And Table 2:

| | # LC stations | # MC stations | # LMC stations |
|---|---|---|---|
| NP | 5 | 3 | 8 |
| NH | 14 | 5 | 19 |
| TR | 5 | 8 | 13 |
| SH | 3 | 0 | 3 |
| SP | 3 | 1 | 4 |

And Table 6:

| | Long coverage | Long and medium coverage | % of additions |
|---|---|---|---|
| NP | 9556 | 12717 | 33.08 |
| NH | 37818 | 42004 | 11.07 |
| TR | 6274 | 12380 | 97.32 |
| SH | 4407 | 4407 | 0 |
| SP | 6040 | 7273 | 20.41 |

The following lines are rewritten:

lines 109-110: "Thanks to the contribution of the three networks, the unified database has a total of 153 stations, for 1962-2022, and 105611 ozonesondes profiles."

Lines 167-170: see the comment about it in "specific comment" section.

Lines 230-231: "In this work, 79603 profiles were checked and only 18358 (23. 06%) did not pass the check in certain atmospheric regions. The majority of profiles that do not pass this check were from before 1990 (5439 profiles, 29.75%)."

"The discussion of the trend differences (based on the tables 7 to 13) between the different trend estimation tools and the different datasets or vertical ranges really needs the inclusion

*(in the table and in the interpretation) of the trend uncertainties! I know that the significance of the trend is mainly assessed by using the MK test statistic, but every used linear regression trend estimation tool provides an uncertainty estimation on the slope coefficient (i.e. trend), which should be displayed and used in the discussion between the trend estimates. In the trend discussion, it is also important to note that "no significant trend" can be a physical result, and the presence of a trend does not mean that one dataset is more "reliable", more "robust", or "outperforms" another one. In my specific comments, I'll point to these wordings."*

The uncertainty values of the linear regression parameters, for each robust regressor used, have been estimated using the bootstrapping approach. In Section 3.2 we added, at line 316, a brief explanation about this procedure: "Uncertainty of the regression slopes, for the non-parametric regressors, have been estimated using the bootstrapping method. Bootstrapping is a statistical technique that involves resampling the data with replacement to create numerous simulated samples. This method allows for the estimation of the sampling distribution of a statistic and provides measures of accuracy such as confidence intervals and standard errors (Tibshirani & Efron, 1993). By applying bootstrapping, we provide a confidence in the estimated ozone trends (Hou & Shen, 2022)." Values of the uncertainties on the regression slopes have been also displayed on the trend plots (Figure 10-13).

Following the reference added in this paragraph:

Hou, Y., & Shen, Z. (2022). Research Trends, Hotspots and Frontiers of Ozone Pollution from 1996 to 2021: A Review Based on a Bibliometric Visualization Analysis. *Sustainability*, *14*(17), 10898. https://doi.org/10.3390/su141710898.

Tibshirani, R. J., & Efron, B. (1993). An introduction to the bootstrap. *Monographs on statistics and applied probability*, *57*(1), 1-436.

In addition to the linear regressors, we reported the trend estimates calculated with the LOTUS regression model. So, at line 295, a brief explanation of this method was added as follows: "The fourth method employed is the LOTUS regression model, which uses multiple linear regression to estimate time series variability from explanatory variables such as the Quasi-Biennial Oscillation (QBO), El Niño-Southern Oscillation (ENSO), the 11-year solar cycle, Stratospheric Aerosol Optical Depth (sAOD), and a long-term trend. The LOTUS model includes independent linear trend terms evaluating long-term changes before and after the peak of ozone-depleting substances (ODS) around January 1997 and January 2000. The model applies to weightless ozone records, adjusting for seasonal variations using Fourier components (Godin-Beekmann et al., 2022; Petropavlovskikh et al., 2019)."

Furthermore, Tables 7-13 have been updated reporting the new values of the linear trends for the updated clusters of stations, as well as for adding the LOTUS estimates (SP and SH are not reported in table has clarified in the text of the manuscript), as follows:

**Table 7:**

| 1978-1999 | | TR | NH | NP |
|---|---|---|---|---|
| LC | 50-1hPa | I | D | D |
| | 100-50hPa | I | D | NT |
| | 200-100hPa | I | D | NT |
| | 300-200hPa | I | D | I |
| LMC | 50-1hPa | I | D | D |
| | 100-50hPa | I | D | D |
| | 200-100hPa | I | D | D |
| | 300-200hPa | NT | D | D |

**Table 8:**

| 2000-2022 | | TR | NH | NP |
|---|---|---|---|---|
| LC | 50-1hPa | NT | D | D |
| | 100-50hPa | I | I | NT |
| | 200-100hPa | I | I | NT |
| | 300-200hPa | I | I | NT |
| LMC | 50-1hPa | I | NT | D |
| | 100-50hPa | I | I | NT |
| | 200-100hPa | I | I | NT |
| | 300-200hPa | NT | I | NT |

We rephrased lines 370-381 to better align their content with the results of the MK test shown in Tables 7-8: "For the period 1978-1999, differences emerge between the two clusters across latitude ranges. In the tropics (TR), the LC and LMC clusters are consistent, showing similar trends, except for the 300-200 hPa layer. In the Northern Hemisphere (NH), both clusters exhibit consistent trends. However, in the Northern Polar (NP) region, the LC shows no significant trends in certain pressure ranges, likely due to the limited number of stations, which makes the use of the LMC cluster more appropriate.

For the period 2000-2022, at the NP, trends in the LMC cluster, despite being influenced by incorporated MC stations, do not differ significantly from those in the LC. Notably, in the TR, all clusters align, with the LC showing non-significant trends in two pressure ranges and the LMC providing an intermediary solution. In the NH, the LC cluster shows significant trends across pressure ranges, while the LMC cluster presents a non-significant trend in the 50-1 hPa vertical layer. In the NP, both clusters show significant trends only in the 50-1 hPa vertical layer."

**Table 9:**

| 50-1hPa | | % dec$^{-1}$ | TR | NH | NP |
|---|---|---|---|---|---|
| **1978-1999** | **LC** | **LIN** | 7 ± 1 | -5.9 ± 0.5 | -10 ± 2 |
| | | **LAD** | 8 ± 1 | -5 ± 1 | -10 ± 6 |
| | | **TS** | 7 ± 2 | -6 ± 2 | -8 ± 8 |
| | | **LOTUS** | 8 ± 1 | -6.3 ± 0.2 | -10 ± 1 |
| | **LMC** | **LIN** | 4 ± 1 | -5.6 ± 0.5 | -14 ± 2 |
| | | **LAD** | 5 ± 2 | -5 ± 1 | -12 ± 3 |
| | | **TS** | 7 ± 3 | -6 ± 2 | -11 ± 5 |
| | | **LOTUS** | 4 ± 1 | -6.0 ± 0.2 | -14.9 ± 0.4 |
| **2000-2022** | **LC** | **LIN** | NT | -1.1 ± 0.4 | -6 ± 1 |
| | | **LAD** | NT | -1 ± 1 | -5 ± 2 |
| | | **TS** | NT | -1 ± 1 | -6 ± 2 |
| | | **LOTUS** | NT | -1.1 ± 0.1 | -5.4 ± 0.2 |
| | **LMC** | **LIN** | 0.7 ± 0.2 | NT | -5 ± 1 |
| | | **LAD** | 0.6 ± 0.6 | NT | -5 ± 2 |
| | | **TS** | 0.7 ± 0.7 | NT | -5 ± 2 |
| | | **LOTUS** | 0.9 ± 0.1 | NT | -4.6 ± 0.2 |

We also rephrased the caption to align the new format of the table: "Table 9. Trend estimates, with the uncertainty, as percentage per decade (% dec$^{-1}$) obtained with the Linear (LIN), Least Absolute Deviation (LAD) and Theil-Sen (TS) and LOTUS regressors for all latitudinal sectors considered at the 50-1hPa vertical range for each cluster. Legend: Positive value (increasing trend); Negative value (decreasing trend); NT (No Trend)."

We also rephrased lines 395-454 as follows: "At the 50-1 hPa vertical range (Table 9) for the TR during the period 1978-1999, estimates display discrepancies of up to about 4% ±2% per decade. This discrepancy likely arises from the greater number of stations available for the LMC cluster, enhancing coverage and representativeness compared to the LC cluster, as discussed in Section 2.5.

For the NH, the clusters agree for the 1978-1999 period (with a maximum discrepancy less than 0.5% ±0.7% per decade), but in contrast for 2000-2022, where the trend of the LMC cluster, for the MK test, is not significant.

Finally, in the NP, trends correspond only in the 2000-2022 period (with a maximum discrepancy of 1% ±2% per decade for the TS regressor). For 1978-1999, differences among clusters nearly reach 5% ±2% per decade.

**Table 10. Same as Table 9 but for the vertical range 100-50 hPa.**

| 100-50hPa | | % dec$^{-1}$ | TR | NH | NP |
|---|---|---|---|---|---|
| **1978-1999** | **LC** | LIN | 12 ± 2 | -9 ± 1 | NT |
| | | LAD | 14 ± 7 | -8 ± 1 | NT |
| | | TS | 11 ± 10 | -9 ± 2 | NT |
| | | LOTUS | 11 ± 1 | -8.7 ± 0.4 | NT |
| | **LMC** | LIN | 8 ± 2 | -10 ± 1 | -13 ± 1 |
| | | LAD | 8 ± 7 | -10 ± 2 | -12 ± 3 |
| | | TS | 2 ± 9 | -9 ± 3 | -12 ± 4 |
| | | LOTUS | 8 ± 1 | -10.4 ± 0.4 | -12.9 ± 0.5 |
| **2000-2022** | **LC** | LIN | 4 ± 1 | 3 ± 1 | NT |
| | | LAD | 3 ± 2 | 4 ± 1 | NT |
| | | TS | 4 ± 3 | 4 ± 1 | NT |
| | | LOTUS | 2.5 ± 0.2 | 2.8 ± 0.1 | NT |
| | **LMC** | LIN | 4 ± 1 | 3 ± 1 | NT |
| | | LAD | 3 ± 2 | 3 ± 1 | NT |
| | | TS | 2 ± 3 | 3 ± 1 | NT |
| | | LOTUS | 1.6 ± 0.4 | 2.9 ± 0.1 | NT |

In Table 10, focusing on the 100-50 hPa range, the NH sector stands out with trend estimates in the period 2000-2022 showing 1% ±1% per decade sampling error and significant trend estimates, attributed to higher data density and greater variability in this vertical range. Nonetheless, even in NH, the LC and LMC clusters exhibit discrepancies in the period 1978-1999 of around 2% ±3% per decade in estimated trends. Conversely, the NP sector lacks significant trends, except for the LMC cluster during 1978-1999, with especially noticeable trends.

Additionally, the TR sector presents a considerable discrepancy, in the 1978-1999 period, of 9% ±10% per decade while, in the period 2000-2022, the LC and LMC clusters show discrepancies of 2% ±3% per decade. Interestingly, the trends estimated by the LMC and LC clusters show low discrepancies among regressors with structural uncertainties of 2.4% ±3% per decade and 1.5% ±3% per decade for 1978-1999 respectively. The same cannot be said for 2000-2022 due to the discrepancies of 6% ±9% per decade for the LMC cluster and 3% ±10% per decade for the LC. This performance can be attributed to the

paucity of available data for the TR, in 1978-1999, despite the LMC cluster comprised 8 more stations than LC (see Table 2). For conducting trend analysis using data from the unified database, high-quality measurements similar to those from the LC cluster are crucial. However, in regions with a limited number of LC stations, such as the tropics (TR), data from the LMC cluster can offer better representativeness than the LC cluster.

**Table 11. Same as Table 9 but for the vertical range 200-100 hPa.**

| 200-100hPa | | % dec$^{-1}$ | TR | NH | NP |
|---|---|---|---|---|---|
| **1978-1999** | **LC** | LIN | - | -10 ± 1 | NT |
| | | LAD | - | -10 ± 3 | NT |
| | | TS | - | -10 ± 5 | NT |
| | | LOTUS | - | -10 ± 1 | NT |
| | **LMC** | LIN | 10 ± 3 | -13 ± 1 | -19 ± 2 |
| | | LAD | 8 ± 9 | -14 ± 3 | -15 ± 4 |
| | | TS | 10 ± 15 | -13 ± 5 | -19 ± 6 |
| | | LOTUS | 10 ± 1 | -13 ± 1 | -19 ± 1 |
| **2000-2022** | **LC** | LIN | 10 ± 1 | 7 ± 1 | NT |
| | | LAD | 9 ± 4 | 7 ± 2 | NT |
| | | TS | 8 ± 4 | 5 ± 3 | NT |
| | | LOTUS | 7.5 ± 0.4 | 8.1 ± 0.2 | NT |
| | **LMC** | LIN | 6 ± 1 | 6 ± 1 | NT |
| | | LAD | 3 ± 4 | 6 ± 2 | NT |
| | | TS | 6 ± 4 | 8 ± 3 | NT |
| | | LOTUS | 2.3 ± 0.4 | 7.5 ± 0.2 | NT |

In Table 11, trend estimates in the NP sector are non-significant except for the LMC cluster in 1978-1999. Trend assessments for the LC cluster in the TR for the period 1978-1999 should be approached with caution due to the significantly small amount of data available before 1995 compared to subsequent years, which largely inflates uncertainties on decadal trends. However, it is important to acknowledge that the absence of a significant trend could indeed be a valid physical observation, particularly in regions or periods where ozone dynamics are more stable or less prone to significant change.

In addition, for 2000-2022, the LC cluster reveals a structural uncertainty of 2.5% ±4% per decade, and a discrepancy with the LMC of 6% ±4% per decade. Conversely, trend estimates for the LMC cluster in TR are significant, but exhibit considerable discrepancies among regressors, with structural uncertainty of 2% ±15% per decade for 1978-1999 and 3.7% ±4% per decade for 2000-2022. Moreover, the NH sector is the only one with all valid

trends, although there is a disagreement between LC and LMC (up to about 4% ±5% per decade for 1978-1999 and 3% ±3% per decade for 2000-2022). Furthermore, it is worth considering the discrepancy between the results of the TS regressor with those of the LIN and LAD regressors (2% ±3% and 3% ±4%, respectively) that occurred for LC in the period 2000-2022. The LIN, LAD and LOTUS regressors agree with each other, reporting a structural uncertainty of 1.1% ±1% per decade, on the contrary TS where this uncertainty rises to around 3% ±3% per decade.

**Table 12. Same as Table 9 but for the vertical range 300-200hPa.**

| 300-200hPa | | % dec$^{-1}$ | TR | NH | NP |
|---|---|---|---|---|---|
| **1978-1999** | **LC** | LIN | 17 ± 3 | -10 ± 1 | 5 ± 5 |
| | | LAD | 19 ± 7 | -10 ± 4 | 11 ± 11 |
| | | TS | 17 ± 11 | -13 ± 6 | 10 ± 13 |
| | | LOTUS | 17 ± 1 | -10.2 ± 0.5 | 5 ± 2 |
| | **LMC** | LIN | NT | -12 ± 2 | - |
| | | LAD | NT | -11 ± 3 | - |
| | | TS | NT | -11 ± 6 | - |
| | | LOTUS | NT | -12.8 ± 0.4 | - |
| **2000-2022** | **LC** | LIN | 5 ± 1 | 6 ± 1 | NT |
| | | LAD | 3 ± 3 | 7 ± 2 | NT |
| | | TS | 4 ± 3 | 7 ± 4 | NT |
| | | LOTUS | 3.7 ± 0.2 | 7.6 ± 0.4 | NT |
| | **LMC** | LIN | NT | 6 ± 1 | NT |
| | | LAD | NT | 7 ± 2 | NT |
| | | TS | NT | 5 ± 3 | NT |
| | | LOTUS | NT | 7.2 ± 0.4 | NT |

The scenario for the 300-200 hPa layer (Table 12) is similar to the previous layer. Additionally, this vertical interval exhibits the fewest significant trends. The most suitable vertical ranges for trend analysis are 50-1 hPa and 100-50 hPa, due to their more stable ozone concentrations and lower variability. In contrast, the high variability of ozone near the tropopause complicates the detection of trends in that region. Table 13 highlights the most and least significant disparities in the trends obtained by comparing the different non-parametric regression techniques used with the parametric regressor LOTUS, taken as a reference, facilitating the discussion of the structural uncertainties. The LOTUS regressor was taken as a reference because of the use of indicators such as ENSO (El Niño-Southern Oscillation) and QBO (Quasi-Biennial Oscillation) within it that allow a

refinement of the trend estimates. These indicators can have variable impacts depending on the latitude band, improving the accuracy of the trend analysis in some regions and reducing the influence in others (Olsen et al., 2019). In comparison between the regressors, the uncertainties were propagated in quadrature (Morgan & Henrion,1990; Stauffer et al., 2022).

Table 13. Maximum and minimum difference in estimated trends between different non-parametric regressors with the parametric LOTUS regressor, taken as reference, for all vertical ranges, latitudinal sectors, clusters and periods (NT means No Trends). The reported uncertainties were propagated in quadrature.

| | | | Maximum and minimum slope difference | | | | | | | |
|---|---|---|---|---|---|---|---|---|---|---|
| | | | 50-1hPa | | 100-50hPa | | 200-100hPa | | 300-200hPa | |
| | % dec$^{-1}$ | | MIN | MAX | MIN | MAX | MIN | MAX | MIN | MAX |
| 1978-1999 | LC | TR | 0 ± 1.4 | 1 ± 2.2 | 0 ± 10 | 3 ± 7 | - | - | 0 ± 3.1 | 2 ± 7.1 |
| | | NH | 0.3 ± 2 | 1.3 ± 1 | 0.3 ± 1 | 0.7 ± 2 | 0 ± 1.4 | 0 ± 5.1 | 0.2 ± 1.1 | 2.8 ± 6 |
| | | NP | 0 ± 2.2 | 2 ± 8 | NT | NT | NT | NT | 0 ± 5.4 | 6 ± 11.2 |
| | LMC | TR | 0 ± 1.4 | 3 ± 3.1 | 0 ± 2.2 | 6 ± 9 | 0 ± 3.1 | 2 ± 9 | NT | NT |
| | | NH | 0 ± 2.2 | 1 ± 1 | 0.4 ± 1 | 1.4 ± 3 | 0 ± 1.4 | 1 ± 3.1 | 0.8 ± 2 | 1.8 ± 6 |
| | | NP | 0.9 ± 2 | 3.9 ± 5 | 0.1 ± 1.1 | 0.9 ± 4 | 0 ± 2.2 | 4 ± 4.1 | - | - |
| 2000-2022 | LC | TR | NT | NT | 0.5 ± 2 | 1.5 ± 3 | 0.5 ± 4 | 2.5 ± 1.1 | 0.3 ± 3 | 1.7 ± 3 |
| | | NH | 0 ± 0.4 | 0.1 ± 1 | 0.2 ± 1 | 1.2 ± 1 | 1.1 ± 1 | 3.1 ± 3 | 0.6 ± 2 | 1.6 ± 1.1 |
| | | NP | 0.4 ± 2 | 0.6 ± 2 | NT | NT | NT | NT | NT | NT |
| | LMC | TR | 0.2 ± 0.2 | 0.3 ± 0.6 | 0.4 ± 3 | 2.4 ± 1.1 | 0.7 ± 4 | 3.7 ± 4 | NT | NT |
| | | NH | NT | NT | 0.1 ± 1 | 0.1 ± 1 | 0.5 ± 3 | 1.5 ± 2 | 0.2 ± 2 | 2.2 ± 3 |
| | | NP | 0.4 ± 1 | 0.4 ± 2 | NT | NT | NT | NT | NT | NT |

As shown in Table 13, the structural uncertainties among the regressors are minimal, as reveal the minimum and maximum differences, except for 1978-1999 in the NP at 300-200 hPa (LC cluster) and in the TR at 100-50 hPa (LMC cluster), and the amount of data and number of stations are adequate to characterize the vertical variability of ozone concentration. However, for both the LC and LMC clusters, the uncertainties in the TR sector for 1978-1999 are generally high, particularly at the vertical ranges of 100-50 hPa, 200-100 hPa, and 300-200 hPa, due to the scarcity of available data for this time period.

It is also interesting to note that most of the minimal differences are observed between the LOTUS and the LIN regression, indicating a conformity between the two in terms of both trend magnitude and uncertainty. This suggests that, in this study, the LIN regressor provides similar results to the LOTUS, making it a reliable method for ozone trend analysis.

On the other hand, the maximal differences are mostly found with the TS regressor, which is the most distant from the LOTUS in terms of both trend magnitude and uncertainty. This indicates that the TS may introduce more variations in the results than the LOTUS, highlighting the need to carefully consider this regression method used for trend analysis. The trend estimations provided in Tables 9-12 are influenced by the varying amounts of data available for each latitudinal area. The NH sector has the largest quantity of profiles, followed by the TR. In the 50-1 hPa and 100-50 hPa vertical ranges, the patterns of these two latitudinal sectors have been compared to the literature and the results are presented below."

Following the reference added:

Morgan MG, Henrion M. The Propagation and Analysis of Uncertainty. In: *Uncertainty: A Guide to Dealing with Uncertainty in Quantitative Risk and Policy Analysis*. Cambridge University Press; 1990:172-219.

Olsen, M. A., Manney, G. L., & Liu, J. (2019). The ENSO and QBO impact onozone variability andstratosphere-troposphere exchangerelative to the subtropical jets. Journalof Geophysical Research: Atmospheres,124, 7379–7392. https://doi.org/10.1029/2019JD030435

Finally, we added, at line 554, the following sentences: "The minimum structural uncertainties are observed between the LOTUS regressor, taken as a reference, with the LIN regressor suggesting a good reliability of the latter for trend analysis. On the contrary, the maximum structural uncertainties are found mainly with the TS regressor that could introduce greater variations in the results compared to the LOTUS."

*"Nowadays, most trend estimation tools for stratospheric ozone trend assessment make use of different proxies for attributing some dynamical behavior (QBO, ENSO, solar cycle, stratospheric AOD) in the ozone time series (e.g. LOTUS multiple linear regression model). In the ozone monthly anomaly time series plots (Figs. 10 to 13), which show a distinct non-linear behavior, only the impact of volcanic eruptions on the ozone data has been discussed. The authors might also discuss the possible impact of e.g. QBO and ENSO on the tropical ozone time series or, even better, use a multiple linear regression technique to give more credibility to the estimated trends and better enable the interpretation of the long-term time behavior."*

As reported in the previous comment, we decided to add the analysis with the LOTUS regression model. Figures 10-13 have been modified to present the trend estimate with the LOTUS regressor, for the two periods. See the specific comments for rephrasing of discussion on this matter.

*"At several places, the authors do not give enough details and more clarifications or interpretation is needed. In the specific comments, I'll give an overview.*

*Specific comments*

- *Lines 42-43: nowadays, ozonesonde measurements are not normalized to the total ozone column anymore"*

We rephrase lines 42-43 as follow: "Ozonesonde profiles failing to reach 10-20 hPa are often disregarded due to their inability to provide complete vertical profiles. This approach aligns with current practices, as modern ozonesonde measurements are no longer normalized to the total ozone column (Petropavlovskikh et al. 2019)."

"

- *In the introduction: mention the existence of the homogenized ozonesonde data (O3S-DQA activity or HEGIFTOM website) and the difference in concept between "unified" and "homogenized"."*

We added, at line 60, the following sentences to mention the existence of HEGIFTOM and the difference between unified and homogenized dataset: "In addition to the unified dataset, it is important to mention the existence of homogenized ozonesonde data, such as those provided by the Ozonesonde Data Quality Assessment (O3S-DQA) activity (Stauffer et al., 2022) and available on the HEGIFTOM website (https://hegiftom.meteo.be/datasets/ozonesondes). Homogenized datasets are processed to remove known biases and ensure consistency across different measurement sites and times, whereas unified datasets combine multiple sources to enhance coverage. The distinction between 'unified' and 'homogenized' datasets lies in their respective goals: unification aims to merge data for comprehensive coverage, while homogenization focuses on data consistency and accuracy."

"

- *Line 100: it is not clear which selection criterion is used here. Please specify "the dataset's maturity and availability of measurement uncertainties on ozone concentration profiles selection criteria"!"*

We rephrased lines 99-100 as follows: "If multiple duplicates meet the same number of QCs, the selection is guided by specific criteria, including the dataset's maturity (Thorne et al., 2017) and the availability of measurement uncertainties on ozone concentration profiles. These criteria ensure that the most reliable and accurate data are utilized for the analysis."

"

- *Figures 4 and 5: ozone(sonde) profiles are commonly displayed with the ozone partial pressure (in mPa) on the horizontal axis (x-axis), with the vertical coordinate (pressure here, decreasing in amplitude) on the vertical axis (y-axis)."*

OK, this has been fixed.

"

- *Lines 165-170. The criteria for the different datasets (LC, MC, SC) are incomplete and perhaps even not objective. As it is written now, Praha and Macquarie Island sites should be LC stations, not a SC and MC station, respectively. It looks like the criterion is "20 full years" for a SC station, and Macquarie Island is "degraded" to a MC station because it has a 3-month gap in 2003. This rises two questions: (i) how flexible have you been with those criteria for other sites (this is not obvious and some subjective assessment cannot be ruled out → make it objective), (ii) to which extent would a 3-month gap (or larger) has a significant impact on the calculated trend over an at least 20 year time period?"*

We acknowledge that the criteria for classifying the datasets (LC, MC, SC) necessitate further clarification. The classification is ultimately based on predefined limits on the data coverage to ensure consistency and objectivity. However, the classification criteria have been updated to avoid inappropriate exclusions of stations with a remarkable data record but with some small gaps. We revised the manuscript to include a more detailed explanation of these criteria and their application, at lines 164-170, as follows: "Data coverage plays a critical role in accurately estimating anomalies and trends while minimizing uncertainties. To address this, only stations with at least one ozonesonde profile available per month were included in the unified dataset. These stations were then grouped according to their monthly coverage, with a month deemed covered if at least one ozonesonde ascent was available. Based on their temporal coverage, the 153 selected stations were classified into three categories:

- **Long coverage (LC): 30 stations with a continuous data time series of at least 20 years with at most 5% of months not covered.**

- **Medium Coverage (MC): 17 stations with a continuous data time series of at least 10 years, but not more than 20 years, with at most 5% of months not covered.**

- **Short coverage (SC): 106 stations with continuous data time series less than 10 years, with at most 5% of months not covered, or no data available between 1978 and 2022."**

Regarding the specific cases of Praha and Macquarie Island, we classified these stations based on their overall data continuity. For Macquarie Island, the 3-month gap in 2003, after your comment, was not considered significant enough to classify it as an MC station, so its classification is changed in LC. For Praha, instead, we classified it as SC station because its data time series covers only the winter months from 1982 to 2022, providing no data for the other months. We applied these criteria consistently across all stations to maintain objectivity.

At Figure 6, we substituted Macquarie Island with Resolute station, classified as MC station, producing the following updated heatmaps:

[Figure]

Despite the long time series, Resolute was not classified as LC because it does not provide a continuous data time series of at least 20 years with at most 5% of months not covered. At lines 182-187, we updated the caption of Figure 6: "Data coverage of three stations: Praha (50.02°N, 14.45°E, 302m a.s.l., WMO code: 0-20000-0-11520), classified as SC (left panel), Resolute (74.72°N, 94.98°W, 64m a.s.l., WMO code: 0-20008-0-RSL), labelled as MC (centre panel), and South Pole (90°S, 0°E, 2835m a.s.l., WMO code: 0-20008-0-SPO), categorized as LC (right panel). Missing gaps in more than 5% of the data pushed Resolute station to be classified as MC. Praha station shows profiles available only during the first four months of the year, resulting in an SC classification. The South Pole station provides continuous data over 20 years, consistently with the LC classification.".

"

- *Figures 7 and 9: as your different latitudinal regions are marked by 30°, use an increment of 30° instead of 25° (Figure 7) or 50° (Figure 9) for the latitude axis."*

**OK**

"

- *Line 219: same comment as Reviewer 1: I don't understand the completeness check criterion, as most ozonesonde sites have weekly launches."*

We meant that an individual site has at least 5 profiles in a given month over all years of the time series. We rephrase line 219 as: "Completeness checks were performed to ensure that each individual site had at least 5 ascents per month in all years of the time

**series, allowing for up to 5% of months not covered, as mentioned in the classification criteria;"**

"

- *Table 3: the difference between the flagged NDACC percentages for the ozone partial pressures and ozone concentrations is striking. It is not because the ozone concentrations (in ppmv) are not provided, that they could not be easily calculated from the ozone partial pressures (in Pa). The flagged percentages for those two variables should be very similar, as for SHADOZ. So, what is the meaning (and relevance) of this 86.22% flagged NDACC data for the ozone concentrations?"*

**We reported incorrectly the percentage of flagged NDACC data for ozone concentrations, adding also the missing value. However, for the reasons mentioned above, we recalculated the percentages of flagged values, as follows:**

| Variable | Plausible physical range | Flagged values SHADOZ | Flagged values NDACC | Flagged values WOUDC |
|---|---|---|---|---|
| Air pressure | 0 Pa ≤ x ≤110000 Pa | <0.01% | <0.01% | <0.01% |
| Air temperature | 150K ≤ x ≤ 330K | 0.24% | 3.49% | 0.02% |
| Relative humidity | 0% ≤ x ≤ 120% | 7.56% | 17.06% | 12.17% |
| Ozone partial pressure | 0 Pa ≤ x ≤ 0.25 Pa | 1.85% | 8.78% | 0.57% |
| Ozone concentration | 0 ppmv ≤ x ≤ 30 ppmv | 1.86% | 8.82% | N.A. |
| Ozone partial pressure total uncertainty | N.D. | 18.01% | 73.35% | N.A. |

"

- *Line 229: vertical completeness check. "At least one point every 50 m in an ozone profile is required". With a typical rise rate of 5 m/s and an inherent response time of the ozonesonde of about 18-28s, the effective vertical resolution of the ozone profile is 100-150 m. So, how relevant is the 50m criterion?"*

**Thank you for this comment, the sentences at lines 229-230 was modified as follows: "Vertical completeness checks ensure that a minimum number of records are available for each vertical region covered by the ozonesoundings. Given that the typical rise rate of the ozonesounding is around 5 m/s, with an inherent response time of approximately 18-28 seconds, this check verifies that at least one data point is present every 100 meters in the profile."**

"

- *Line 238: Give more details on the single quality flag you generated for each ozone profile: what are the possible values and how have the different quality checks been compiled to obtain one single flag?"*

We rephrased the sentence at line 238, adding more details, as follows: "The quality checks mentioned above are employed collectively to generate a single quality flag for each ozone profile. This flag indicates the percentage of successful data checks, with possible values ranging from 0 to 3. The plausibility and outliers' checks are used to exclude anomalous or implausible values from the database, while the remaining three checks assess the structural quality of the ozonesounding profiles. The final quality flag is determined by the number of structural checks passed, with higher values indicating better quality profiles."

"

- *Fig 8: specify that the bad profile in the right panel is the brown one."*

OK, following the rephrased sentence: "In the right panel, the brown profile, which exhibits high noise levels, is likely the result of measurement errors introduced by the station operator."

"

- *Section 2.5: Several clarifications/additions are needed here. First, clearly mention if the correlations between the total ozone column measurements have been done for deseasonalized monthly values. Secondly, it is also important to add that/if the total ozone column from the ozonesonde profiles have been obtained by integrating the profile, and that the ozone column above the burst altitude is missing w.r.t. the TOMS-EP total ozone measurements. Thirdly, also add how the correlation maps in Fig. 9 have been obtained (you calculate for each grid point the correlation coefficient with each unified station, so which of those correlation coefficients is displayed in Fig. 9? Maximum correlation coefficient, as suggested by Weatherhead et al.?). And finally, mention from which correlation coefficient we can speak of good correlation (according to Weatherhead et al., I think it is higher than 0.7).*
- *Caption Fig. 9: you should mention that the (linear) Pearson correlation coefficients are calculated between the deseasonalized monthly averages from the total column ozone measurements (please confirm)."*

We followed the comments made in the General Comment section regarding representativeness, so we modified the code using the ozone monthly gridded profiles derived from nadir-viewing satellite instrument merged in the MERGED-NP dataset, available in the Copernicus Climate Data Store (https://cds.climate.copernicus.eu/datasets/satellite-ozone-v1?tab=overview), that merges 5 vertical profiles products from UV sensors GOME, GOME2-A, GOME2-B, OMI and

SCIAMACHY, still following the approach developed in Wheatherhead et al. (2017). We rewrote section 2.5 as follows:

"Before examining trends in the Upper Troposphere/Lower Stratosphere (UT/LS), the representativeness of the network was initially assessed to ensure an accurate capture of ozone variability across different latitudes. This assessment utilized an approach developed by Weatherhead et al. (2017), utilizing the MERGED-NP dataset, that merges 5 ozone monthly gridded vertical profiles products, derived from nadir-viewing satellite instrument, from UV sensors GOME, GOME2-A, GOME2-B, OMI and SCIAMACHY. The dataset is available in the Copernicus Climate Data Store (https://cds.climate.copernicus.eu/datasets/satellite-ozone-v1?tab=overview) and provides measurements from January 2003 to December 2020. The correlations were calculated for deseasonalized monthly values. Spatial representativeness of the LMC and LC datasets for ozone concentrations was quantified using Pearson correlation coefficient for each grid point with each unified station considering specific vertical ranges (170 hPa, 100 hPa, 50 hPa and 1 hPa) provided by MERGED-NP dataset, as depicted in Figure 9. According to Weatherhead et al. (2017), a good correlation is indicated by a coefficient higher than 0.7."

[Figure]

*Figure 9. Assessment of representativeness for unified stations, with the left column depicting the LC cluster at 10 hPa (upper panel), 50 hPa (central panel) and 100 hPa (lower panel) and the same for the right panel showing the LMC cluster, utilizing the MERGED-NP dataset from January 2003 to December 2020. The map illustrates the correlation level between each station and its surrounding area. The linear Pearson correlation coefficients were calculated between the deseasonalized monthly averages from the ozone profile measurements. A correlation value of 1 indicates a complete correlation with the surrounding grid points, while 0 signifies no correlation with neighbouring points. A good correlation is indicated by a coefficient higher than 0.7.*

Figure 9 reveals the stations' representativeness for the LC (left column) and LMC (right column) at three different pressure levels: 10 hPa (upper row), 50 hPa (central row) and 100

hPa (lower row). These levels were considered the most representative among those available in the MERGED-NP dataset (1 hPa, 5 hPa, 10 hPa, 50 hPa, 100 hPa, 170 hPa and 450 hPa), with respect to the vertical ranges chosen for trend estimation. Figure 9 highlights that the LMC cluster generally exhibits greater representativeness than the LC cluster, particularly in the TR and the SH. This difference is particularly evident at 10 hPa, where LC stations show significantly lower representativeness compared to LMC stations, especially in the TR. At 50 hPa and 100 hPa, the representativeness of both clusters is largely similar, with only minor variations. In the other latitudinal sectors, the difference between the LC and LMC clusters is not as evident as in the TR and SH sectors, resulting in the LC cluster having a similar representativeness to the LMC cluster in those regions.

Consequently, the additional stations within the LMC cluster may be deemed redundant for total column and stratospheric ozone for these regions, with preference given to utilizing data from the LC station due to its higher-quality correlated time series. However, with different station launch schedules, this density remains important and not redundant for tropospheric measurements. Furthermore, it is worth noting that, at NH, the correlation is strong for only a few regions (North America, Europe and Japan), as the LC and LMC clusters do not include stations in the other regions. Given the limited representativeness of the LMC and LC datasets for the entire total ozone band of NH, it is difficult to draw definitive conclusions about trends in this region. This limitation implies that NH trends calculated from these datasets may not fully capture the ozone variability in all areas within this latitude band. Consequently, trends observed in different vertical ranges should be interpreted with caution, recognizing the potential gaps in spatial coverage and representativeness.

Figure 9 also reveals that the representativeness of ozone at 10 hPa is lower compared to that at 50 hPa and 100 hPa, primarily due to greater spatial and temporal variability in the upper stratosphere. This variability is influenced by complex dynamic and chemical processes, such as the Brewer-Dobson circulation and photochemical reactions, which cause increased ozone variability. Additionally, the density of measurements at 10 hPa may be lower due to instrumental limitations, reducing the representativeness of stations to the surrounding areas. Planetary waves, which significantly impact the distribution of ozone in the upper stratosphere, can also contribute to reducing the correlation between station measurements and surrounding areas (WMO, 2022). However, despite these challenges, the representativeness at 10 hPa is considered acceptable, as correlation coefficients are above 0.7 for most of the sector, indicating good correlation with the surrounding grid points."

Following the reference added in this section:

World Meteorological Organization (WMO). Scientific Assessment of Ozone Depletion: 2022, GAW Report No. 278, 509 pp.; WMO: Geneva, 2022.

"

- *Section 3.3: this section does not provide what is announced in its first sentence (lines 317-318). As far as I understand from this sentence (and from my experience with statistical breakpoint detection tools as SNHT), you compare the individual ozonesonde time series with the corresponding unified ozonesonde time series at the same site and the SNHT test looks for a breakpoint in the difference time series (a significant shift in the mean of the segments before and after the breakpoint). The presence of such breakpoints in the unified vs. individual time series will bring very relevant information. Instead, in Table 5, you provide the mean and maximum values of the SNHT test statistic, but it is far from clear what those values represent and no real interpretation is given. Much more useful information would be the time epochs for which breakpoints in the difference time series are detected. A breakpoint (shift in the mean) in one time series w.r.t. the other would result in trend differences between the two time series, which is very relevant for the remaining of the paper. This section should be seriously rewritten to better reflect its aim, as was outlined in the first sentence."*

**We added, at line 339, the following sentences: "The SNHT test was applied to assess whether the merging operation of the data could increase the inhomogeneities in the time series. The aim is not to identify the structural breakpoint points of the merged series, as this would require a reference series, for example, or different information for properly homogenizing the time series avoiding false positives. A good approach for the homogenization is, when feasible, to perform a tailored data reprocessing based on metadata, in the same way as done within the HEGIFTOM dataset. Therefore, this section should be understood solely as an additional investigation into the potential increase in discontinuities within the time series, which may become apparent in the SNHT test statistics when a significant increase of the test statistics is found at the same point of discontinuity revealed in the non-merged series, or any other new discontinuity present in the merged series. The purpose of this approach is to focus on investigating these potential discontinuities rather than addressing the issue of homogenization of historical series, which has already been addressed in the HEGIFTOM dataset. It is also worth adding that the interpretation provided in the manuscript of the very few cases where the mean and maximum values of the SNHT statistic increases indicates that the merging procedure does not drastically affect the bias of the time series."**

- *Tables 7 to 13: I find it very strange that you include the latitudinal zones SP and SH in your analysis, as you write in lines 201-204: "Due to the limited availability of LC and MC stations in the Southern Polar (SP) and Southern Hemisphere (SH), the analysis presented hereinafter is not conducted on these two regions. Although the related estimated trends are significant, the scarcity of stations' number and data availability in these sectors substantially enhances the uncertainties on the calculated trends, making them unreliable." This is really a contradiction with including the trends of these regions in the tables. Therefore: drop those zones in the tables and discussion, you gave the arguments for doing so.*

- *Tables 9, 10, 11, 12, 13: add the (two-sigma) uncertainties of the trend estimations, also in the discussion!"*

**We modified the tables with these comments in mind and reported them in the General Comment section. Please see the specific comment in that section.**

"

- *Line 356: argue why you consider two different periods 1978-1999 and 2000-2022 for calculating trends."*

**We added a sentence to argue this at line 358, please see General Comment section.**

"

- *Line 357: in your analysis, you do not explicitly discriminate between geographical and temporal sampling impacts on trends, so add "temporal" here."*

**Ok. We modified the sentence as follows: "The purpose of the study that follows is to measure how geographical and temporal sampling affects trend estimates and, consequently, our understanding of the variability of ozone across time."**

"

- *Line 367-368: as you do not consider the MC cluster trends (time series not long enough, see line 172), this statement cannot be verified."*

**We rewrote the sentences to explain better this statement: "The significance of trends in the LMC cluster is influenced by whether the LC or MC clusters predominate, which is determined by the number of observations and stations available for each. Although trends from the MC cluster were not considered due to the insufficient length of the time series, the LMC cluster trends, which occasionally fall between the LC and MC results, suggest a potential improvement over the LC. This observation emphasizes the importance of considering the available data and the valuable insights that can be derived from the LMC cluster trends."**

"

- *Line 371: I don't have a clue what is meant with "In the TR, results resemble those of the SP."*

**We modified the discussion on the results shown in Tables 7-13 as reported in the General Comment section. Please see the specific comment in that section.**

"

- *Line 372 "the LC lacks significant trends in certain pressure ranges due to the limited number of stations", Line 375 "LC showing slightly better performance via the MK test", Line 381 "show comparable abilities to estimate significant changes", Line 418 "This*

*performance improvement", Line 425 "trend estimates are considered unreliable", Line 429 "Notably, the performance of the LMC cluster in the TR sector outperforms that of the LC cluster in this layer", Line 440 "providing a more robust dataset for trend estimation, ...: all these statements assume that no trend means a worse performance of a dataset for trend calculation, and do not take into account that no significant trend can be a true, physical result for some zones, vertical ranges or time periods. The entire discussion in this section should be realigned in this sense. As mentioned earlier, the comparison of the monthly anomaly time series plots for the two datasets (LC vs. LMC) will reveal more insight whether or not they both display a similar time variability (or trend)."*

**We agree that the interpretation of "no significant trend" should be carefully addressed, as it may indeed reflect a true physical result rather than a deficiency in the dataset. In light of this, we will revise the language in the sections you've highlighted to better align with this understanding.**

**For example, when we mention that "the LC lacks significant trends in certain pressure ranges due to the limited number of stations" (Line 372), we will clarify that this could reflect a lack of detectable trends in those regions rather than indicating a poorer performance of the dataset. Similarly, phrases such as "LC showing slightly better performance via the MK test" (Line 375) and "trend estimates are considered unreliable" (Line 425) will be revised to acknowledge that the absence of a significant trend could indeed be a valid physical observation, particularly in regions or periods where ozone dynamics are more stable or less prone to significant change.**

**Moreover, we also recognize that the presence or absence of significant trends in the LC and LMC clusters can be influenced by data availability and the inclusion of the MC cluster. If a trend is observed in the LC cluster but not in the LMC cluster, the issue may stem from the additional data provided by the MC cluster, which could affect the significance of the trend. Conversely, if a trend is observed in the LMC cluster but not in the LC cluster, it may be due to the limited number of data points in the LC cluster, which may not provide enough statistical power to produce a significant trend. In both cases, the non-significance of the trend suggests that the observed changes could be due to variability or insufficient data, rather than a clear, reliable pattern.**

**We added sentences at line 370 for clarity: "It is also important to note that the presence or absence of significant trends in the LC and LMC clusters can be influenced by the data availability and the inclusion of the MC cluster. If a trend is observed in the LC cluster but not in the LMC cluster, the issue may stem from the additional data provided by the MC cluster, which could affect the significance of the trend. Conversely, if a trend is observed in the LMC cluster but not in the LC cluster, it may be due to the limited number of data points in the LC cluster, which may not provide enough statistical power to produce a significant trend. In both cases, the non-significance of the trend suggests that the observed changes could be due to variability or insufficient data, rather than a clear, reliable pattern."**

Additionally, the comparison of monthly anomaly time series plots between the LC and LMC datasets, as you suggested, has been incorporated to provide more context on how these datasets compare in terms of time variability and trend estimation.

See comments about the results of Table 7-13 in the General Comment section.

"

- *Line 420-421: what is meant with "enhanced representativeness". This cannot be deduced from the trend analysis presented here, right?"*

We rewrote the sentence as: "However, in sectors with few LC stations, such as in the TR, data from the LMC cluster can ensure more representativeness than the LC cluster.". See comments about the results of Table 7-13 in the General Comment section.

"

- *Line 439-440: "The most suitable vertical ranges for trend analysis are 50-1 hPa and 100-50 hPa due to their richer data content, providing a more robust dataset for trend estimation." It is not the amount of points in the vertical range that plays a role, but the dispersion/variability between the points, because the means are considered to calculate trends. If you want to make a comparison between the different vertical ranges, the standard deviation of the means would be a better metric to compare instead of the number of data points."*

We rephrased the lines 430-440 as follows: "

We rephrased lines 440-441 as follows: "The most suitable vertical ranges for trend analysis are 50-1 hPa and 100-50 hPa, due to their more stable ozone concentrations and lower variability. In contrast, the high variability of ozone near the tropopause complicates the detection of trends in that region."

See comments about the results of Table 7-13 in the General Comment section.

"

- *Section 4.1: when comparing your trends with the estimates from other studies, you only consider the trend uncertainties from the other studies; include yours in the comparison as well."*

OK. We rewrote the lines 456-487, including our uncertainties and the new figures: "Figures 10-11 illustrate the computed trends for the NH sector within the 50-1 hPa and 100-50 hPa vertical ranges. For this latitudinal sector, the LC cluster was selected for the following trend analysis.

[Figure]

Figure 10. Estimated trends for the NH latitudinal sector in the 50-1 hPa vertical range using LC cluster data. On the left is the 1978-1999 trend, while on the right is the 2000-2022 trend. The legend shows the resulting percentage per decade (% dec$^{-1}$) with uncertainty for each regressor utilized. Finally, at the bottom centre, the MK test result and the slope coefficients for each regressor are shown.

[Figure]

Figure 11. Same as Figure 10 but for the vertical range100-50hPa.

The trends depicted in Figures 10 and 11 can be cross-referenced with the ozone trends outlined in Petropavlovskikh et al. (2019) and Sofieva et al. (2021), where more sophisticated regression models and broader dataset, including both in situ soundings and remote sensing (ground-based and satellite) measurements, were used. In Petropavlovskikh et al. (2019), both satellite data (from SBUV Merged Ozone Data Set (SBUV MOD), SBUV Cohesive data set (SBUV COH), GlobalOZone Chemistry And Related trace gas Data records for the Stratosphere (GOZCARDS), Stratospheric Water and OzOne Satellite Homogenized (SWOOSH),Global Ozone Monitoring by Occultation of Stars (SAGE-GOMOS), and Optical Spectrograph and InfraRed Imaging System(SAGE-OSIRIS)) and data from ground-based stations (from the NDACC, WOUDC and SHADOZ ozonesoundings, LIDAR, Microwave, FTIR and Brewer/Dobson Umkehr) were used to

estimate the trends; in Sofieva et al. (2021), instead, only satellite data from the MErged GRIdded Dataset of Ozone Profiles (MEGRIDOP) were used. In addition, in Petropavlovskikh et al. (2019), LOTUS regression was employed to estimate trends, incorporating a lag-1 autocorrelation correction. Conversely, Sofieva et al. (2021) utilized multiple linear regression, with autocorrelation addressed using the Cochrane-Orcutt transformation (Cochrane and Orcutt, 1949).

The trends in Figures 10-11 reveal substantial differences compared to the two periods considered. For 50-1 hPa (Figure 10), the trend for 1978-1999 is approximately 5% per decade less than 2000-2022, due to the higher level of atmospheric ODS concentrations. For 100-50 hPa (Figure 11), 1978-1999 is about 13% per decade lower than 2000-2022 for the same reason described above. In both cases, the ozone recovery due to the implementation of the Montreal Protocol in the period 2000-2022 is evident, although a slightly negative trend is estimated for 50-1 hPa.

Regarding the 1978-1999 period as presented in Petropavlovskikh et al. (2019), negative trends of 5% per decade at 50-1 hPa and 10% per decade at 100-50 hPa were observed in the NH lower stratosphere. These estimates align with the trends identified in this study: a negative trend of 6% ±1% per decade at 50-1 hPa and 9% ±1% per decade at 100-50 hPa. For the period 2000-2022, the trend reported by Petropavlovskikh et al. (2019) at 50-1 hPa was negative 1% per decade, and similar to the results of the LC cluster (-1% ±1% per decade). Similarly, at 100-50 hPa, the reported negative trend of 2% per decade, with an uncertainty range of ±7%, is consistent with the LC cluster value (4% ±1% per decade positive trend), falling within the above uncertainty range. Additionally, Sofieva et al. (2021) present a 1-2% per decade negative trend for the NH lower stratosphere at 50-1 hPa, consistent with the trends shown with the LC cluster for the period 2000-2022. Indeed, the result presented by Sofieva et al. (2021) is based on a 15-year dataset from 2003 to 2018, which is comparable with the period 2000-2022 of this study."

Furthermore, we modified the sentence at lines 545-549, as follows: "The difference between the trends for the clusters reveals that usage of the smallest but longer cluster, the "long coverage" cluster (LC), provides the largest number of significant trends in the Northern Hemisphere mid-latitude (NH) at the different vertical and latitude ranges, with a small effect of the spatial sampling, lower than 2% per decade at 100-50 hPa and 0.5% per decade at 50-1 hPa, and the smallest structural uncertainties, not exceeding 1.2% per decade at 100-50hPa and 1.3% per decade at 50-1 hPa."

And the sentence at lines 555-557: "In the NH, there is a negative trend of 6% ±1% per decade for the period 1978-1999 at 50-1 hPa layer, reaching a negative trend of 9% ±1% per decade at 100-50 hPa, and a negative trend of 1% ±1% per decade for the period 2000-2022 at 50-1 hPa, with a positive trend of 4% ±1% per decade at 100-50 hPa."

"

• *Section 4.2, lines 502-503: which tropical stations have data before 1998? Mention those!"*

As for the NH, we rewrote the lines 489-533, including our uncertainties, the new figures and the tropical stations that have data before 1998: "Figures 12-13 illustrate the trends computed for the TR sector within the 50-1 hPa and 100-50 hPa vertical ranges, respectively, for the periods 1978-1999 and 2000-2022. As previously mentioned, the LMC cluster was selected for the trend analysis in the tropics because of its greater spatial representativeness in this sector compared to the LC cluster.

[revised manuscript text omitted]

Furthermore, we modified the sentence at lines 551-554, as follows: “For this reason, in the Tropics, it is recommended the usage of the "long and medium coverage" cluster (LMC), which provides reliable trend estimation and better spatial representativeness compared to LC, with sampling errors up to about 9% at 100-50 hPa and 4% at 50-1 hPa and structural uncertainties of 6% at 100-50 hPa and lower than 3% at 50-1 hPa.”

And the sentence at lines 557-559: “For the Tropics sector in the period 1978-1999, a positive trend of about 5% ±2% per decade at 50-1 hPa is found, reaching 8% ±2% per decade at 100-50 hPa and, for 2000-2022, a positive trend of 0.7% ±0.6% per decade at 50-1 hPa, and a positive trend of 3% ±2% per decade at 100-50 hPa.”

**Other addition in the paper:**

**At the end of Sector 4.1, on line 487, we have added a further citation to a study that explains the anomalous increase in ozone concentration observed in 2011, as follows: "Furthermore, Figures 10-11 reveal an anomalous increase in ozone concentration in 2011. This phenomenon can be attributed to a reduction in ozone destruction due to minor sudden stratospheric warmings. According to the study by de Laat and van Weele (2011), these warmings led to a decrease in the efficiency of ozone depletion processes, resulting in higher ozone levels. The study highlights that the observed reduction in ozone destruction was significant enough to cause a noticeable increase in ozone concentrations during this period."**

**Following the reference added:**

de Laat, A., van Weele, M. The 2010 Antarctic ozone hole: Observed reduction in ozone destruction by minor sudden stratospheric warmings. Sci Rep 1, 38 (2011). https://doi.org/10.1038/srep00038